# Weighted Deep Ensemble Under Misspecification

## Abstract

Deep neural networks are supported by the universal approximation theorem, which guarantees that sufficiently large architectures can approximate smooth functions. In practice, however, this guarantee holds only under restrictive conditions, and violations of these conditions give rise to model misspecification. We categorize such misspecification into three sources: variable misspecification, arising from insufficiently informative features; structural misspecification, stemming from the limited width and depth of networks that cannot fully capture the underlying complexity; and inherent misspecification, occurring when the true model possesses properties such as discontinuities that cannot be faithfully represented. To mitigate the impact of these forms of misspecification, ensemble methods have become a common strategy for enhancing predictive performance. However, standard ensembles composed of identically architected and equally weighted models may suffer from "collective blindness", where shared errors are amplified and lead to systematically biased predictions with high confidence. To mitigate this issue, we introduce weighted deep ensemble method that learns the optimal weights. We prove that our method provably attains the convergence rate of the best single model in the ensemble and asymptotically achieves oracle-level predictive risk. To the best of our knowledge, this is the first work to provide rigorous theoretical guarantees for weighted deep ensemble under both well-specified and misspecified settings.

## 1 Introduction

Model misspecification in statistics arises from the omission of relevant variables, inclusion of irrelevant variables, incorrect functional forms and incorrect distributional assumptions (Maasoumi, 1990; White, 1982). When a model suffers from such misspecification, the best possible approximation $f^* \in \mathcal{F}$, with $\mathcal{F}$ denoting the function class used for estimation, may incur a significant approximation error from the true function $f_0$. In deep learning, the neural networks are always assumed to be well-specified (Barron, 1994; Elbrächter et al., 2019). As shown in the universal approximation theorem, sufficiently large neural networks have the ability to approximate any continuous function, which in principle allows the approximation error $\|f_0 - f^*\|$ to approach zero (Hornik et al., 1989; Park et al., 2020; Lu et al., 2021). Therefore, existing studies always focus on overcoming challenges in optimization and estimation errors (Barron, 1994; Soltanolkotabi et al., 2018; Adcock & Dexter, 2021).

However, the assumption that neural networks are well-specified is frequently violated in practice, as model misspecification is common. Unlike in traditional statistics, misspecification in deep learning manifests in several distinct forms. First, it may arise from an information deficit, where the input features and their latent representations lack the necessary information to capture the true data-generating process. Second, practical constraints on network depth and width impose finite capacity, leading to non-negligible approximation error when the true function is highly complex. Finally, misspecification can occur when the true function has properties such as discontinuities, which cannot be exactly represented by neural networks and can only be approximated with non-vanishing error at the discontinuity points.

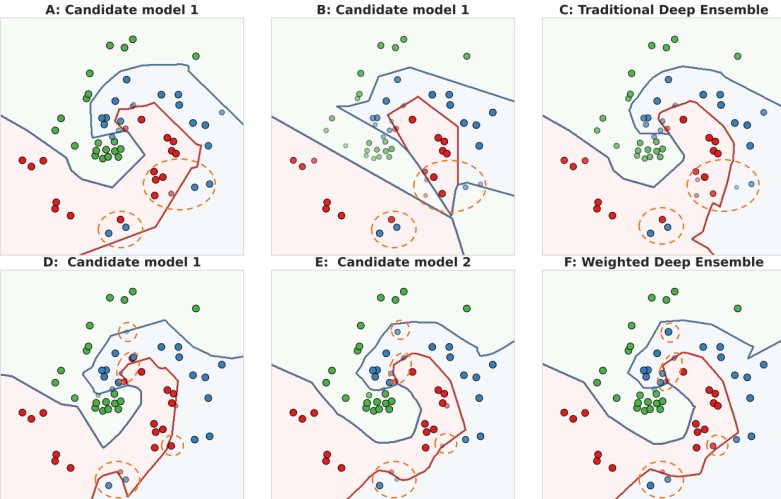

Figure 1: Comparison of the decision boundaries and confidence levels of different ensemble methods, with darker shading indicating higher confidence. The Traditional Deep Ensemble (C) shows "collective blindness" by having lower confidence in correctly classified areas but higher confidence in the misclassified areas, e.g., blue points within the red region. Weighted Deep Ensemble (F) corrects these errors by maintaining high confidence in correct areas while showing low confidence at uncertain boundaries. The key difference is highlighted in the circled area.

"All models are wrong, but some are useful (Box, 1976)." Ensemble methods are the most intuitive method to leverage the useful parts of multiple wrong models (Fort et al., 2019; Huang et al., 2024). However, it still raises a critical question: if a neural network model is misspecified, can the estimators from an ensemble of such models still be trusted? We find that traditional deep ensembles, which consist of models with identical architectures (Hansen & Salamon, 2002; Lakshminarayanan et al., 2017; Zhang et al., 2020; Schweighofer et al., 2024), tend to learn in highly correlated ways when faced with the same misspecification. By deviating in the same incorrect direction, they produce an adverse effect we term "collective blindness". This phenomenon is caused by the ensemble reinforcing, rather than correcting, the biases shared by all members. As illustrated in Figure 1 (C), in the orange circle, the traditional deep ensemble produces misclassified predictions with high confidence. The root of the problem is that traditional ensemble methods not only employ similar model architectures but, more critically, typically aggregate predictions using equal weights. When all models are plagued by the same misspecification, this simplistic averaging only serves to amplify their shared error. To address this, we propose a novel and effective solution: an optimally weighted deep ensemble built upon architectural diversity. By ensuring that exploitable differences exist among the models, we can theoretically derive data-driven weights that minimize the ensemble's prediction error on a held-out validation set. As illustrated in Figure 1 (F), the use of optimal weights helps the ensemble aggregate complementary strengths of its constituent models and produce more accurate predictions than relying on a single misspecified model. Our primary contributions are as follows:

(1) We are the first to systematically define and categorize misspecification in deep learning into variable, structural, and inherent forms. We propose weighted deep ensemble to mitigate the "collective blindness" effect seen in traditional ensembles and provide a theoretical guarantee for our weighted deep ensemble for well-specified and misspecified models.

(2) We establish an asymptotic error bound for the weighted deep ensemble estimator and show that the bound converges at the same rate as the smallest error bound among all candidate networks. This guarantees that the ensemble inherits the speed of the best individual model, so including slower or misspecified models can never slow it down, while any rapidly converging model immediately improves the overall rate. Furthermore, we provide detailed analyses for common networks such as MLP, CNN, and RNN.

(3) We prove that the weight vector yields a prediction risk that converges to the oracle minimum, even though the oracle weight itself depends on unknown population quantities and cannot be computed. Thus the proposed weighted choice method recovers the infeasible optimal weight asymptotically, giving the first rigorous guarantee that weighted deep ensemble can attain oracle-level accuracy

using only observable data. To the best of our knowledge, this is the first study to offer a theoretical guarantee for weighted deep ensemble.

## 2 RELATED WORK

**Misspecifiction.** Model misspecification, which occurs when a chosen model fails to accurately capture the true data-generating process, is a common challenge in statistics (McGuirk et al., 1993; Cerreia-Vioglio et al., 2025). Misspecification is categorized into several types: omitted varaible bias, where the exclusion of a relevant variables leads to biased and inconsistent parameter estimates (Gospodinov & Maasoumi, 2021), incorrect functional form, such as assuming a linear relationship when the true function is nonlinear (Gerds & Schumacher, 2001; Kasparis, 2011), and mismatch distribution when the assumed probability distribution for the error term or the response variable in models is incorrect (Masiha et al., 2021; Kuang et al., 2020). These types of misspecification always degrade model predictive performance (Lanzani, 2025). In deep learning, the universal approximation theorem states that a sufficiently large neural network can approximate any continuous function (Raghu et al., 2017; Kratsios et al., 2021). Therefore, previous research always assumed that deep models are well-specified. However, misspecification is a widespread issue in practice due to limited information and finite model capacity of neural networks with limited width and depth. Our work is the first to provide a clear definition for misspecification in deep learning and theoretical guarantees for deep ensembles under misspecified conditions.

**Ensemble Learning.** Deep ensembles, typically composed of identical architectures with different random initialization, have been shown to outperform single deep learning models in terms of accuracy (Lakshminarayanan et al., 2017; Mohammed & Kora, 2023). However, relying solely on the same model structure may limit the effectiveness of the ensemble. To address it, several works have introduced greater diversity by varying neural network architectures (Zhang et al., 2020) and training methods (Gontijo-Lopes et al., 2022). Recent studies have begun exploring weighted deep ensembles (Kim et al., 2018; Matena & Raffel, 2022), but these works provide only empirical evidence. Existing theoretical results for traditional ensemble learning (Wolpert, 1992; Van der Laan et al., 2007) primarily derive from classical settings, where convergence rates are often slower than $n^{-1/2}$ (Stone, 1982; Schmidt-Hieber, 2020) and typically apply to linear models rather than deep neural networks. Moreover, the existing diversity literature shows that uniformly averaged ensembles achieve a risk no worse than the average individual risk, but this line of work does not compare the ensemble to the best individual model and only focus on equal weighting (Zhang et al., 2020; Wood et al., 2023; Abe et al., 2022). PAC-Bayesian theory further provides generalization bounds for deep ensembles (Masegosa et al., 2020; Ortega et al., 2022). However, they fundamentally require the prior distribution that cannot be learned from the data and cannot identify optimal weights. In contrast, we extend validation-based weighting in traditional stacking methods to deep neural networks and establish the first asymptotic optimality result for weighted deep ensembles. We show that, using only observable validation data, a properly weighted deep ensemble can asymptotically achieve oracle-level predictive accuracy.

## 3 METHODOLOGY

### 3.1 PROBLEM SETUP

We consider a general model where the input features $\boldsymbol{X} = (X_1, \ldots, X_d) \in \mathbb{R}^d$ and the output $Y$ can be either real-valued or categorical. In regression tasks, $Y \in \mathbb{R}$ and follows the model $Y = f_0(\boldsymbol{X}) + \varepsilon$, where $f_0$ is an unknown true function and $\varepsilon$ is a noise term satisfying $\mathbb{E}(\varepsilon|\boldsymbol{X}) = 0$. In classification tasks with $C$ classes, $\boldsymbol{Y} = (Y_1, \ldots, Y_C)^\top \in \{0,1\}^C$ is the one-hot vector, where only one entry is 1 indicating the true class and all others are 0. The conditional probability of $\boldsymbol{Y}$ is modeled as $P(\boldsymbol{Y} \mid \boldsymbol{X}) = f_0(\boldsymbol{X})$ for $c = 1, \ldots, C$, where $f_0(\boldsymbol{X}) = (f_{0,1}(\boldsymbol{X}), \ldots, f_{0,C}(\boldsymbol{X}))^\top$ and $f_{0,c}(\boldsymbol{X}) = P(Y_c = 1 \mid \boldsymbol{X})$. We assume that $n$ independent observable samples $(\boldsymbol{X}_i, Y_i)$ are drawn from a joint distribution over $(\boldsymbol{X}, Y)$. The supremum norm is defined as $\|f\|_\infty = \sup_{\boldsymbol{X}} |f(\boldsymbol{X})|$, while the $L^2$ norm is $\|f\|_{L^2} = (\int |f(\boldsymbol{X})|^2 \, dP_{\boldsymbol{X}}(\boldsymbol{X}))^{1/2}$.

**Model training.** The observable data is split into two parts, a training set with size $n_{\text{train}}$ for training the neural network and the other $n_{\text{val}} = n - n_{\text{train}}$ for choosing weights. Specifically, a base model $\widehat{f}$

is obtained by empirical risk minimization: $\widehat{f} = \arg\min_{f \in \mathcal{F}} 1/n_{\text{train}} \sum_{i=1}^{n_{\text{train}}} \ell(f(\boldsymbol{X}_i), Y_i)$, where $\mathcal{F}$ denotes the model function class and $\ell$ is the loss function. And $f^*$ is defined as the minimizer of the true expected risk: $f^* = \arg\min_{f \in \mathcal{F}} \mathbb{E}\left[\ell(f(\boldsymbol{X}), Y)\right]$.

## 3.2 MISSPECIFICATION IN DEEP LEARNING

In this section, we systematically define three types of misspecification in deep learning. Let $f_0 : \mathcal{X}_{\text{true}} \to \mathcal{Y}$ denote the true data-generating function.

**Definition 1** (Variable Misspecification). *Let $\mathcal{X}_{\text{model}}$ be the feature space available to a given model. Define the projection map $\pi : \mathcal{X}_{\text{true}} \to \mathcal{X}_{\text{model}}$ that restricts each $x \in \mathcal{X}_{\text{true}}$ to its coordinates in $\mathcal{X}_{\text{model}}$. We say that the model exhibits **variable misspecification** if $f_0$ cannot be expressed as a composition of $\pi$ with a function on $\mathcal{X}_{\text{model}}$. Formally,*

$$\nexists g : \mathcal{X}_{\text{model}} \to \mathcal{Y} \quad \text{such that} \quad f_0(x) = g(\pi(\boldsymbol{X})) \quad \text{for almost every } \boldsymbol{X} \in \mathcal{X}_{\text{true}}.$$

**Example.** Consider an image classification task where the true label depends on latent features $Z = (Z_1, Z_2)$ (e.g., ear shape and nose shape). The true function is $f_0(Z) = \mathbb{I}_{Z_1 + Z_2 > 0}$ (e.g., predicting cat if positive, dog if negative). If the model's feature space $\mathcal{X}_{\text{model}}$ corresponds only to $Z_2$ due to missing feature input or incomplete feature extraction, the model suffers from variable misspecification.

**Definition 2** (Structural Misspecification). *Let $\mathcal{H}$ be the function class corresponding to a given neural network architecture. Let $\mathcal{L}$ be a loss function and define the risk*

$$R(h) = \mathbb{E}_{\boldsymbol{X} \sim P_{\boldsymbol{X}}}\left[\mathcal{L}\big(h(\boldsymbol{X}), f_0(\boldsymbol{X})\big)\right].$$

*For a tolerance level $\delta > 0$, we say that the architecture exhibits **structural misspecification** if the minimal achievable risk within $\mathcal{H}$ exceeds this tolerance. Formally,*

$$\inf_{h \in \mathcal{H}} R(h) > \delta.$$

**Example.** Consider ReLU neural networks with both depth and width restricted to the order of $\log(n)$. For $\beta$-smooth functions in dimension $d$, it is known that the approximation error in this class satisfies

$$\inf_{h \in \mathcal{H}} R(h) = c \log(n)^{-4\beta/d},$$

for some constant $c > 0$ depending only on $\beta$ and $d$ (Jiao et al., 2023). We set the tolerance level to $\delta = n^{-1/4}$, which corresponds to the rate condition commonly required in double machine learning (Chernozhukov et al., 2018). For sufficiently large $n$, we have

$$\inf_{h \in \mathcal{H}} R(h) = c \log(n)^{-4\beta/d} > \delta,$$

so the architecture is structurally misspecified relative to the tolerance $\delta$.

**Definition 3** (Inherent Misspecification). *Let $\mathcal{H}$ be the function class corresponding to a given neural network architecture. Approximation results for neural networks typically require $f_0$ to belong to a smoothness class, such as a Hölder or Sobolev space, in order to guarantee vanishing $L^2$ approximation error. We say that the architecture exhibits **inherent misspecification** if $f_0$ does not satisfy the required smoothness conditions, so that the minimal achievable $L^2$ error is bounded away from zero. Formally, for some tolerance level $\delta > 0$,*

$$\inf_{h \in \mathcal{H}} \|h - f_0\|_{L^2} > \delta.$$

**Example.** Consider the Dirichlet function

$$f_0(x) = \begin{cases} 1, & x \in \mathbb{Q} \cap [0,1], \\ 0, & x \in (\mathbb{R} \setminus \mathbb{Q}) \cap [0,1]. \end{cases}$$

This function is nowhere continuous and does not belong to any Hölder or Sobolev class. As a result, neural networks cannot approximate $f_0$ in $L^2$ with vanishing error, and the approximation gap remains strictly positive. Therefore, the model class $\mathcal{H}$ suffers from inherent misspecification.

### 3.3 WEIGHTED DEEP ENSEMBLE

First, we introduce the standard deep ensemble method.

**Deep ensembles.** A standard *deep ensemble* consists of $M$ candidate models $\widehat{f}_1(\cdot), \ldots, \widehat{f}_M(\cdot)$. The *deep ensemble prediction* $\bar{f}(\boldsymbol{X})$ is: $\bar{f}(\boldsymbol{X}) = \sum_{m=1}^{M} w_m \widehat{f}_m(\boldsymbol{X})$, where all weights are set equally as $w_m = 1/M$. Typically, the candidate models share the same neural network architecture and are trained independently on the same dataset using the same loss function $\ell(f_m(\boldsymbol{X}), Y)$, which corresponds to minimizing the empirical risk: $\mathcal{L}_{\text{avg}} = \mathbb{E}\left[\ell(\bar{f}(\boldsymbol{X}), Y)\right]$.

As stated before, traditional deep ensembles may suffer from "collective blindness" in the presence of variable, structural, or inherent misspecification. This motivates us to apply a more flexible weighting scheme.

**Weighted deep ensemble.** In this paper, we propose a method called *Weighted Deep Ensemble (WDE)*, which combines multiple neural networks with different structures into a single predictive model. Let $\boldsymbol{w} = (w_1, \ldots, w_M)^{\mathsf{T}}$ be the vector of ensemble weights. We restrict the weights to the simplex $\mathcal{W} = \{\boldsymbol{w} \in [0,1]^M : \sum_{m=1}^{M} w_m = 1\}$. The *weighted deep ensemble prediction* is defined as $\widehat{f}(\boldsymbol{X}; \boldsymbol{w}) = \sum_{m=1}^{M} w_m \widehat{f}_m(\boldsymbol{X})$. In practice, the weight vector $\boldsymbol{w}$ is unknown and need to be estimated from observable data.

**Weight choice criterion.** We adopt a *validation-risk minimization* (VRM) criterion: the estimator $\widehat{\boldsymbol{w}}$ is chosen to minimize the empirical loss on a validation set, under the simplex constraints

$$\widehat{\boldsymbol{w}} = \arg\min_{\boldsymbol{w} \in \mathcal{W}} \frac{1}{n_{\text{val}}} \sum_{i=1}^{n_{\text{val}}} \ell(\widehat{f}(\boldsymbol{X}_i; \boldsymbol{w}), Y_i).$$

This VRM strategy links the weighting scheme directly to out-of-sample performance, introducing no extra hyperparameters beyond the standard validation split. Specifically, we consider two general tasks: (i) Regression: with the squared loss $\ell_{\text{reg}}(f(\boldsymbol{X}; \boldsymbol{w}), Y) = (f(\boldsymbol{X}; \boldsymbol{w}) - Y)^2$, the VRM objective reduces to a strictly convex quadratic program on the simplex $\mathcal{W}$ (Li et al., 2023; Qu et al., 2025). (ii) Classification: Using cross-entropy loss $\ell_{\text{class}}(f(\boldsymbol{X}; \boldsymbol{w}), \boldsymbol{Y}) = -\boldsymbol{Y}^{\top} \log(f(\boldsymbol{X}; \boldsymbol{w}))$, where $\boldsymbol{Y} \in \{0,1\}^C$ denotes the one-hot encoding of the true class, and $f(\boldsymbol{X}; \boldsymbol{w})$ denotes the predicted probability. Since the mapping $\boldsymbol{w} \mapsto f(\boldsymbol{X}; \boldsymbol{w})$ is affine and $-\log(\cdot)$ is convex, the loss function remains convex in $\boldsymbol{w}$ and can be optimized using projected-gradient (Boyd & Vandenberghe, 2004) methods. So we construct the *weighted deep ensemble estimator* $\widehat{f}(\boldsymbol{X}; \widehat{\boldsymbol{w}}) = \sum_{m=1}^{M} \widehat{w}_m \widehat{f}_m(\boldsymbol{X})$. Due to the convexity of the loss function with respect to $\boldsymbol{w}$, we can obtain the global optimal solution for the ensemble weights.

### 3.4 ASYMPTOTIC ERROR BOUNDS

Our goal is to establish the asymptotic error bound of the proposed weighted deep ensemble estimator. Specifically, we aim to show that our proposed weighted deep ensemble effectively combines multiple small neural networks to achieve an asymptotic error bound at least as fast as that of the best candidate. By appropriately combining these models, the estimator adapts to various data structures and retains the ability to capture intricate features without resorting to a large, monolithic network. Consequently, the weighted deep ensemble benefits from more flexible modeling choices while still maintaining an asymptotic error bound that is as fast as the best candidate. Importantly, the presence of misspecified or poorly performing candidate models does not slow down the convergence rate of the estimator. We provide theoretical guarantees for both regression (Theorem 1) and classification (Theorem 2) settings. To establish these results, we begin by introducing the following condition:

**Condition 1.** *(i). There exists a positive constant $C$ such that $\|f_0(\boldsymbol{X})\|_\infty < C$, $\|\widehat{f}_m(\boldsymbol{X})\|_\infty < C$ for $m = 1, \ldots, M$; (ii). $\mathbb{E}(\varepsilon|\boldsymbol{X}) = 0$, and $\varepsilon$ is sub-Gaussian with parameter $\sigma$.*

This condition restricts the upper bound of $f_0$, $f^*$, and that $\varepsilon$ has mean zero and sub-Gaussian tails. This condition is also widely used in the literature; see Schmidt-Hieber (2020) and Jiao et al. (2023) for example.

**Theorem 1.** *Suppose Condition 1 holds and assume that the candidate model with the fastest convergence rate has an asymptotic error bound of order $S$, i.e., the candidate model with the fastest*

*asymptotic error bound (denoted as $\tilde{f}$ without loss of generality) satisfies $\|f_0(\boldsymbol{X}) - \widetilde{f}(\boldsymbol{X})\|_{L^2} = O_p(S)$, then our weighted deep ensemble estimator can also achieve this rate asymptotically:*

$$\|f_0(\boldsymbol{X}) - \widehat{f}(\boldsymbol{X}; \widehat{\boldsymbol{w}})\|_{L^2} = O_p(S + \sqrt{\frac{\log(M)}{n}}).$$

Besides the asymptotic error bound $S$, the result also includes an additional term of order $\sqrt{\log(M)/n}$. In practice, the minimax rate for nonparametric methods such as neural networks is typically of order $n^{-\beta/(2\beta+d)}$ for some smoothness $\beta$ and input dimension $d$ (Stone, 1982), which is slower than $\sqrt{\log(M)/n}$ when the number of candidate model is fixed; hence the overall asymptotic error bound is dominated by the nonparametric estimation error. Therefore, Theorem 1 establishes that the asymptotic error bound of the weighted deep ensemble estimator is no worse than that of the best individual candidate model. In other words, regardless of which candidate model achieves the smallest asymptotic error bound, the ensemble procedure guarantees at least comparable asymptotic performance and will not converge more slowly than that benchmark.

Theorem 1 presents the asymptotic error bound of the deep ensemble estimator under regression tasks. In fact, we can establish similar properties for classification tasks. Before presenting the theorem, we define $R_{0/1}(f) = \mathbb{E}\{I(\arg\max_c Y_c \neq \arg\max_c f_c(\boldsymbol{X}))\} - \mathbb{E}\{I(\arg\max_c Y_c \neq \arg\max_c f_{0,c}(\boldsymbol{X}))\}$ to be the excess misclassification rate.

**Theorem 2.** *Suppose $f_0(\boldsymbol{X})$ is uniformly bounded away from 0 and 1 and assume that the best candidate model has a misclassification rate of $S$, i.e., the candidate model with the smallest misclassification rate (denoted as $\tilde{f}$ without loss of generality) satisfies $R_{0/1}(\tilde{f}) = O_p(S)$, then our weighted deep ensemble estimator can also achieve this misclassification rate asymptotically:*

$$R_{0/1}(\widehat{f}(\boldsymbol{X}; \widehat{\boldsymbol{w}})) = O_p(S + \sqrt{\frac{\log(M)}{n}}).$$

Theorems 1 and 2 show that, in both regression and classification tasks, the asymptotic error bound of our weighted deep ensemble estimator matches the asymptotic error bound of the best single candidate model. Detailed proofs are provided in the Appendix.

In addition, when the pool of candidate models is altered, the estimator's attainable asymptotic error bound necessarily shifts in response to the new composition. For better understanding, the next three corollaries provide the asymptotic error bound of the weighted deep ensemble estimator when the candidate models are chosen from MLP-based networks, CNN-based networks, and RNN-based networks. It is worth noting that Theorem 1 holds under the listed moment conditions, but additional assumptions are implicitly embedded in the asymptotic error bound $O_p(S)$. Since different neural network architectures require different conditions to guarantee convergence, we do not enumerate them explicitly here and instead present the result in terms of the general order $O_p(S)$. Accordingly, the subsequent corollaries impose further conditions on the candidate models, ensuring that they can indeed attain the corresponding asymptotic error bound in those specific settings. Define $\widetilde{O}_p(\cdot)$ as the rate by ignoring logarithmic factors.

**Corollary 1** (MLP case)**.** *If all $M = O(1)$ candidate models are MLP-based models, $f_0$ is $\beta$-Hölder smooth with $\beta > 1$, and Conditions of Theorem 4.2 in Jiao et al. (2023) holds, then with some specifically designed candidate models, the weighted deep ensemble estimators can achieve asymptotic error bound of*

$$\|f_0(\boldsymbol{X}) - \widehat{f}(\boldsymbol{X}; \widehat{\boldsymbol{w}})\|_{L^2}^2 = \widetilde{O}_p(n^{-2\beta/(d+2\beta)}).$$

**Corollary 2** (CNN case)**.** *If all $M = O(1)$ candidate models are CNN-based models, $f_0$ is $\beta$-Hölder smooth with $\beta > 1$, and Conditions of Theorem 4.6 in Shen et al. (2022) holds, then with some specifically designed candidate models, the weighted deep ensemble estimators can achieve asymptotic error bound of*

$$\|f_0(\boldsymbol{X}) - \widehat{f}(\boldsymbol{X}; \widehat{\boldsymbol{w}})\|_{L^2}^2 = \widetilde{O}_p(n^{-2\beta/(d+2\beta)}).$$

**Corollary 3** (RNN case)**.** *If all $M = O(1)$ candidate models are RNN-based models, $f_0$ is $\beta$-Hölder smooth with $\beta > 1$ and Conditions of Theorem 6 in Jiao et al. (2024) holds, then with some*

*specifically designed candidate models, the weighted deep ensemble estimator can achieve asymptotic error bound of*

$$\|f_0(\boldsymbol{X}) - \widehat{f}(\boldsymbol{X}; \widehat{\boldsymbol{w}})\|_{L^2}^2 = \widetilde{O}_p(n^{-2\beta/(dl+2\beta)}),$$

*where l is the length of the input sequence, i.e., the number of time steps processed by the RNN.*

These results can also be extended to ResNet, Transformer, and other network structures. As long as we have the asymptotic error bound of a single network, Theorems 1 and 2 ensure that the weighted deep ensemble estimator can achieve the same asymptotic error bound as the fastest candidate model.

### 3.5 ASYMPTOTIC OPTIMALITY UNDER MODEL MISSPECIFICATION

Unlike traditional ensembles with uniform weights, our method learns data-dependent weights, ensuring that the ensemble performs at least as well as the best candidate asymptotically. Since equal weights lie within our admissible weight space, our approach is guaranteed to match or exceed the performance of equal-weighted averaging asymptotically. In practice, the oracle weight vector is unknown and cannot be directly computed from observable data. By instead using the VRM criterion, our method significantly reduces computational cost while maintaining competitive efficiency, making it particularly suitable for large-scale scenarios. In the following, we present theoretical results showing that the proposed weighted deep ensemble estimator achieves asymptotic optimality and attains oracle-level accuracy using only observable data.

Let $R(\boldsymbol{w}) = \|f_0(\boldsymbol{X}) - \widehat{f}(\boldsymbol{X}; \boldsymbol{w})\|_{L^2}^2$, $R^*(\boldsymbol{w}) = \|f_0(\boldsymbol{X}) - f^*(\boldsymbol{X}; \boldsymbol{w})\|_{L^2}^2$, $\xi_n = \inf_{\boldsymbol{w} \in \mathcal{W}} R^*(\boldsymbol{w})$, and $\phi_n = \sup_{\boldsymbol{w} \in \mathcal{W}} \|\widehat{f}(\boldsymbol{X}; \boldsymbol{w}) - f^*(\boldsymbol{X}; \boldsymbol{w})\|_{L^2}$. To establish the asymptotic optimality of $\widehat{\boldsymbol{w}}$, we require the following condition.

**Condition 2.** *(i).* $\xi_n^{-1} n^{-1/2} M^{1/2} = o(1)$; *(ii).* $\xi_n^{-1} \phi_n = o_p(1)$.

This condition regulates the divergence speed of $\xi_n$, and it is frequently used in FMA research, such as Ando & Li (2014); Zhang et al. (2016). Condition 2 requires $\xi_n$ to grow faster than $\sqrt{1/n}$ and $\phi_n$. Importantly, this condition should be interpreted as a *misspecification condition*, ensuring that the oracle risk $\xi_n$ does not vanish too quickly relative to the estimation error. It naturally aligns with the three forms of misspecification introduced above:

(1) **Variable misspecification.** When crucial variables are missing, or when the essential features of $f_0$ cannot be generated from the available input space, $f^*$ cannot converge to $f_0$. In this case, the approximation error remains bounded away from zero, which implies that the oracle risk $\xi_n$ does not vanish and Condition 2 (i) is satisfied. This requirement should be viewed as a stronger form of variable misspecification, as it excludes cases where the omitted variables have only negligible influence on $f_0$, for instance when their contribution diminishes asymptotically.

(2) **Structural misspecification.** When networks have limited depth or width, their approximation error remains bounded away from zero. In this case $\xi_n$ decreases at a much slower rate than $n^{-1/2}$, so Condition 2 (i) is satisfied.

(3) **Inherent misspecification.** When the true function lies outside the smoothness classes typically required for neural network approximation, the minimal $L^2$ error remains strictly positive. Consequently, $\xi_n$ is bounded away from zero, and Condition 2 (i) holds.

Unlike $\xi_n$, the term $\phi_n$ measures the estimation error between $\widehat{f}$ and $f^*$, which depends on sample size rather than model specification. Condition 2 (ii) therefore requires that $\phi_n$ converges to zero at a faster rate than $\xi_n$, ensuring that estimation error does not dominate the asymptotic behavior.

**Theorem 3.** *Suppose that Conditions 1 and 2 hold, then*

$$\frac{R(\widehat{\boldsymbol{w}})}{\inf_{\boldsymbol{w} \in \mathcal{W}} R(\boldsymbol{w})} \to 1$$

*in probability as $n \to \infty$.*

This theorem states that under Conditions 1 and 2, as the sample size $n$ goes to infinity, the ratio of the risk of $\widehat{\boldsymbol{w}}$ to the infimum of the risk over all possible weights converges to 1. In other words, although the weight that minimizes the risk is infeasible, the proposed weight choice criterion identifies a

weight $\widehat{w}$ whose risk becomes asymptotically equal to the minimal risk. This means our procedure performs asymptotically as well as this ideal benchmark. This provides strong theoretical support for our weight selection method, assuring that no asymptotic loss is incurred compared to the unattainable optimum. All theoretical proofs are provided in the Appendix B.

## 4 NUMERICAL RESULTS

In this section, we investigate the models that are well-specified or suffer from variable misspecification, structural misspecification, and inherent misspecification.

**Baselines.** We compare several ensemble strategies: (1) Deep Ensemble (DE): homogeneous MLPs with different initializations and equal weights; (2) Equal-Weight Heterogeneous (EW): heterogeneous MLPs with equal weights; (3) Our Method, WDE: heterogeneous MLPs with optimal weights.

**Experimental details.** Datasets are split 6:2:2 for training, validation, and testing. We use Adam with early stopping (patience=20 epochs). Hyperparameters: learning rate searched in [0.001, 0.1], batch size 128, max 5000 epochs. Our method uses 4 heterogeneous networks with total parameters matching a 4×MLP (2 hidden layers of 30 nodes) DE for fair comparison.

Table 1: Performance comparison across different complexity types with varying missing variables, with sample size 5000.

| Task | # Missing | DE | EW | WDE | $\Delta_{\text{DE}}$ | $\Delta_{\text{EW}}$ |
|---|---|---|---|---|---|---|
| Nested | 0 | $7.452 \pm 0.319$ | $7.711 \pm 0.527$ | $\mathbf{7.025 \pm 0.247}$ | 6.07% | 9.76% |
| | 1 | $8.175 \pm 0.472$ | $8.686 \pm 0.957$ | $\mathbf{8.028 \pm 1.236}$ | 1.84% | 8.19% |
| | 3 | $8.390 \pm 0.681$ | $8.825 \pm 0.814$ | $\mathbf{8.077 \pm 1.211}$ | 3.88% | 9.27% |
| | 5 | $8.440 \pm 0.788$ | $8.493 \pm 0.482$ | $\mathbf{7.792 \pm 0.287}$ | 8.31% | 9.00% |
| | 7 | $9.122 \pm 0.498$ | $8.687 \pm 0.510$ | $\mathbf{7.967 \pm 0.431}$ | 14.50% | 9.04% |
| Interaction | 0 | $1.975 \pm 0.330$ | $1.854 \pm 0.187$ | $\mathbf{1.773 \pm 0.131}$ | 11.35% | 4.57% |
| | 1 | $3.251 \pm 0.740$ | $3.640 \pm 0.782$ | $\mathbf{2.949 \pm 0.778}$ | 10.22% | 23.43% |
| | 3 | $3.949 \pm 0.991$ | $4.434 \pm 1.046$ | $\mathbf{3.890 \pm 0.984}$ | 20.76% | 13.98% |
| | 5 | $5.239 \pm 1.149$ | $5.223 \pm 1.176$ | $\mathbf{5.077 \pm 1.099}$ | 22.36% | 2.87% |
| | 7 | $5.747 \pm 1.206$ | $5.617 \pm 1.266$ | $\mathbf{5.548 \pm 1.168}$ | 13.27% | 1.24% |
| Periodic | 0 | $2.154 \pm 0.101$ | $2.197 \pm 0.121$ | $\mathbf{2.023 \pm 0.116}$ | 6.47% | 8.59% |
| | 1 | $2.578 \pm 0.175$ | $2.844 \pm 0.419$ | $\mathbf{2.474 \pm 0.193}$ | 4.17% | 14.93% |
| | 3 | $2.994 \pm 0.213$ | $3.148 \pm 0.439$ | $\mathbf{2.901 \pm 0.251}$ | 3.19% | 8.48% |
| | 5 | $3.631 \pm 0.254$ | $3.666 \pm 0.410$ | $\mathbf{3.362 \pm 0.273}$ | 7.98% | 9.04% |
| | 7 | $3.841 \pm 0.316$ | $3.932 \pm 0.481$ | $\mathbf{3.714 \pm 0.331}$ | 12.52% | 5.87% |

**Well-specified Models and Variable Misspecification.** We consider some simple data-generating processes (DGPs) that can be well-approximated by simple MLPs, i.e., the approximation error is small. Let $\boldsymbol{X} \in \mathbb{R}^p$ be a random vector where each feature $X_j$ is independently sampled from $\mathcal{N}(0, 1)$, with the number of features $p = 10$. We define the different DGPs: (i) Nested: $f_0(\boldsymbol{x}) = \sin(\sum_{j=1}^{10} x_j^2) + \sum_{j=1}^{10} (\cos x_j)^2$; (ii) Interaction: $f_0(\boldsymbol{X}) = \frac{1}{2} \sum_{j=1}^{5} \sum_{k=6}^{10} X_j X_k$; (iii) Periodic: $f_0(\boldsymbol{X}) = \sum_{j=1}^{5} \sin(X_j) + \sum_{j=6}^{10} \cos(X_j)$.

When all relevant features are included in the model, the setting is considered well-specified. We introduce variable misspecification by randomly dropping a subset of features during training, with the number of dropped features controlling the degree of misspecification. The results are summarized in Table 2. Our method WDE consistently achieves the lowest MSE.

**Structural Misspecification.** To investigate structural misspecification, we define the DGP as a convex combination of a simple function and a more complex function: $f_0(\boldsymbol{X}) = \alpha f_{\text{simple}}(\boldsymbol{X}) + (1 - \alpha) f_{\text{complex}}(\boldsymbol{X})$, where $f_{\text{simple}} = \text{MLP}_{2 \times 30}$ and $f_{\text{complex}} = \text{MLP}_{3 \times 100}$. The function used for fitting is $f_{\text{simple}}(\boldsymbol{X})$. The parameter $\alpha \in [0, 1]$ controls the degree of misspecification, from well-specified ($\alpha = 0$) to totally misspecified ($\alpha = 1$). We report the relative MSE under varying degrees of misspecification with a sample size of $N = 5000$ in Table 2. The results show that as the misspecification degree increases, the MSE of DE grows rapidly. In contrast, our method (WDE) significantly mitigates this performance degradation and maintains robust accuracy even under high misspecification.

Table 2: Performance comparison across different parameter discrepancies with sample size 5000

| $\Delta$ | $\alpha$ | DE | EW | WDE | $\Delta_{DE}$ | $\Delta_{EW}$ |
|---|---|---|---|---|---|---|
| 30000 | 0.2 | $0.598 \pm 0.222$ | $0.402 \pm 0.085$ | $\mathbf{0.341 \pm 0.034}$ | 75.47% | 17.83% |
| | 0.5 | $0.630 \pm 0.259$ | $0.438 \pm 0.179$ | $\mathbf{0.397 \pm 0.146}$ | 58.59% | 10.37% |
| | 0.7 | $0.634 \pm 0.230$ | $0.473 \pm 0.128$ | $\mathbf{0.398 \pm 0.083}$ | 59.53% | 18.87% |
| | 0.9 | $0.654 \pm 0.157$ | $0.524 \pm 0.185$ | $\mathbf{0.417 \pm 0.089}$ | 56.83% | 25.76% |
| 50000 | 0.2 | $0.587 \pm 0.159$ | $0.374 \pm 0.075$ | $\mathbf{0.339 \pm 0.032}$ | 73.10% | 10.22% |
| | 0.5 | $0.632 \pm 0.193$ | $0.425 \pm 0.098$ | $\mathbf{0.399 \pm 0.091}$ | 58.31% | 6.49% |
| | 0.7 | $0.701 \pm 0.201$ | $0.457 \pm 0.099$ | $\mathbf{0.419 \pm 0.067}$ | 67.21% | 9.17% |
| | 0.9 | $0.790 \pm 0.171$ | $0.488 \pm 0.074$ | $\mathbf{0.437 \pm 0.061}$ | 80.65% | 11.64% |
| 100000 | 0.2 | $0.576 \pm 0.206$ | $0.335 \pm 0.031$ | $\mathbf{0.331 \pm 0.026}$ | 74.31% | 1.51% |
| | 0.5 | $0.582 \pm 0.175$ | $0.350 \pm 0.050$ | $\mathbf{0.340 \pm 0.039}$ | 71.42% | 3.15% |
| | 0.7 | $0.638 \pm 0.139$ | $0.353 \pm 0.052$ | $\mathbf{0.346 \pm 0.043}$ | 84.26% | 1.84% |
| | 0.9 | $0.677 \pm 0.226$ | $0.350 \pm 0.044$ | $\mathbf{0.348 \pm 0.043}$ | 94.50% | 0.50% |

Table 3: Comparison with relative improvement percentages ($\Delta$) across DE, EW and WDE.

| Complexity | $N$ | DE | EW | WDE | $\Delta_{DE}$ | $\Delta_{EW}$ |
|---|---|---|---|---|---|---|
| Square Wave | 5000 | $0.96068 \pm 0.02707$ | $0.94702 \pm 0.01385$ | $\mathbf{0.93171 \pm 0.02422}$ | 3.11% | 1.64% |
| | 10000 | $0.81132 \pm 0.02665$ | $0.80467 \pm 0.02623$ | $\mathbf{0.76520 \pm 0.03430}$ | 6.03% | 5.16% |
| | 15000 | $0.74766 \pm 0.03754$ | $0.71625 \pm 0.03140$ | $\mathbf{0.64963 \pm 0.06092}$ | 15.09% | 10.25% |
| Infinite | 5000 | $0.00137 \pm 0.00021$ | $0.00141 \pm 0.00010$ | $\mathbf{0.00117 \pm 0.00012}$ | 17.34% | 20.40% |
| | 10000 | $0.00085 \pm 0.00015$ | $0.00086 \pm 0.00013$ | $\mathbf{0.00067 \pm 0.00014}$ | 27.38% | 28.86% |
| | 15000 | $0.00057 \pm 0.00013$ | $0.00065 \pm 0.00010$ | $\mathbf{0.00044 \pm 0.00012}$ | 30.13% | 49.55% |

**Inherent Misspecification.** We consider two discontinuous DGPs to simulate inherent misspecification: (i) Infinite Discontinuity: $f_0(\boldsymbol{X}) = \sum_{j=1}^{d} \frac{1}{j} \left( \sum_{k=1}^{K} \frac{1}{k^2} \cdot \mathbb{I} \left( X_j > \frac{1}{k} \right) \right)$ is the indicator function and $K = 1000$; (ii) Square Wave: $y = \sum_{i=1}^{d} \frac{1}{i+1} \cdot \text{sgn} \left( \sin \left( \frac{2\pi(i+1)}{d} x_i + \frac{i\pi}{d} \right) \right)$, where $\text{sgn}(\cdot)$ is the signum function. The results in Table 3 demonstrate that WDE achieves substantial improvement.

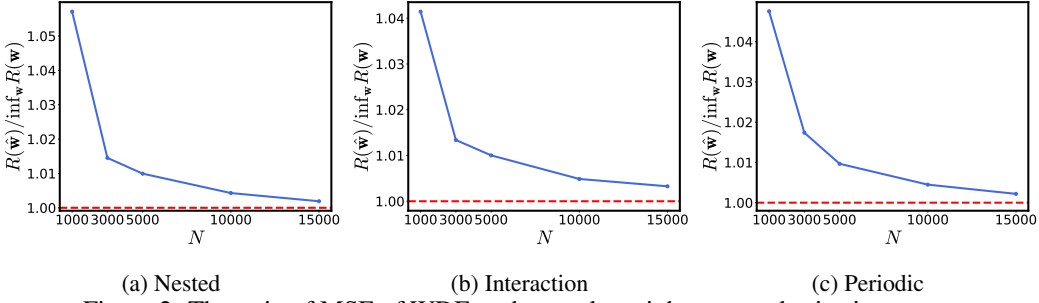

(a) Nested        (b) Interaction        (c) Periodic

Figure 2: The ratio of MSE of WDE to the oracle weight as sample size increases.

**Theoretical results.** To further validate Theorem 3, we plot the ratio of the MSE of our proposed weighted deep ensemble estimator to that of the oracle weight as the sample size increases, as shown in Figure 2. We can observe that as the sample size grows, the weighted deep ensemble estimator asymptotically approaches the optimal oracle weight using only the observed data.

**Comparing with other weighted methods.** We compare our method against two alternatives in Table 4: (i) Greedy Ensemble: Sequentially adds models to minimize validation loss, then uses equal weighting. (ii) In-sample Ensemble: Weights are optimized directly on the training MSE.

More simulation results can be found in Section D in the Appendix.

## 5 CONCLUSION

Our paper formally defines the misspecification problem in deep learning and establishes a theoretical foundation for the weighted deep ensemble, including its error bound and asymptotic optimality. Extensive experiments demonstrate that our method consistently outperforms traditional deep ensembles across various misspecification scenarios and significantly mitigates the adverse effects of model misspecification.

Table 4: Performance comparison of weighted methods in variable misspecification

| Complexity | # Missing | WDE | In-sample Ensemble | Greedy Ensemble |
|---|---|---|---|---|
| Nested | 0 | **7.025 ± 0.247** | 7.258 ± 0.329 | 7.136 ± 0.232 |
| | 1 | **8.028 ± 1.236** | 8.149 ± 1.150 | 8.108 ± 1.187 |
| | 3 | **8.077 ± 1.211** | 8.186 ± 1.208 | 8.177 ± 1.217 |
| | 5 | **7.792 ± 0.287** | 7.963 ± 0.338 | 7.904 ± 0.346 |
| | 7 | **7.967 ± 0.431** | 8.076 ± 0.393 | 8.045 ± 0.400 |
| Interaction | 0 | **1.773 ± 0.131** | 1.869 ± 0.161 | 1.794 ± 0.116 |
| | 1 | **2.949 ± 0.778** | 2.990 ± 0.813 | 2.948 ± 0.778 |
| | 3 | **3.890 ± 0.984** | 3.914 ± 0.995 | 3.908 ± 0.981 |
| | 5 | **5.077 ± 1.099** | 5.114 ± 1.038 | 5.125 ± 1.084 |
| | 7 | **5.548 ± 1.168** | 5.651 ± 1.124 | 5.547 ± 1.170 |
| Periodic | 0 | **2.023 ± 0.116** | 2.086 ± 0.137 | 2.051 ± 0.126 |
| | 1 | **2.474 ± 0.193** | 2.535 ± 0.178 | 2.518 ± 0.226 |
| | 3 | **2.901 ± 0.251** | 2.961 ± 0.261 | 2.929 ± 0.260 |
| | 5 | **3.362 ± 0.273** | 3.400 ± 0.270 | 3.378 ± 0.279 |
| | 7 | **3.714 ± 0.331** | 3.751 ± 0.310 | 3.730 ± 0.336 |

## 6 ETHICS STATEMENT

Our work is committed to the highest standards of scientific excellence, grounded in the principles of honesty, reliability, and transparency. The core technical contribution of this paper is to address the challenge of model misspecification in deep learning. This is not merely a technical problem but an ethical imperative. A misspecified model can produce unreliable predictions, perpetuate and amplify societal biases, and ultimately cause harm if deployed in critical real-world applications such as healthcare, finance, or autonomous systems. By developing methods to better understand, identify, and correct for misspecification, our research aims to contribute to the creation of more robust, fair, and trustworthy AI systems. We believe that this work is a necessary step toward the responsible development of artificial intelligence, ensuring that its benefits can be realized while minimizing potential negative societal consequences.

## 7 REPRODUCIBILITY STATEMENT

We are committed to the full reproducibility of our work. To this end, we have included the complete and detailed derivations of our theoretical proofs in the Appendix. Furthermore, all experimental results presented in this paper can be fully reproduced according to experimental details and and the source code will be made available in our code repository. [1]

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

## A    Large Language Model Usage Disclosure

In this work, we made limited use of a large language model (LLM) as an auxiliary tool. In particular:

**Language polishing**: We used ChatGPT-5 to improve the readability, grammar, and fluency of the English text. The authors reviewed all edits and manually adjusted phrasing as needed.
**Code assistance**: We asked ChatGPT-5 to assist in generating boilerplate code for data preprocessing, but in a minimal and constrained way; the authors carefully verified, tested, modified, and adapted all generated code to ensure correctness.

We emphasize that all content in the submission is attributed to the authors. We take full responsibility for the correctness of all claims and any content originally generated by the LLM that contained errors or inconsistencies that were revised or removed. We confirm that the LLM was not included as an author, and no portion of the submission is entirely generated without human oversight.

## B    Proof of theorems

To facilitate the proof of the theorem, we begin by stating a useful lemma.

### B.1    Lemma 1

**Lemma 1** (Lemma 1 in Zhang (2010))**.** *Let*

$$\widehat{\boldsymbol{w}} = \arg \min_{\boldsymbol{w} \in \mathcal{W}} \{R(\boldsymbol{w}) + a_n(\boldsymbol{w}) + b_n\}.$$

*If*

$$\sup_{\boldsymbol{w} \in \mathcal{W}} \frac{|a_n(w)|}{R^*(w)} = o_p(1)$$

*and*

$$\sup_{w \in W} \frac{|R(\boldsymbol{w}) - R^*(\boldsymbol{w})|}{R^*(\boldsymbol{w})} = o_p(1),$$

*and there exists a positive constant c so that* $\lim_{n \to \infty} \inf_{\boldsymbol{w} \in \mathcal{W}} R^*(\boldsymbol{w}) \geq c$ *almost surely, then we have*

$$\frac{R(\widehat{\boldsymbol{w}})}{\inf_{w \in \mathcal{W}} R(\boldsymbol{w})} \to 1$$

*in probability.*

### B.2 PROOF OF THEOREM 1

The weight choice criterion can be decomposed as

$$\mathcal{L}(\boldsymbol{w})$$
$$= \frac{1}{n_{\text{val}}} \sum_{i=1}^{n_{\text{val}}} \{Y_i - \widehat{f}(\boldsymbol{X}_i; \boldsymbol{w})\}^2$$
$$= \frac{1}{n_{\text{val}}} \sum_{i=1}^{n_{\text{val}}} \{Y_i - f_0(\boldsymbol{X}_i) + f_0(\boldsymbol{X}_i) - \widehat{f}(\boldsymbol{X}_i; \boldsymbol{w})\}^2$$
$$= \frac{1}{n_{\text{val}}} \sum_{i=1}^{n_{\text{val}}} \{Y_i - f_0(\boldsymbol{X}_i)\}^2 + \frac{1}{n_{\text{val}}} \sum_{i=1}^{n_{\text{val}}} \{f_0(\boldsymbol{X}_i) - \widehat{f}(\boldsymbol{X}_i; \boldsymbol{w})\}^2$$
$$+ \frac{2}{n_{\text{val}}} \sum_{i=1}^{n_{\text{val}}} \{Y_i - f_0(\boldsymbol{X}_i)\}\{f_0(\boldsymbol{X}_i) - \widehat{f}(\boldsymbol{X}_i; \boldsymbol{w})\}.$$

We first analyze $1/n_{\text{val}} \sum_{i=1}^{n_{\text{val}}} \{Y_i - f_0(\boldsymbol{X}_i)\}\{f_0(\boldsymbol{X}_i) - \widehat{f}(\boldsymbol{X}_i; \boldsymbol{w})\}$. Let $\mathcal{G}_r = \{g_{\boldsymbol{w}}(\boldsymbol{X}) = f_0(\boldsymbol{X}) - \widehat{f}(\boldsymbol{X}; \boldsymbol{w}) : \boldsymbol{w} \in \mathcal{W}, \|f_0(\boldsymbol{X}) - \widehat{f}(\boldsymbol{X}; \boldsymbol{w})\|_{L^2} \leq r\}$, and let $\mathcal{D} = \{\mathcal{D}_{\text{train}}, \mathcal{D}_{\text{validation}}\}$ collect all observed samples. By the multiplier inequality (Van Der Vaart & Wellner, 1996; Bartlett et al., 2005), we have

$$\sup_{g \in \mathcal{G}_r} \Big[\frac{1}{n_{\text{val}}} \sum_{i=1}^{n_{\text{val}}} (Y_i - f_0(\boldsymbol{X}_i)) g(\boldsymbol{X}_i) - \mathbb{E}\{(Y - f_0(\boldsymbol{X})) g(\boldsymbol{X}) | \mathcal{D}\}\Big] = O_p\Big(\sigma \mathbb{E} \mathcal{R}_{n_{\text{val}}} \mathcal{G}_r + \sigma \frac{1}{\sqrt{n_{\text{val}}}} r\Big), \quad (1)$$

where $\mathcal{R}_{n_{\text{val}}} \mathcal{G}_r$ is the Rademacher complexity of $\mathcal{G}_r$, and $\sigma$ is the sub-Gaussian parameter of the noise $\varepsilon = Y - f_0(\boldsymbol{X})$. Given that $\mathcal{W} = \{\boldsymbol{w} \in [0,1]^M, \sum_{m=1}^{M} w_m = 1\}$, we have $\mathcal{R}_{n_{\text{val}}} \mathcal{G}_r \leq r\sqrt{2\log(M)/n}$. Since $\sigma$ is finite, and $n_{\text{val}}$ has the same order as the total sample size $n$, the first term on the right hand side of (1) is $O_p(\sigma \mathbb{E} \mathcal{R}_n \mathcal{G}_r) = O_p(r\sqrt{\log(M)}/\sqrt{n})$, and the second term is $O_p(\sigma\sqrt{\log(n_{\text{val}})/n_{\text{val}}} r) = O_p(r/\sqrt{n})$. Then (1) becomes

$$\sup_{g \in \mathcal{G}_r} \Big[\frac{1}{n_{\text{val}}} \sum_{i=1}^{n_{\text{val}}} (Y_i - f_0(\boldsymbol{X}_i)) g(\boldsymbol{X}_i) - \mathbb{E}\{(Y - f_0(\boldsymbol{X})) g(\boldsymbol{X}) | \mathcal{D}\}\Big] = O_p\Big(\frac{r\sqrt{\log(M)}}{\sqrt{n}}\Big). \quad (2)$$

Let $\boldsymbol{w}_0$ be the one-hot vector with entry 1 at the position corresponding to the model with the fastest convergence rate and 0 elsewhere. Then it is straightforward to show $\|f_0(\boldsymbol{X}) - \widehat{f}(\boldsymbol{X}; \boldsymbol{w}_0)\|_{L^2}^2 = \|f_0(\boldsymbol{X}) - \widetilde{f}(\boldsymbol{X})\|_{L^2}^2 = O_p(S^2)$. Moreover, since $\widehat{f}$ only depends on $\mathcal{D}$, and $\boldsymbol{X}$ is an independent sample drawn from the same distribution but independent of $\mathcal{D}$, we have

$$\mathbb{E}\{(Y - f_0(\boldsymbol{X}))(f_0(\boldsymbol{X}) - \widehat{f}(\boldsymbol{X}; \boldsymbol{w}_0)) | \mathcal{D}\}$$
$$= \mathbb{E}\Big[\mathbb{E}\Big\{(Y - f_0(\boldsymbol{X}))(f_0(\boldsymbol{X}) - \widehat{f}(\boldsymbol{X}; \boldsymbol{w}_0)) | \boldsymbol{X}, \mathcal{D}\Big\} | \mathcal{D}\Big]$$
$$= \mathbb{E}\Big\{\mathbb{E}(Y - f_0(\boldsymbol{X}) | \boldsymbol{X}, \mathcal{D})(f_0(\boldsymbol{X}) - \widehat{f}(\boldsymbol{X}; \boldsymbol{w}_0)) | \mathcal{D}\Big\}$$
$$= \mathbb{E}\{\mathbb{E}(\varepsilon | \boldsymbol{X})(f_0(\boldsymbol{X}) - \widehat{f}(\boldsymbol{X}; \boldsymbol{w}_0)) | \mathcal{D}\}$$
$$= 0. \quad (3)$$

Taking $r = S$, we have $g_{\boldsymbol{w}_0} \in \mathcal{G}_S$ and thus by (2) and (3), we have

$$\frac{1}{n_{\text{val}}} \sum_{i=1}^{n_{\text{val}}} \{Y_i - f_0(\boldsymbol{X}_i)\}\{f_0(\boldsymbol{X}_i) - \widehat{f}(\boldsymbol{X}_i; \boldsymbol{w}_0)\}$$

$$= \frac{1}{n_{\text{val}}} \sum_{i=1}^{n_{\text{val}}} (Y_i - f_0(\boldsymbol{X}_i))(f_0(\boldsymbol{X}_i) - \widehat{f}(\boldsymbol{X}_i; \boldsymbol{w}_0)) - \mathbb{E}\{(Y - f_0(\boldsymbol{X}))(f_0(\boldsymbol{X}) - \widehat{f}(\boldsymbol{X}; \boldsymbol{w}_0))|\mathcal{D}\}$$

$$+ \mathbb{E}\{(Y - f_0(\boldsymbol{X}))(f_0(\boldsymbol{X}) - \widehat{f}(\boldsymbol{X}; \boldsymbol{w}_0))|\mathcal{D}\}$$

$$\leq \sup_{g \in \mathcal{G}_r} [\frac{1}{n_{\text{val}}} \sum_{i=1}^{n_{\text{val}}} (Y_i - f_0(\boldsymbol{X}_i))g(\boldsymbol{X}_i) - \mathbb{E}\{(Y - f_0(\boldsymbol{X}))g(\boldsymbol{X})|\mathcal{D}\}]$$

$$= O_p(\sqrt{\frac{\log(M)}{n}}S), \tag{4}$$

Note that (3) remains valid when $\boldsymbol{w}_0$ is replaced by $\widehat{\boldsymbol{w}}$, because $\widehat{\boldsymbol{w}}$ is entirely determined by $\mathcal{D}$, and hence is independent of the new sample $\boldsymbol{X}$. Taking $r = \|f_0(\boldsymbol{X}) - \widehat{f}(\boldsymbol{X}; \widehat{\boldsymbol{w}})\|_{L^2}$, we have

$$\frac{1}{n_{\text{val}}} \sum_{i=1}^{n_{\text{val}}} \{Y_i - f_0(\boldsymbol{X}_i)\}\{f_0(\boldsymbol{X}_i) - \widehat{f}(\boldsymbol{X}_i; \widehat{\boldsymbol{w}})\}$$

$$= \frac{1}{n_{\text{val}}} \sum_{i=1}^{n_{\text{val}}} (Y_i - f_0(\boldsymbol{X}_i))(f_0(\boldsymbol{X}_i) - \widehat{f}(\boldsymbol{X}_i; \widehat{\boldsymbol{w}})) - \mathbb{E}\{(Y - f_0(\boldsymbol{X}))(f_0(\boldsymbol{X}) - \widehat{f}(\boldsymbol{X}; \widehat{\boldsymbol{w}}))|\mathcal{D}\}$$

$$+ \mathbb{E}\{(Y - f_0(\boldsymbol{X}))(f_0(\boldsymbol{X}) - \widehat{f}(\boldsymbol{X}; \widehat{\boldsymbol{w}}))|\mathcal{D}\}$$

$$\leq \sup_{g \in \mathcal{G}_r} [\frac{1}{n_{\text{val}}} \sum_{i=1}^{n_{\text{val}}} (Y_i - f_0(\boldsymbol{X}_i))g(\boldsymbol{X}_i) - \mathbb{E}\{(Y - f_0(\boldsymbol{X}))g(\boldsymbol{X})|\mathcal{D}\}]$$

$$= O_p(\sqrt{\frac{\log(M)}{n}}\|f_0(\boldsymbol{X}) - \widehat{f}(\boldsymbol{X}; \widehat{\boldsymbol{w}})\|_{L^2}). \tag{5}$$

Then we analyze $1/n_{\text{val}} \sum_{i=1}^{n_{\text{val}}} \{f_0(\boldsymbol{X}_i) - \widehat{f}(\boldsymbol{X}_i; \boldsymbol{w})\}^2$. Let $\mathcal{H}_r = \{h_{\boldsymbol{w}} = \{f_0(\boldsymbol{X}) - \widehat{f}(\boldsymbol{X}; \boldsymbol{w})\}^2 : \boldsymbol{w} \in \mathcal{W}, \text{Var}[\{f_0(\boldsymbol{X}) - \widehat{f}(\boldsymbol{X}; \boldsymbol{w})\}^2|\mathcal{D}] \leq r^2\}$. Similar to (1), we obtain

$$\sup_{h \in \mathcal{H}_r} \left[\frac{1}{n_{\text{val}}} \sum_{i=1}^{n_{\text{val}}} h(\boldsymbol{X}_i)^2 - \mathbb{E}\left\{h(\boldsymbol{X})^2|\mathcal{D}\right\}\right] = O_p\left(\mathbb{E}\mathcal{R}_n\mathcal{H}_r + \sqrt{\frac{1}{n_{\text{val}}}}r\right) = O_p\left(\sqrt{\frac{\log(M)}{n}}r\right). \tag{6}$$

Since $f_0$ and $\widehat{f}_m$ are uniformly bounded, there exists a positive constant $C$ such that

$$\text{Var}[\{f_0(\boldsymbol{X}) - \widehat{f}(\boldsymbol{X}; \boldsymbol{w})\}^2|\mathcal{D}]$$

$$\leq \mathbb{E}[\{f_0(\boldsymbol{X}) - \widehat{f}(\boldsymbol{X}; \boldsymbol{w})\}^4|\mathcal{D}]$$

$$\leq \|f_0(\boldsymbol{X}) - \widehat{f}(\boldsymbol{X}; \boldsymbol{w})\|_{L^2}^2 \|f_0(\boldsymbol{X}) - \widehat{f}(\boldsymbol{X}; \boldsymbol{w})\|_{\infty}^2$$

$$\leq C^2 \|f_0(\boldsymbol{X}) - \widehat{f}(\boldsymbol{X}; \boldsymbol{w})\|_{L^2}^2. \tag{7}$$

When taking $\boldsymbol{w} = \boldsymbol{w}_0$ and $r = CS$ in $\mathcal{H}_r$, we have $h_{\boldsymbol{w}_0} = \{f_0(\boldsymbol{X}) - \widehat{f}(\boldsymbol{X}; \boldsymbol{w}_0)\}^2 \in \mathcal{H}_{CS}$ because $\text{Var}[\{f_0(\boldsymbol{X}) - \widehat{f}(\boldsymbol{X}; \boldsymbol{w}_0)\}^2|\mathcal{D}] \leq C^2\|f_0(\boldsymbol{X}) - \widehat{f}(\boldsymbol{X}; \boldsymbol{w}_0)\|_{L^2}^2 = C^2S^2$ from (7). By (6), it follows that

$$\frac{1}{n_{\text{val}}} \sum_{i=1}^{n_{\text{val}}} \{f_0(\boldsymbol{X}_i) - \widehat{f}(\boldsymbol{X}_i; \boldsymbol{w}_0)\}^2$$

$$= \frac{1}{n_{\text{val}}} \sum_{i=1}^{n_{\text{val}}} \{f_0(\boldsymbol{X}_i) - \widehat{f}(\boldsymbol{X}_i; \boldsymbol{w}_0)\}^2 - \mathbb{E}\left[\{f_0(\boldsymbol{X}) - \widehat{f}(\boldsymbol{X}; \boldsymbol{w}_0)\}^2|\mathcal{D}\right] + \mathbb{E}\left[\{f_0(\boldsymbol{X}) - \widehat{f}(\boldsymbol{X}; \boldsymbol{w}_0)\}^2|\mathcal{D}\right]$$

$$\leq \sup_{h \in \mathcal{H}_{CS}} \left[\frac{1}{n_{\text{val}}} \sum_{i=1}^{n_{\text{val}}} h(\boldsymbol{X}_i)^2 - \mathbb{E}\left\{h(\boldsymbol{X})^2|\mathcal{D}\right\}\right] + \|f_0(\boldsymbol{X}) - \widehat{f}(\boldsymbol{X}; \boldsymbol{w}_0)\|_{L^2}^2$$

$$= O_p(\sqrt{\frac{\log(M)}{n}}S + S^2). \tag{8}$$

Similarly, taking $\boldsymbol{w} = \widehat{\boldsymbol{w}}$ and $r_{\widehat{\boldsymbol{w}}} = C\|f_0(\boldsymbol{X}) - \widehat{f}(\boldsymbol{X}; \widehat{\boldsymbol{w}})\|_{L^2}$, we have $h_{\widehat{\boldsymbol{w}}} = \{f_0(\boldsymbol{X}) - \widehat{f}(\boldsymbol{X}; \widehat{\boldsymbol{w}})\}^2 \in \mathcal{H}_{r_{\widehat{\boldsymbol{w}}}}$ and thus by (6), the following bound holds:

$$
\frac{1}{n_{\text{val}}} \sum_{i=1}^{n_{\text{val}}} \{f_0(\boldsymbol{X}_i) - \widehat{f}(\boldsymbol{X}_i; \widehat{\boldsymbol{w}})\}^2
$$

$$
= \frac{1}{n_{\text{val}}} \sum_{i=1}^{n_{\text{val}}} \{f_0(\boldsymbol{X}_i) - \widehat{f}(\boldsymbol{X}_i; \widehat{\boldsymbol{w}})\}^2 - \mathbb{E}\left[\{f_0(\boldsymbol{X}) - \widehat{f}(\boldsymbol{X}; \widehat{\boldsymbol{w}})\}^2 | \mathcal{D}\right] + \mathbb{E}\left[\{f_0(\boldsymbol{X}) - \widehat{f}(\boldsymbol{X}; \widehat{\boldsymbol{w}})\}^2 | \mathcal{D}\right]
$$

$$
\leq \sup_{h \in \mathcal{H}_{r_{\widehat{\boldsymbol{w}}}}} \left[\frac{1}{n_{\text{val}}} \sum_{i=1}^{n_{\text{val}}} h(\boldsymbol{X}_i)^2 - \mathbb{E}\left\{h(\boldsymbol{X})^2 | \mathcal{D}\right\}\right] + \|f_0(\boldsymbol{X}) - \widehat{f}(\boldsymbol{X}; \widehat{\boldsymbol{w}})\|_{L^2}^2
$$

$$
= O_p(\sqrt{\frac{\log(M)}{n}}\|f_0(\boldsymbol{X}) - \widehat{f}(\boldsymbol{X}; \widehat{\boldsymbol{w}})\|_{L^2}) + \|f_0(\boldsymbol{X}) - \widehat{f}(\boldsymbol{X}; \widehat{\boldsymbol{w}})\|_{L^2}^2. \tag{9}
$$

Although $\widehat{\boldsymbol{w}}$ depends on the validation data, the bound still holds because the supremum is taken over the class $\mathcal{H}_{r_{\widehat{\boldsymbol{w}}}}$. Combining (5) and (9), $\mathcal{L}(\widehat{\boldsymbol{w}})$ can be written as

$$
\mathcal{L}(\widehat{\boldsymbol{w}}) = \frac{1}{n_{\text{val}}} \sum_{i=1}^{n_{\text{val}}} \{Y_i - f_0(\boldsymbol{X}_i)\}^2 + \|f_0(\boldsymbol{X}) - \widehat{f}(\boldsymbol{X}; \widehat{\boldsymbol{w}})\|_{L^2}^2 + O_p(\sqrt{\frac{\log(M)}{n}})\|f_0(\boldsymbol{X}) - \widehat{f}(\boldsymbol{X}; \widehat{\boldsymbol{w}})\|_{L^2}^2, \tag{10}
$$

and according to (4) and (8), $\mathcal{L}(\boldsymbol{w}_0)$ can be written as

$$
\mathcal{L}(\boldsymbol{w}_0) = \frac{1}{n_{\text{val}}} \sum_{i=1}^{n_{\text{val}}} \{Y_i - f_0(\boldsymbol{X}_i)\}^2 + O_p(S^2 + \sqrt{\frac{\log(M)}{n}}S). \tag{11}
$$

Using the expansions in (10) and (11), together with the fact that $\mathcal{L}(\widehat{\boldsymbol{w}})$ minimizes the validation loss, i.e., $\mathcal{L}(\widehat{\boldsymbol{w}}) \leq \mathcal{L}(\boldsymbol{w}_0)$, we have

$$
\|f_0(\boldsymbol{X}) - \widehat{f}(\boldsymbol{X}; \widehat{\boldsymbol{w}})\|_{L^2}^2 + O_p(\sqrt{\frac{\log(M)}{n}})\|f_0(\boldsymbol{X}) - \widehat{f}(\boldsymbol{X}; \widehat{\boldsymbol{w}})\|_{L^2} = O_p(S^2 + \sqrt{\frac{\log(M)}{n}}S). \tag{12}
$$

By completing the square, (12) can be written as

$$
\left[\|f_0(\boldsymbol{X}) - \widehat{f}(\boldsymbol{X}; \widehat{\boldsymbol{w}})\|_{L^2} + O_p(\sqrt{\frac{\log(M)}{n}})\right]^2 = \{O_p(S + \sqrt{\frac{\log(M)}{n}})\}^2,
$$

i.e.,

$$
\|f_0(\boldsymbol{X}) - \widehat{f}(\boldsymbol{X}; \widehat{\boldsymbol{w}})\|_{L^2} = O_p(S + \sqrt{\frac{\log(M)}{n}}).
$$

This completes the proof of Theorem 1.

### B.3 PROOF OF THEOREM 2

The regression and multiclass-classification problems differ only in the choice of loss function. In classification problems, the performance measure of primary interest is the misclassification error (the 0–1 loss). Because this loss is discontinuous, it is typically replaced by a continuous surrogate, most commonly the cross-entropy loss. The surrogate is smooth and differentiable, which facilitates gradient-based optimization. Importantly, since cross-entropy is a calibrated surrogate, its excess risk dominates the squared excess misclassification risk; see Tewari & Bartlett (2007) for example. Let $\boldsymbol{w}_0$ be the one-hot vector that places 1 on the coordinate corresponding to the model with the fastest convergence rate and 0 elsewhere. Thus, if the excess misclassification risk of $\boldsymbol{w}_0$ is of order $S$, the corresponding cross-entropy excess risk satisfies

$$
\mathbb{E}\left\{f_0(\boldsymbol{X}) \log \frac{f_0(\boldsymbol{X})}{\widehat{f}(\boldsymbol{X}; \boldsymbol{w}_0)} | \mathcal{D}\right\} = O_p(S^2).
$$

Let

$$
\mathcal{H}_r = \left\{h_{\boldsymbol{w}}(\boldsymbol{X}) = \{\log f_0(\boldsymbol{X}) - \log \widehat{f}(\boldsymbol{X}; \boldsymbol{w})\} : w \in \mathcal{W}, \ \text{Var}\{h_{\boldsymbol{w}}(\boldsymbol{X}) | \mathcal{D}\} \leq r^2\right\}.
$$

By the multiplier inequality,

$$\sup_{h \in \mathcal{H}_r} \Big| \frac{1}{n_{\text{val}}} \sum_{i=1}^{n_{\text{val}}} \{Y_i - f_0(\boldsymbol{X}_i)\} h(\boldsymbol{X}_i) - \mathbb{E}[\{Y - f_0(\boldsymbol{X})\} h(\boldsymbol{X}) | \mathcal{D}] \Big| = O_p\Big(\gamma \, \mathbb{E}\mathcal{R}_{n_{\text{val}}} \mathcal{H}_r + \gamma \sqrt{\tfrac{\log(M)}{n}} \, r\Big), \quad (13)$$

where $\gamma^2 = \text{Var}(Y - f_0(\boldsymbol{X}))$ is finite because $\text{Var}(Y - f_0(\boldsymbol{X})) \leq 1/4$ in classification tasks. Moreover, we have $\mathbb{E}\mathcal{R}_{n_{\text{val}}} \mathcal{H}_r \leq C r \sqrt{\log(M)/n_{\text{val}}}$, and $n_{\text{val}}$ has the same order as $n$, then (13) becomes

$$\sup_{h \in \mathcal{H}_r} \Big| \frac{1}{n_{\text{val}}} \sum_{i=1}^{n_{\text{val}}} \{Y_i - f_0(\boldsymbol{X}_i)\} h(\boldsymbol{X}_i) - \mathbb{E}[\{Y - f_0(\boldsymbol{X})\} h(\boldsymbol{X}) | \mathcal{D}] \Big| = O_p\Big(\sqrt{\tfrac{\log(M)}{n}} \, r\Big). \quad (14)$$

Similarly,

$$\sup_{h \in \mathcal{H}_r} \Big| \frac{1}{n_{\text{val}}} \sum_{i=1}^{n_{\text{val}}} f_0(\boldsymbol{X}_i) h(\boldsymbol{X}_i) - \mathbb{E}[f_0(\boldsymbol{X}) h(\boldsymbol{X}) | \mathcal{D}] \Big| = O_p\Big(\sqrt{\tfrac{\log(M)}{n}} \, r\Big). \quad (15)$$

Moreover, we have

$$\mathbb{E}[\{Y - f_0(\boldsymbol{X})\} h(\boldsymbol{X}) | \mathcal{D}] = \mathbb{E}[\mathbb{E}\{Y - f_0(\boldsymbol{X}) | \boldsymbol{X}\} h(\boldsymbol{X}) | \mathcal{D}] = 0. \quad (16)$$

By (14)-(16), we have

$$\frac{1}{n_{\text{val}}} \sum_{i=1}^{n_{\text{val}}} Y_i h(\boldsymbol{X}_i)$$

$$= \frac{1}{n_{\text{val}}} \sum_{i=1}^{n_{\text{val}}} \{Y_i - f_0(\boldsymbol{X}_i)\} h(\boldsymbol{X}_i) + \frac{1}{n_{\text{val}}} \sum_{i=1}^{n_{\text{val}}} f_0(\boldsymbol{X}_i) h(\boldsymbol{X}_i)$$

$$\leq \sup_{h \in \mathcal{H}_r} \Big| \frac{1}{n_{\text{val}}} \sum_{i=1}^{n_{\text{val}}} \{Y_i - f_0(\boldsymbol{X}_i)\} h(\boldsymbol{X}_i) - \mathbb{E}[\{Y - f_0(\boldsymbol{X})\} h(\boldsymbol{X}) | \mathcal{D}] \Big|$$

$$+ \sup_{h \in \mathcal{H}_r} \Big| \frac{1}{n_{\text{val}}} \sum_{i=1}^{n_{\text{val}}} f_0(\boldsymbol{X}_i) h(\boldsymbol{X}_i) - \mathbb{E}[f_0(\boldsymbol{X}) h(\boldsymbol{X}) | \mathcal{D}] \Big|$$

$$+ \mathbb{E}[\{Y - f_0(\boldsymbol{X})\} h(\boldsymbol{X}) | \mathcal{D}] + \mathbb{E}[f_0(\boldsymbol{X}) h(\boldsymbol{X}) | \mathcal{D}]$$

$$= \mathbb{E}[f_0(\boldsymbol{X}) h(\boldsymbol{X}) | \mathcal{D}] + O_p\Big(\sqrt{\tfrac{\log(M)}{n}} r\Big). \quad (17)$$

Taking $r = S$ and $h(\boldsymbol{X}) = \log(f_0(\boldsymbol{X})/f_0(\boldsymbol{X}; \boldsymbol{w}_0))$, (17) becomes

$$\mathcal{L}(\boldsymbol{w}_0) = \frac{1}{n_{\text{val}}} \sum_{i=1}^{n_{\text{val}}} Y_i \log(\widehat{f}(\boldsymbol{X}_i; \boldsymbol{w}_0))$$

$$= \frac{1}{n_{\text{val}}} \sum_{i=1}^{n_{\text{val}}} Y_i \log \frac{f_0(\boldsymbol{X}_i)}{\widehat{f}(\boldsymbol{X}_i; \boldsymbol{w}_0)} - \frac{1}{n_{\text{val}}} \sum_{i=1}^{n_{\text{val}}} Y_i \log(f_0(\boldsymbol{X}_i))$$

$$= \Big| \mathbb{E}\{f_0(\boldsymbol{X}) \log \frac{f_0(\boldsymbol{X})}{\widehat{f}(\boldsymbol{X}; \boldsymbol{w}_0)} | \mathcal{D}\} \Big| + O_p\Big(\sqrt{\tfrac{\log(M)}{n}} S\Big) - \frac{1}{n_{\text{val}}} \sum_{i=1}^{n_{\text{val}}} Y_i \log(f_0(\boldsymbol{X}_i))$$

$$= O_p\Big(S^2 + \sqrt{\tfrac{\log(M)}{n}} S\Big) - \frac{1}{n_{\text{val}}} \sum_{i=1}^{n_{\text{val}}} Y_i \log(f_0(\boldsymbol{X}_i))$$

Similarly, we have

$$\mathcal{L}(\widehat{\boldsymbol{w}}) = \frac{1}{n_{\text{val}}} \sum_{i=1}^{n_{\text{val}}} Y_i \log(\widehat{f}(\boldsymbol{X}_i; \widehat{\boldsymbol{w}}))$$

$$= \mathbb{E}\{f_0(\boldsymbol{X}) \log \frac{f_0(\boldsymbol{X})}{\widehat{f}(\boldsymbol{X}; \widehat{\boldsymbol{w}})} | \mathcal{D}\} + O_p(\sqrt{\tfrac{\log(n)}{n}}) \sqrt{\mathbb{E}\{f_0(\boldsymbol{X}) \log \frac{f_0(\boldsymbol{X})}{\widehat{f}(\boldsymbol{X}; \widehat{\boldsymbol{w}})} | \mathcal{D}\}}$$

$$- \frac{1}{n_{\text{val}}} \sum_{i=1}^{n_{\text{val}}} Y_i \log(f_0(\boldsymbol{X}_i)).$$

Since $\mathcal{L}(\widehat{\boldsymbol{w}}) \leq \mathcal{L}(\boldsymbol{w}_0)$, This implies

$$\mathbb{E}\{f_0(\boldsymbol{X})\log\frac{f_0(\boldsymbol{X})}{\widehat{f}(\boldsymbol{X};\widehat{\boldsymbol{w}})}|\mathcal{D}\} + O_p(\sqrt{\frac{\log(M)}{n}})\sqrt{\mathbb{E}\{f_0(\boldsymbol{X})\log\frac{f_0(\boldsymbol{X})}{\widehat{f}(\boldsymbol{X};\widehat{\boldsymbol{w}})}|\mathcal{D}\}} = O_p\Big(S^2 + \sqrt{\frac{\log(M)}{n}}S\Big),$$

and thus $\sqrt{\mathbb{E}\{f_0(\boldsymbol{X})\log\{f_0(\boldsymbol{X})/\widehat{f}(\boldsymbol{X};\widehat{\boldsymbol{w}})\}|\mathcal{D}\}} = O_p(\sqrt{\frac{\log(M)}{n}} + S)$. This further implies that the misclassification rate $R_{0/1}(\widehat{\boldsymbol{w}}) = O_p(S + \sqrt{\frac{\log(M)}{n}})$.

### B.4 PROOF OF COROLLARIES

In addition to the moment conditions discussed in the main text, specific network architectures require additional structural assumptions, such as smoothness conditions, to achieve the desirable convergence rates. Below we present the concrete sets of assumptions for MLPs, CNNs, and RNNs under which the corresponding rates can be attained.

Now we introduce the definition of Hölder class. Hölder class $\mathcal{H}^{\beta}([0,1]^d, \mathbb{R}, B_0)$ is defined as

$$\mathcal{H}^{\beta}([0,1]^d, \mathbb{R}, B_0) = \Big\{f : [0,1]^d \to \mathbb{R} : \max_{\|\boldsymbol{\alpha}\|_1 \leq \lfloor\beta\rfloor}\|\partial^{\boldsymbol{\alpha}}f\|_{\infty} \leq B_0, \max_{\|\boldsymbol{\alpha}\|_1=\lfloor\beta\rfloor}\sup_{x\neq y}\frac{|\partial^{\boldsymbol{\alpha}}f(x) - \partial^{\boldsymbol{\alpha}}f(y)|}{\|x-y\|^{\beta-\lfloor\beta\rfloor}} \leq B_0\Big\},$$

where $\partial^{\boldsymbol{\alpha}} = \partial^{\alpha_1}\cdots\partial^{\alpha_d}$ with $\boldsymbol{\alpha} = (\alpha_1,\ldots,\alpha_d)^{\top} \in \mathbb{N}_0^d$ and $\|\boldsymbol{\alpha}\|_1 = \sum_{i=1}^{d}\alpha_i$.

**Condition 3.** *(i). The true target function $f_0$ belongs to a Hölder class $\mathcal{H}^{\beta}([0,1]^d, \mathbb{R}, B_0)$, and $\beta > 0$, $B_0 > 0$.*

*(ii). The candidate DNNs are in the function class of ReLU MLPs:*

$$\mathcal{F}_{B_0,\mathcal{W},\mathcal{D},\mathcal{S},d} = \{f : [0,1]^d \to \mathbb{R} : width \ \mathcal{W}, \ depth \ \mathcal{D}, \ total \ parameters \ \mathcal{S}, \ \|f\|_{\infty} \leq B_0\}, \quad (18)$$

*and at least one candidate model satisfies*

$$\mathcal{W} = O\Big(n^{\frac{d}{4(d+2\beta)}}\log_2(n)\Big), \quad \mathcal{D} = O\Big(n^{\frac{d}{4(d+2\beta)}}\log_2(n)\Big), \quad \mathcal{S} = O\Big(n^{\frac{3d}{4(d+2\beta)}}(\log n)^4\Big).$$

**Condition 4.** *(i). The true target function $f_0$ belongs to the Hölder class $\mathcal{H}^{\beta}([0,1]^d, \mathbb{R}, B_0)$ with $\beta > 0$ and $B_0 > 0$.*

*(ii). The candidate CNNs are in the function class of CNNs:*

$$\mathcal{F}_{B_0,\mathcal{W},\mathcal{D},\mathcal{S},s_{\min},s_{\max},d} = \Big\{f_{\mathrm{CNN}} : [0,1]^d \to \mathbb{R} : width \ \mathcal{W}, depth \ \mathcal{D}, \ total \ parameters \ \mathcal{S},$$

$$filter \ lengths \ s_{\min} \leq s^{(i)} \leq s_{\max}, \ \|f\|_{\infty} \leq B_0\Big\},$$

*and at least one candidate model satisfies*

$$\mathcal{D} \leq 42(\lfloor\beta\rfloor+1)^2 M\Big\lceil\log_2(8M)\Big\lceil\frac{\mathcal{W}-1}{s_{\min}-1}\Big\rceil\Big\rceil, \qquad 2 \leq s_{\min} \leq s_{\max} \leq \mathcal{W}, \qquad \mathcal{S} \leq 8\mathcal{W}\mathcal{D},$$

*where*

$$\mathcal{W} = 38^2(\lfloor\beta\rfloor+1)^4 d^{2\lfloor\beta\rfloor+2}N^2\big\lceil\log_2(8N)\big\rceil^2.$$

*(iii). The conditional probability $\mathbb{P}\{Y = 1 \mid \boldsymbol{X}\}$ is continuous on the support of $\mathcal{X}$, and the probability measure of $\boldsymbol{X}$ is absolutely continuous with respect to the Lebesgue measure.*

**Condition 5.** *(i). The true target function $f_0$ belongs to the Hölder class $f_0 \in H^{\beta}([0,1]^{d\times l}, \mathbb{R}^{d_y}, B_0)$ with $\beta > 0$ and $B_0 > 0$..*

*(ii). The candidate RNNs are in the function class of RNNs:*

$$\mathcal{F}_{B_0,\mathcal{W},\mathcal{D},d,d_y,l} = \Big\{f_{\mathrm{RNN}} : [0,1]^{d\times l} \to \mathbb{R}^{d_y} : width \ \mathcal{W}, depth \ \mathcal{D}, \|f\|_{\infty} \leq B_0\Big\},$$

*and there exists at least one candidate RNN satisfying*

$$\mathcal{W} = n^{\alpha}\log n, \qquad \mathcal{D} = n^{\frac{dl}{2dl+4\beta}-\alpha}\log n.$$

Corollary 5.3 in Jiao et al. (2023) guarantees that under Condition 3, the empirical risk minimizer $\widehat{f}$ satisfies

$$\mathbb{E}\|\widehat{f}(\boldsymbol{X}) - f_0(\boldsymbol{X})\|_{L^2}^2 \leq \mathcal{B}_0^5(\lfloor\beta\rfloor+1)^4 d^{2\lfloor\beta\rfloor+\beta\vee1} n^{-2\beta/(d+2\beta)}(\log n)^{11} = \widetilde{O}(n^{-2\beta/(d+2\beta)}).$$

Substituting $S = n^{-\beta/(d+2\beta)}$ into Theorem 1 implies that, when the candidate models are MLPs and include one with width $\mathcal{W}$, depth $\mathcal{D}$, and size $\mathcal{S}$, the asymptotic error bound of the model averaging estimator is $\widetilde{O}_p(n^{-2\beta/(d+2\beta)})$.

For the CNN and RNN settings, the results are analogous. Under Conditions 1, 4-5, the convergence rates of CNNs and RNNs is $O_p(n^{-2\beta/(d+2\beta)})$ and $O_p(n^{-2\beta/(dl+2\beta)})$ according to Theorem 4.6 in Shen et al. (2022) and Theorem 6 in Jiao et al. (2024), respectively. Substituting these rates into Theorem 1 yields Corollary 2 and 3.

### B.5 Proof of Theorem 3

Let

$$\widetilde{\mathcal{L}}(\boldsymbol{w}) = \mathcal{L}(\boldsymbol{w}) - \frac{1}{n_{\mathrm{val}}} \sum_{i=1}^{n_{\mathrm{val}}} (Y_i^2 - f_0(\boldsymbol{X}_i)^2).$$

Observe that the newly added component is unrelated to $\boldsymbol{w}$, so we have

$$\widehat{\boldsymbol{w}} = \underset{\boldsymbol{w}\in\mathcal{W}}{\arg\min}\, \mathcal{L}(\boldsymbol{w}) = \underset{\boldsymbol{w}\in\mathcal{W}}{\arg\min}\{R(\boldsymbol{w}) + \widetilde{\mathcal{L}}(\boldsymbol{w}) - R(\boldsymbol{w})\}.$$

According to Lemma 1, to prove Theorem 3, it is sufficient to prove

$$\sup_{\boldsymbol{w}\in\mathcal{W}} \frac{|R(\boldsymbol{w}) - R^*(\boldsymbol{w})|}{R^*(\boldsymbol{w})} = o_p(1) \tag{19}$$

and

$$\sup_{\boldsymbol{w}\in\mathcal{W}} \frac{|\widetilde{\mathcal{L}}(\boldsymbol{w}) - R(\boldsymbol{w})|}{R^*(\boldsymbol{w})} = o_p(1). \tag{20}$$

For (19), we have

$$\begin{aligned}
&\sup_{\boldsymbol{w}\in\mathcal{W}} \frac{|R(\boldsymbol{w}) - R^*(\boldsymbol{w})|}{R^*(\boldsymbol{w})} \\
\leq\quad & \frac{\sup_{\boldsymbol{w}\in\mathcal{W}}|R(\boldsymbol{w}) - R^*(\boldsymbol{w})|}{\inf_{\boldsymbol{w}\in\mathcal{W}} R^*(\boldsymbol{w})} \\
=\quad & \xi_n^{-1} \sup_{\boldsymbol{w}\in\mathcal{W}} |\|\widehat{f}(\boldsymbol{X};\boldsymbol{w}) - f_0(\boldsymbol{X})\|_{L^2}^2 - \|f^*(\boldsymbol{X};\boldsymbol{w}) - f_0(\boldsymbol{X})\|_{L^2}^2| \\
\leq\quad & \xi_n^{-1} \sup_{\boldsymbol{w}\in\mathcal{W}} |(\|\widehat{f}(\boldsymbol{X};\boldsymbol{w}) - f_0(\boldsymbol{X})\|_{L^2} + \|f^*(\boldsymbol{X};\boldsymbol{w}) - f_0(\boldsymbol{X})\|_{L^2})\|\widehat{f}(\boldsymbol{X};\boldsymbol{w}) - f^*(\boldsymbol{X};\boldsymbol{w})\|_{L^2}| \\
\leq\quad & C\xi_n^{-1} \sup_{\boldsymbol{w}\in\mathcal{W}} \|\widehat{f}(\boldsymbol{X};\boldsymbol{w}) - f^*(\boldsymbol{X};\boldsymbol{w})\|_{L^2} \\
=\quad & O_p(\xi_n^{-1}\phi_n) \\
=\quad & o_p(1),
\end{aligned}$$

where the last step comes from Condition 2. Thus (19) is obtained. Next, we will prove (20):

$$
\sup_{\boldsymbol{w} \in \mathcal{W}} \frac{|\widetilde{\mathcal{L}}(\boldsymbol{w}) - R(\boldsymbol{w})|}{R^*(\boldsymbol{w})}
$$

$$
\leq \quad \xi_n^{-1} \sup_{\boldsymbol{w} \in \mathcal{W}} |\widetilde{\mathcal{L}}(\boldsymbol{w}) - R(\boldsymbol{w})|
$$

$$
= \quad \xi_n^{-1} \sup_{\boldsymbol{w} \in \mathcal{W}} \left| \frac{1}{n_{\mathrm{val}}} \sum_{i=1}^{n_{\mathrm{val}}} \left\{ y_i - \widehat{f}(\boldsymbol{x}_i; \boldsymbol{w}) \right\}^2 - \frac{1}{n_{\mathrm{val}}} \sum_{i=1}^{n_{\mathrm{val}}} \left\{ y_i^2 - f_0(\boldsymbol{x}_i)^2 \right\} - \|\widehat{f}(\boldsymbol{X}; \boldsymbol{w}) - f_0(\boldsymbol{X})\|_{L^2}^2 \right|
$$

$$
\leq \quad \xi_n^{-1} \sup_{\boldsymbol{w} \in \mathcal{W}} \left| \frac{1}{n_{\mathrm{val}}} \sum_{i=1}^{n_{\mathrm{val}}} \left\{ \widehat{f}(\boldsymbol{x}_i; \boldsymbol{w}) - f_0(\boldsymbol{X}_i) \right\}^2 - \|\widehat{f}(\boldsymbol{X}; \boldsymbol{w}) - f_0(\boldsymbol{X})\|_{L^2}^2 \right|
$$

$$
+ \quad \xi_n^{-1} \sup_{\boldsymbol{w} \in \mathcal{W}} \left| \frac{1}{n_{\mathrm{val}}} \sum_{i=1}^{n_{\mathrm{val}}} \{\widehat{f}(\boldsymbol{X}_i; \boldsymbol{w}) - f^*(\boldsymbol{X}_i; \boldsymbol{w})\}\{Y_i - f_0(\boldsymbol{X}_i)\} \right|.
$$

$$
+ \quad \xi_n^{-1} \sup_{\boldsymbol{w} \in \mathcal{W}} \left| \frac{1}{n_{\mathrm{val}}} \sum_{i=1}^{n_{\mathrm{val}}} f^*(\boldsymbol{X}_i; \boldsymbol{w})\{Y_i - f_0(\boldsymbol{X}_i)\} \right|. \tag{21}
$$

To prove (20), it is sufficient to prove the next three equations

$$
\xi_n^{-1} \sup_{\boldsymbol{w} \in \mathcal{W}} \left| \frac{1}{n_{\mathrm{val}}} \sum_{i=1}^{n_{\mathrm{val}}} \left\{ \widehat{f}(\boldsymbol{x}_i; \boldsymbol{w}) - f_0(\boldsymbol{X}_i) \right\}^2 - \|\widehat{f}(\boldsymbol{X}; \boldsymbol{w}) - f_0(\boldsymbol{X})\|_{L^2}^2 \right| = o_p(1), \tag{22}
$$

$$
\xi_n^{-1} \sup_{\boldsymbol{w} \in \mathcal{W}} \left| \frac{1}{n_{\mathrm{val}}} \sum_{i=1}^{n_{\mathrm{val}}} \{\widehat{f}(\boldsymbol{X}_i; \boldsymbol{w}) - f^*(\boldsymbol{X}_i; \boldsymbol{w})\}\{Y_i - f_0(\boldsymbol{X}_i)\} \right| = o_p(1), \tag{23}
$$

and

$$
\xi_n^{-1} \sup_{\boldsymbol{w} \in \mathcal{W}} \left| \frac{1}{n_{\mathrm{val}}} \sum_{i=1}^{n_{\mathrm{val}}} f^*(\boldsymbol{X}_i; \boldsymbol{w})\{Y_i - f_0(\boldsymbol{X}_i)\} \right| = o_p(1). \tag{24}
$$

Let $\mathcal{H}_r = \{h_{\boldsymbol{w}} = \{\widehat{f}(\boldsymbol{X}; \boldsymbol{w}) - f_0(\boldsymbol{X})\}^2 : \boldsymbol{w} \in \mathcal{W}\}$. By the multiplier inequality, we have

$$
\sup_{\boldsymbol{w} \in \mathcal{W}} \left| \frac{1}{n_{\mathrm{val}}} \sum_{i=1}^{n_{\mathrm{val}}} \left\{ \widehat{f}(\boldsymbol{x}_i; \boldsymbol{w}) - f_0(\boldsymbol{X}_i) \right\}^2 - \|\widehat{f}(\boldsymbol{X}; \boldsymbol{w}) - f_0(\boldsymbol{X})\|_{L^2}^2 \right|
$$

$$
= \quad \sup_{h \in \mathcal{H}_r} \left| \frac{1}{n_{\mathrm{val}}} \sum_{i=1}^{n_{\mathrm{val}}} h(\boldsymbol{X}_i) - \mathbb{E}\{h(\boldsymbol{X})|\mathcal{D}\} \right|
$$

$$
= \quad O_p\left( \sqrt{\frac{\log(M)}{n}} \right), \tag{25}
$$

The difference between (6) and (25) is that we set $r = 1$ here. This is because we take the supremum over all $\boldsymbol{w} \in \mathcal{W}$, instead of localizing to a particular neighborhood of $\boldsymbol{w}$. And the bound in (25) is also bigger than that in (6).

Therefore, by Condition 2, (22) can be proved by

$$
\xi_n^{-1} \sup_{\boldsymbol{w} \in \mathcal{W}} \left| \frac{1}{n_{\mathrm{val}}} \sum_{i=1}^{n_{\mathrm{val}}} \left\{ \widehat{f}(\boldsymbol{x}_i; \boldsymbol{w}) - f_0(\boldsymbol{X}_i) \right\}^2 - \|\widehat{f}(\boldsymbol{X}; \boldsymbol{w}) - f_0(\boldsymbol{X})\|_{L^2}^2 \right| = O_p(\xi_n^{-1} \log(M)/\sqrt{n}) = o_p(1). \tag{26}
$$

For (23), we have

$$\xi_n^{-1} \sup_{\boldsymbol{w} \in \mathcal{W}} \left| \frac{1}{n_{\text{val}}} \sum_{i=1}^{n_{\text{val}}} \{\widehat{f}(\boldsymbol{X}_i; \boldsymbol{w}) - f^*(\boldsymbol{X}_i; \boldsymbol{w})\}\{Y_i - f_0(\boldsymbol{X}_i)\} \right|$$

$$= \xi_n^{-1} \sup_{\boldsymbol{w} \in \mathcal{W}} \left| \frac{1}{n_{\text{val}}} \sum_{i=1}^{n_{\text{val}}} \{\widehat{f}(\boldsymbol{X}_i; \boldsymbol{w}) - f^*(\boldsymbol{X}_i; \boldsymbol{w})\}\{Y_i - f_0(\boldsymbol{X}_i)\} \right.$$

$$\left. - \mathbb{E}[\{\widehat{f}(\boldsymbol{X}; \boldsymbol{w}) - f^*(\boldsymbol{X}; \boldsymbol{w})\}\{Y - f_0(\boldsymbol{X})\}|\mathcal{D}] \right|$$

$$+ \sup_{\boldsymbol{w} \in \mathcal{W}} \left| \mathbb{E}[\{\widehat{f}(\boldsymbol{X}; \boldsymbol{w}) - f^*(\boldsymbol{X}; \boldsymbol{w})\}\{Y - f_0(\boldsymbol{X})\}|\mathcal{D}] \right|$$

$$= \xi_n^{-1} O_p(\sqrt{\log(M)/n})$$

$$= o_p(1). \tag{27}$$

For (24), according to the Chebyshev's inequality, we have

$$\Pr \left\{ \xi_n^{-1} \sup_{\boldsymbol{w} \in \mathcal{W}} \left| \frac{1}{n_{\text{val}}} \sum_{i=1}^{n_{\text{val}}} f^*(\boldsymbol{X}_i; \boldsymbol{w}) \varepsilon_i \right| > \nu \right\}$$

$$= \Pr \left\{ \xi_n^{-1} \sup_{\boldsymbol{w} \in \mathcal{W}} \left| \frac{1}{n_{\text{val}}} \sum_{i=1}^{n_{\text{val}}} \sum_{m=1}^{M} w_m f_m^*(\boldsymbol{X}_i) \varepsilon_i \right| > \nu \right\}$$

$$\leq \sum_{m=1}^{M} \Pr \left\{ \xi_n^{-1} \left| \frac{1}{n_{\text{val}}} \sum_{i=1}^{n_{\text{val}}} f_m^*(\boldsymbol{X}_i) \varepsilon_i \right| > \nu \right\}$$

$$\leq \sum_{m=1}^{M} \xi_n^{-2} \nu^{-2} \text{Var} \left\{ \frac{1}{n_{\text{val}}} \sum_{i=1}^{n_{\text{val}}} f_m^*(\boldsymbol{X}_i) \varepsilon_i \right\}$$

$$= \xi_n^{-2} \nu^{-2} n^{-1} M \text{Var} \{f_m^*(\boldsymbol{X}) \varepsilon\}$$

$$= M \xi_n^{-2} \nu^{-2} n^{-1}$$

$$= o_p(1) \tag{28}$$

according to Condition 2, which means $\xi_n^{-1} \sup_{\boldsymbol{w} \in \mathcal{W}} \left| \frac{1}{n_{\text{val}}} \sum_{i=1}^{n_{\text{val}}} f^*(\boldsymbol{X}_i; \boldsymbol{w}) \varepsilon_i \right| = o_p(1)$ by Condition 2.

Equations (26)-(28) imply that (22)-(24) hold, and thus we obtain (20). Since (19) and (20) hold, we complete the proof of Theorem 3.

## C  More Experimental Details

**Training resouces.** We use the A800 80G for training the models with PyTorch version 2.5.1.

## D  Additional Results

We report the ratio of WDE MSE to oracle MSE across parameter discrepancies and sample sizes in structural misspecification in Table 5.

We compare with other weighted methods in structural misspecification from Table 6 to Table 8.

## E  Future Work

While the current theoretical results establish asymptotic error bounds for the proposed weighted deep ensemble estimator, extending these results to non-asymptotic settings remains an important direction for future research. Such analysis could provide tighter guarantees in finite-sample regimes, which are often relevant in practical applications. Another promising avenue is to investigate principled

Table 5: Ratio of WDE MSE to oracle MSE across parameter discrepancies and sample sizes

| $\alpha$ | $\Delta = 30000$ | | | $\Delta = 50000$ | | | $\Delta = 100000$ | | |
|---|---|---|---|---|---|---|---|---|---|
| | $N$ | | | $N$ | | | $N$ | | |
| | 5000 | 10000 | 15000 | 5000 | 10000 | 15000 | 5000 | 10000 | 15000 |
| 0.9 | 1.0066 | 1.0038 | 1.0020 | 1.0061 | 1.0033 | 1.0013 | 1.0076 | 1.0032 | 1.0031 |
| 0.8 | 1.0062 | 1.0031 | 1.0022 | 1.0046 | 1.0022 | 1.0020 | 1.0054 | 1.0034 | 1.0025 |
| 0.7 | 1.0045 | 1.0030 | 1.0023 | 1.0078 | 1.0041 | 1.0021 | 1.0061 | 1.0034 | 1.0021 |
| 0.6 | 1.0056 | 1.0020 | 1.0025 | 1.0087 | 1.0034 | 1.0018 | 1.0068 | 1.0038 | 1.0022 |
| 0.5 | 1.0059 | 1.0022 | 1.0014 | 1.0055 | 1.0036 | 1.0025 | 1.0040 | 1.0030 | 1.0014 |
| 0.4 | 1.0061 | 1.0028 | 1.0034 | 1.0081 | 1.0027 | 1.0025 | 1.0051 | 1.0041 | 1.0010 |
| 0.3 | 1.0058 | 1.0031 | 1.0022 | 1.0057 | 1.0027 | 1.0019 | 1.0046 | 1.0038 | 1.0036 |
| 0.2 | 1.0091 | 1.0037 | 1.0024 | 1.0078 | 1.0027 | 1.0018 | 1.0066 | 1.0028 | 1.0014 |
| 0.1 | 1.0088 | 1.0029 | 1.0017 | 1.0062 | 1.0028 | 1.0015 | 1.0058 | 1.0049 | 1.0021 |
| 0.0 | 1.0069 | 1.0020 | 1.0015 | 1.0036 | 1.0034 | 1.0021 | 1.0079 | 1.0045 | 1.0023 |

Table 6: Relative MSE comparison for sample size $N = 5,000$

| $\Delta$ | $1 - \alpha$ | WDE | In-sample Ensemble | Greedy Ensemble |
|---|---|---|---|---|
| | | Mean $\pm$ Std | Mean $\pm$ Std | Mean $\pm$ Std |
| 30,000 | 0.1 | $0.417 \pm 0.089$ | $0.421 \pm 0.090$ | $0.418 \pm 0.091$ |
| | 0.2 | $0.408 \pm 0.081$ | $0.416 \pm 0.081$ | $0.413 \pm 0.083$ |
| | 0.3 | $0.398 \pm 0.083$ | $0.402 \pm 0.086$ | $0.400 \pm 0.085$ |
| | 0.4 | $0.387 \pm 0.095$ | $0.394 \pm 0.097$ | $0.391 \pm 0.098$ |
| | 0.5 | $0.397 \pm 0.146$ | $0.399 \pm 0.146$ | $0.399 \pm 0.146$ |
| | 0.6 | $0.360 \pm 0.048$ | $0.366 \pm 0.052$ | $0.363 \pm 0.049$ |
| | 0.7 | $0.347 \pm 0.043$ | $0.356 \pm 0.048$ | $0.351 \pm 0.044$ |
| | 0.8 | $0.341 \pm 0.034$ | $0.349 \pm 0.037$ | $0.343 \pm 0.036$ |
| | 0.9 | $0.337 \pm 0.031$ | $0.343 \pm 0.034$ | $0.340 \pm 0.032$ |
| | 1.0 | $0.334 \pm 0.031$ | $0.342 \pm 0.030$ | $0.336 \pm 0.031$ |
| 50,000 | 0.1 | $0.437 \pm 0.061$ | $0.442 \pm 0.063$ | $0.439 \pm 0.063$ |
| | 0.2 | $0.476 \pm 0.192$ | $0.483 \pm 0.190$ | $0.479 \pm 0.191$ |
| | 0.3 | $0.419 \pm 0.067$ | $0.424 \pm 0.070$ | $0.423 \pm 0.070$ |
| | 0.4 | $0.391 \pm 0.050$ | $0.396 \pm 0.051$ | $0.394 \pm 0.051$ |
| | 0.5 | $0.399 \pm 0.091$ | $0.382 \pm 0.041$ | $0.402 \pm 0.103$ |
| | 0.6 | $0.354 \pm 0.041$ | $0.361 \pm 0.041$ | $0.357 \pm 0.041$ |
| | 0.7 | $0.344 \pm 0.030$ | $0.350 \pm 0.032$ | $0.345 \pm 0.031$ |
| | 0.8 | $0.339 \pm 0.032$ | $0.345 \pm 0.035$ | $0.340 \pm 0.033$ |
| | 0.9 | $0.334 \pm 0.027$ | $0.342 \pm 0.029$ | $0.337 \pm 0.028$ |
| | 1.0 | $0.330 \pm 0.027$ | $0.335 \pm 0.027$ | $0.334 \pm 0.027$ |
| 100,000 | 0.1 | $0.348 \pm 0.043$ | $0.357 \pm 0.044$ | $0.349 \pm 0.043$ |
| | 0.2 | $0.349 \pm 0.042$ | $0.357 \pm 0.047$ | $0.352 \pm 0.044$ |
| | 0.3 | $0.346 \pm 0.043$ | $0.355 \pm 0.043$ | $0.348 \pm 0.043$ |
| | 0.4 | $0.342 \pm 0.042$ | $0.348 \pm 0.043$ | $0.344 \pm 0.042$ |
| | 0.5 | $0.340 \pm 0.039$ | $0.347 \pm 0.040$ | $0.342 \pm 0.040$ |
| | 0.6 | $0.335 \pm 0.035$ | $0.343 \pm 0.038$ | $0.338 \pm 0.036$ |
| | 0.7 | $0.332 \pm 0.034$ | $0.339 \pm 0.037$ | $0.335 \pm 0.035$ |
| | 0.8 | $0.331 \pm 0.026$ | $0.336 \pm 0.028$ | $0.331 \pm 0.026$ |
| | 0.9 | $0.330 \pm 0.025$ | $0.337 \pm 0.026$ | $0.334 \pm 0.027$ |
| | 1.0 | $0.327 \pm 0.024$ | $0.336 \pm 0.021$ | $0.331 \pm 0.026$ |

Table 7: Relative MSE comparison for sample size $N = 10,000$

| $\Delta$ | $1 - \alpha$ | WDE Mean $\pm$ Std | In-sample Ensemble Mean $\pm$ Std | Greedy Ensemble Mean $\pm$ Std |
|---|---|---|---|---|
| 30,000 | 0.1 | $0.399 \pm 0.108$ | $0.404 \pm 0.108$ | $0.404 \pm 0.110$ |
| | 0.2 | $0.396 \pm 0.117$ | $0.400 \pm 0.116$ | $0.399 \pm 0.120$ |
| | 0.3 | $0.377 \pm 0.094$ | $0.380 \pm 0.093$ | $0.379 \pm 0.094$ |
| | 0.4 | $0.369 \pm 0.069$ | $0.371 \pm 0.068$ | $0.371 \pm 0.069$ |
| | 0.5 | $0.351 \pm 0.053$ | $0.355 \pm 0.051$ | $0.352 \pm 0.054$ |
| | 0.6 | $0.339 \pm 0.044$ | $0.344 \pm 0.045$ | $0.342 \pm 0.047$ |
| | 0.7 | $0.328 \pm 0.038$ | $0.331 \pm 0.038$ | $0.330 \pm 0.038$ |
| | 0.8 | $0.327 \pm 0.034$ | $0.330 \pm 0.034$ | $0.329 \pm 0.036$ |
| | 0.9 | $0.318 \pm 0.032$ | $0.321 \pm 0.032$ | $0.319 \pm 0.032$ |
| | 1.0 | $0.320 \pm 0.034$ | $0.323 \pm 0.035$ | $0.323 \pm 0.036$ |
| 50,000 | 0.1 | $0.399 \pm 0.052$ | $0.405 \pm 0.052$ | $0.401 \pm 0.052$ |
| | 0.2 | $0.398 \pm 0.063$ | $0.401 \pm 0.065$ | $0.400 \pm 0.064$ |
| | 0.3 | $0.384 \pm 0.060$ | $0.388 \pm 0.059$ | $0.387 \pm 0.061$ |
| | 0.4 | $0.396 \pm 0.105$ | $0.398 \pm 0.104$ | $0.397 \pm 0.105$ |
| | 0.5 | $0.349 \pm 0.048$ | $0.353 \pm 0.048$ | $0.351 \pm 0.049$ |
| | 0.6 | $0.334 \pm 0.039$ | $0.338 \pm 0.040$ | $0.336 \pm 0.040$ |
| | 0.7 | $0.324 \pm 0.036$ | $0.327 \pm 0.036$ | $0.325 \pm 0.036$ |
| | 0.8 | $0.318 \pm 0.034$ | $0.321 \pm 0.033$ | $0.319 \pm 0.034$ |
| | 0.9 | $0.314 \pm 0.035$ | $0.320 \pm 0.039$ | $0.316 \pm 0.036$ |
| | 1.0 | $0.311 \pm 0.029$ | $0.314 \pm 0.030$ | $0.313 \pm 0.029$ |

Table 8: Relative MSE comparison for sample size $N = 15,000$

| $\Delta$ | $1 - \alpha$ | WDE Mean $\pm$ Std | In-sample Ensemble Mean $\pm$ Std | Greedy Ensemble Mean $\pm$ Std |
|---|---|---|---|---|
| 30,000 | 0.1 | $0.381 \pm 0.086$ | $0.384 \pm 0.084$ | $0.384 \pm 0.089$ |
| | 0.2 | $0.381 \pm 0.106$ | $0.384 \pm 0.105$ | $0.386 \pm 0.111$ |
| | 0.3 | $0.350 \pm 0.052$ | $0.354 \pm 0.052$ | $0.351 \pm 0.053$ |
| | 0.4 | $0.340 \pm 0.047$ | $0.343 \pm 0.046$ | $0.342 \pm 0.047$ |
| | 0.5 | $0.328 \pm 0.040$ | $0.333 \pm 0.039$ | $0.330 \pm 0.041$ |
| | 0.6 | $0.323 \pm 0.032$ | $0.327 \pm 0.032$ | $0.326 \pm 0.032$ |
| | 0.7 | $0.323 \pm 0.062$ | $0.326 \pm 0.062$ | $0.325 \pm 0.065$ |
| | 0.8 | $0.309 \pm 0.026$ | $0.314 \pm 0.029$ | $0.312 \pm 0.029$ |
| | 0.9 | $0.335 \pm 0.106$ | $0.338 \pm 0.105$ | $0.336 \pm 0.106$ |
| | 1.0 | $0.303 \pm 0.023$ | $0.307 \pm 0.024$ | $0.305 \pm 0.024$ |
| 50,000 | 0.1 | $0.388 \pm 0.049$ | $0.390 \pm 0.049$ | $0.389 \pm 0.050$ |
| | 0.2 | $0.379 \pm 0.052$ | $0.382 \pm 0.053$ | $0.383 \pm 0.055$ |
| | 0.3 | $0.384 \pm 0.071$ | $0.387 \pm 0.071$ | $0.387 \pm 0.071$ |
| | 0.4 | $0.360 \pm 0.049$ | $0.363 \pm 0.049$ | $0.362 \pm 0.049$ |
| | 0.5 | $0.335 \pm 0.040$ | $0.339 \pm 0.040$ | $0.337 \pm 0.041$ |
| | 0.6 | $0.331 \pm 0.040$ | $0.334 \pm 0.042$ | $0.333 \pm 0.042$ |
| | 0.7 | $0.312 \pm 0.025$ | $0.314 \pm 0.025$ | $0.313 \pm 0.026$ |
| | 0.8 | $0.306 \pm 0.024$ | $0.308 \pm 0.025$ | $0.306 \pm 0.024$ |
| | 0.9 | $0.301 \pm 0.023$ | $0.303 \pm 0.023$ | $0.303 \pm 0.023$ |
| | 1.0 | $0.302 \pm 0.023$ | $0.304 \pm 0.022$ | $0.303 \pm 0.022$ |
| 100,000 | 0.1 | $0.306 \pm 0.027$ | $0.308 \pm 0.027$ | $0.307 \pm 0.027$ |
| | 0.2 | $0.303 \pm 0.025$ | $0.306 \pm 0.025$ | $0.305 \pm 0.026$ |
| | 0.3 | $0.304 \pm 0.027$ | $0.306 \pm 0.026$ | $0.305 \pm 0.027$ |
| | 0.4 | $0.301 \pm 0.023$ | $0.304 \pm 0.024$ | $0.302 \pm 0.024$ |
| | 0.5 | $0.299 \pm 0.024$ | $0.302 \pm 0.024$ | $0.300 \pm 0.024$ |
| | 0.6 | $0.297 \pm 0.022$ | $0.299 \pm 0.022$ | $0.298 \pm 0.022$ |
| | 0.7 | $0.297 \pm 0.021$ | $0.299 \pm 0.021$ | $0.298 \pm 0.021$ |
| | 0.8 | $0.294 \pm 0.019$ | $0.296 \pm 0.019$ | $0.295 \pm 0.019$ |
| | 0.9 | $0.297 \pm 0.018$ | $0.299 \pm 0.018$ | $0.298 \pm 0.018$ |
| | 1.0 | $0.299 \pm 0.020$ | $0.301 \pm 0.021$ | $0.300 \pm 0.021$ |

approaches for determining the number and composition of candidate models in the ensemble. Understanding how model diversity and ensemble size affect performance could lead to more efficient and adaptive ensemble design strategies.

# F  REAL-WORLD DATASETS

**Datasets.** (i) CIFAR-10: 60,000 32x32 color images in 10 classes for image classification. (ii) CIFAR-10-C: A corrupted benchmark derived from CIFAR-10 for evaluating model robustness. It applies 15 common corruptions and each at 5 severity levels. It is widely used to test the out-of-distribution (OOD) robustness of image classifiers and to measure performance degradation under real-world perturbations.

**Candidate models.** We consider the following neural networks (i) ResNet18 (He et al., 2016), (ii)PreActResNet18, (iii) StochasticDepth18 (Huang et al., 2016), (iv) MobileNetV2 (Sandler et al., 2018), and (v) GoogleNet (Szegedy et al., 2015).

**Evaluation metrics.** (i) Accuracy (Acc), which reports the fraction of correctly classified samples. (ii) Expected Calibration Error (ECE), which measures how well model confidence aligns with its actual accuracy, with lower values indicating better calibration.

Table 9: Comparison of Accuracy on CIFAR10 and CIFAR10-C

| Corruption | DE | | | | | EW | WDE |
|---|---|---|---|---|---|---|---|
| | PreActResNet18 | ResNet18 | StochasticDepth18 | GoogleNet | MobileNetV2 | | |
| clean | 91.18% | 91.10% | 91.20% | 90.99% | 87.81% | 92.48% | **92.76%** |
| brightness | 29.20% | 27.44% | 27.51% | 28.61% | 25.78% | 31.61% | **32.45%** |
| contrast | 16.05% | 15.58% | 15.36% | 12.17% | 13.71% | 13.21% | **16.53%** |
| defocus blur | 21.07% | 18.79% | 18.95% | 17.95% | 19.00% | 20.42% | **22.03%** |
| elastic transform | 20.89% | 18.02% | 17.85% | 18.13% | 19.95% | 20.43% | **21.83%** |
| fog | 16.68% | 17.30% | 14.43% | 14.16% | 16.16% | 15.68% | **17.48%** |
| frost | 23.75% | 22.87% | 19.67% | 22.75% | 20.57% | 24.17% | **26.31%** |
| gaussian blur | 19.97% | 18.52% | 18.91% | 17.06% | 17.45% | 19.33% | **20.40%** |
| gaussian noise | 22.17% | 19.85% | 18.17% | 27.91% | 12.45% | 27.91% | **28.39%** |
| glass blur | 21.31% | 23.65% | 19.13% | 21.29% | 18.90% | 24.58% | **25.88%** |
| impulse noise | 21.44% | 23.52% | 20.45% | 26.46% | 14.44% | 27.45% | **27.54%** |
| jpeg compression | 24.84% | 20.98% | 19.14% | 21.12% | 22.09% | 23.68% | **25.90%** |
| motion blur | 19.55% | 17.20% | 18.44% | 14.83% | 16.16% | 17.26% | **20.88%** |
| pixelate | 26.56% | 23.28% | 21.61% | 23.95% | 22.71% | 26.59% | **27.76%** |
| saturate | 29.53% | 28.26% | 27.10% | 28.26% | 26.27% | 31.14% | **33.72%** |
| shot noise | 23.00% | 21.30% | 18.83% | 27.49% | 14.30% | 26.16% | **27.92%** |
| snow | 25.47% | 26.29% | 22.08% | 26.51% | 22.27% | 29.24% | **31.57%** |
| spatter | 26.18% | 26.12% | 23.81% | 26.14% | 22.47% | 29.65% | **32.79%** |
| speckle noise | 22.46% | 21.27% | 18.42% | 26.66% | 14.49% | 26.73% | **27.65%** |
| zoom blur | **19.27%** | 17.47% | 17.80% | 15.96% | 16.78% | 18.47% | 19.23% |

Table 10: Comparison of ECE on CIFAR10 and CIFAR10-C

| Corruption | DE | | | | | EW | WDE |
|---|---|---|---|---|---|---|---|
| | PreActResNet18 | ResNet18 | StochasticDepth18 | GoogleNet | MobileNetV2 | | |
| clean | 0.0599 | 0.0564 | 0.0599 | 0.0583 | 0.0768 | 0.0507 | **0.0438** |
| brightness | 0.5594 | 0.6016 | 0.5594 | 0.5166 | 0.4912 | 0.5545 | **0.4475** |
| contrast | 0.4831 | 0.5829 | 0.4831 | 0.6327 | 0.7247 | 0.5056 | **0.4517** |
| defocusblur | 0.5684 | 0.6574 | 0.5684 | 0.5504 | 0.5937 | 0.6046 | **0.4743** |
| elastictransform | 0.5881 | 0.7021 | 0.5881 | 0.5715 | 0.5672 | 0.6227 | **0.4946** |
| fog | 0.5403 | 0.6363 | 0.5403 | 0.6132 | 0.6530 | 0.6498 | **0.5311** |
| frost | 0.5944 | 0.6404 | 0.5944 | 0.5708 | 0.5288 | 0.6138 | **0.4933** |
| gaussian blur | 0.5580 | 0.6273 | 0.5580 | 0.5378 | 0.6269 | 0.5963 | **0.4671** |
| gaussian noise | 0.6481 | 0.7243 | 0.6481 | 0.5931 | 0.5971 | 0.6559 | **0.5394** |
| glass blur | 0.6309 | 0.6251 | 0.6309 | 0.5893 | 0.5350 | 0.6615 | **0.4907** |
| impulse noise | 0.6776 | 0.6509 | 0.6776 | 0.5364 | 0.6188 | 0.6989 | **0.5331** |
| jpeg compression | 0.5825 | 0.6811 | 0.5825 | 0.5798 | 0.5303 | 0.6073 | **0.4991** |
| motion blur | 0.5636 | 0.6607 | 0.5636 | 0.6221 | 0.6631 | 0.5909 | **0.5379** |
| pixelate | 0.5703 | 0.6418 | 0.5703 | 0.5532 | 0.5129 | 0.5828 | **0.4795** |
| saturate | 0.5611 | 0.5903 | 0.5611 | 0.5179 | 0.4896 | 0.5666 | **0.4341** |
| shot noise | 0.6394 | 0.6945 | 0.6394 | 0.5199 | 0.5783 | 0.6510 | **0.5142** |
| snow | 0.6056 | 0.6146 | 0.6056 | 0.5410 | 0.4925 | 0.6036 | **0.4615** |
| spatter | 0.6006 | 0.6121 | 0.6006 | 0.5478 | 0.4979 | 0.6087 | **0.4383** |
| speckle noise | 0.6452 | 0.6837 | 0.6452 | 0.5210 | 0.5766 | 0.6609 | **0.5105** |
| zoom blur | 0.5722 | 0.6627 | 0.5722 | 0.5528 | 0.6187 | 0.6084 | **0.4817** |

**Additional comparison in pre-trained model ensembles on ImageNet.** We apply the pre-trained model used in Wortsman et al. (2022), where different random initializations of the same architecture (CLIP ViT-B/32) are provided. In Table 11, we report the results on ImageNet, as well as the performance averaged over four natural distribution shifts: ImageNetV2, ImageNet-R, ImageNet-Sketch, and ImageNet-A.

We also note that Under this homogeneous-architecture setting, our weighted ensembling achieves clear improvements.

| Method | ImageNet | Dist.shifts |
|---|---|---|
| Best individual model | 80.39 | 49.31 |
| Second best model | 79.89 | 45.50 |
| Excluding soup results | – | – |
| Ensemble | 81.19 | 52.14 |
| Greedy ensemble | 81.90 | 50.19 |
| WDE (homogeneous) | **81.96** | **52.39** |

Table 11: Performance comparison on ImageNet and distribution shifts.

# G    COLLECTIVE BLINDNESS

**Diversity in predictions**: We begin by initializing each architecture 10 times, including both shallower models (e.g., depth 18) and deeper ones (e.g., depth 152), and generating predictions on the CIFAR-10 test set. We then visualize the similarity among these predictions using t-SNE. As shown in the Figure 3, models with the same architecture tend to cluster together, whereas different architectures produce more diverse prediction.

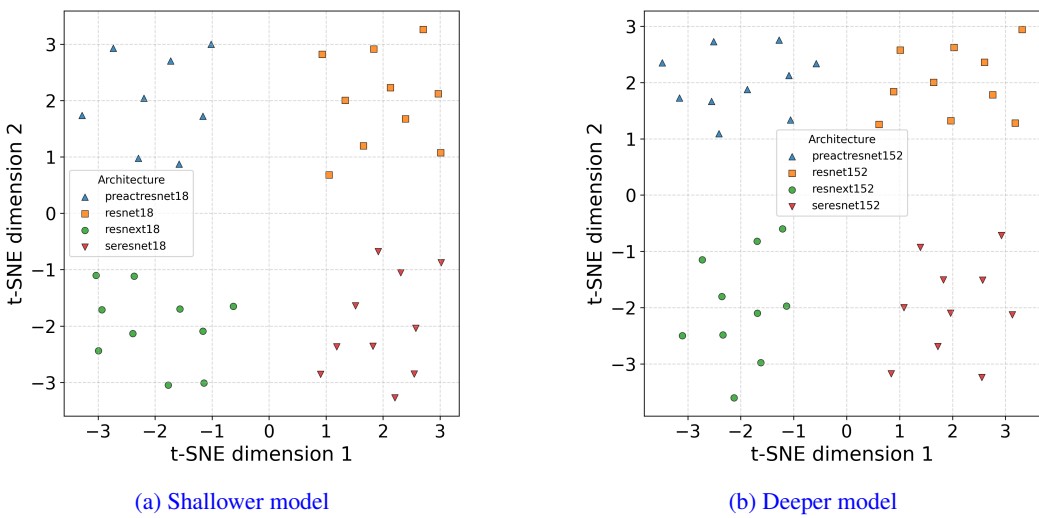

(a) Shallower model

(b) Deeper model

Figure 3: t-SNE visualization of CIFAR-10 predictions

**Evaluation on collective blindness**. To quantify this effect, we introduce three metrics that measure different levels of error correlation within the ensemble:

For each model $m$, We define the error set on slightly hard samples to ensure that correlations are measured only on non-trivial mistakes: $E_m = \{n : |\hat{f}_m(\mathbf{X}_n) - Y_n| > \tau\}$, and for the ensemble, $E_{\text{ens}} = \{n : |\hat{f}(\mathbf{X}_n; \mathbf{w}) - Y_n| > \tau\}$. To measure how two models make errors on the same samples, we define their error correlation $\text{CoErr}(i, j) = \frac{|E_i \cap E_j|}{|E_i \cup E_j|}$, and for each model $m$, the correction with the ensemble is $\text{CoErr}(\text{ens}, m) = \frac{|E_{\text{ens}} \cap E_m|}{|E_{\text{ens}} \cup E_m|}$.

(i) Model–Model Error Correlation (mCo-Err). The average error correction across all model pairs is

$$\text{mCoErr} = \frac{1}{M(M-1)} \sum_{i \neq j} \text{CoErr}(i, j),$$

which measures the typical correction of errors across individual models.

(ii) Ensemble–Model Error Correlation (EnCo-Err). The weighted ensemble–model correction is

$$\text{EnCoErr} = \sum_{m=1}^{M} w_m \, \text{CoErr}(\text{ens}, m),$$

which measures the correction between ensemble errors and those of its base models.

(iii) Tail Error Rate (TER).

$$\text{TER} = \frac{1}{N} \sum_{n=1}^{N} \mathbf{1}\Big( |\hat{f}(\mathbf{X}_n; \mathbf{w}) - Y_n| > \tau \Big),$$

which measures the proportion of predictions that incur non-trivial errors.

We report the corresponding results under variable, inherent and structural misspecification in Tables 12 - 17. These metrics jointly indicate that (a) collective blindness is weaker in diverse architectures than in single-architecture settings. (b) weighted ensembles reduce error correlations with individual models, and (c) extreme prediction errors become less frequent. These results offer a quantitative assessment of collective blindness and demonstrate that WDE effectively mitigates it.

| Task | # Missing | mCoErr ↓ | | EnCoErr ↓ | | |
| --- | --- | --- | --- | --- | --- | --- |
| | | Diverse | Same | DE | EW | WDE |
| Nested | 0 | $0.6837 \pm 0.0565$ | $0.7669 \pm 0.0352$ | $0.8220 \pm 0.0748$ | $0.7765 \pm 0.0366$ | $0.5467 \pm 0.0691$ |
| | 1 | $0.7137 \pm 0.0654$ | $0.7835 \pm 0.0407$ | $0.8303 \pm 0.0598$ | $0.7774 \pm 0.0405$ | $0.5854 \pm 0.0788$ |
| | 3 | $0.7850 \pm 0.0535$ | $0.8210 \pm 0.0316$ | $0.8864 \pm 0.0626$ | $0.8262 \pm 0.0320$ | $0.6819 \pm 0.0655$ |
| | 5 | $0.8335 \pm 0.0162$ | $0.8507 \pm 0.0216$ | $0.9158 \pm 0.0598$ | $0.8616 \pm 0.0457$ | $0.7481 \pm 0.0210$ |
| | 7 | $0.8935 \pm 0.0430$ | $0.9120 \pm 0.0257$ | $0.9532 \pm 0.0358$ | $0.9342 \pm 0.0263$ | $0.8376 \pm 0.0592$ |
| Interaction | 0 | $0.5813 \pm 0.0968$ | $0.6711 \pm 0.0748$ | $0.7009 \pm 0.1175$ | $0.6825 \pm 0.0666$ | $0.4435 \pm 0.0802$ |
| | 1 | $0.7014 \pm 0.0851$ | $0.7743 \pm 0.0346$ | $0.7709 \pm 0.0431$ | $0.7577 \pm 0.0352$ | $0.5798 \pm 0.0852$ |
| | 3 | $0.7674 \pm 0.0536$ | $0.8206 \pm 0.0235$ | $0.8509 \pm 0.0262$ | $0.8415 \pm 0.0355$ | $0.6566 \pm 0.0667$ |
| | 5 | $0.7965 \pm 0.0475$ | $0.8459 \pm 0.0285$ | $0.9014 \pm 0.0394$ | $0.8599 \pm 0.0380$ | $0.6991 \pm 0.0633$ |
| | 7 | $0.8959 \pm 0.0436$ | $0.9208 \pm 0.0421$ | $0.9517 \pm 0.0417$ | $0.9313 \pm 0.0452$ | $0.8471 \pm 0.0673$ |
| Nonlinear | 0 | $0.6294 \pm 0.0348$ | $0.7030 \pm 0.0596$ | $0.7324 \pm 0.0733$ | $0.7067 \pm 0.0620$ | $0.4797 \pm 0.0359$ |
| | 1 | $0.6990 \pm 0.0682$ | $0.7484 \pm 0.0668$ | $0.7608 \pm 0.0391$ | $0.7358 \pm 0.0655$ | $0.5626 \pm 0.0860$ |
| | 3 | $0.7963 \pm 0.0896$ | $0.8589 \pm 0.0373$ | $0.8804 \pm 0.0511$ | $0.8431 \pm 0.0406$ | $0.7054 \pm 0.0924$ |
| | 5 | $0.8736 \pm 0.0364$ | $0.8882 \pm 0.0395$ | $0.9429 \pm 0.0528$ | $0.9039 \pm 0.0463$ | $0.8071 \pm 0.0513$ |
| | 7 | $0.9102 \pm 0.0191$ | $0.9320 \pm 0.0204$ | $0.9562 \pm 0.0213$ | $0.9489 \pm 0.0122$ | $0.8582 \pm 0.0255$ |

Table 12: Comparison of mCoErr and EnCoErr under variable misspecification when $N = 5000$.

| Task | # Missing | TER ↓ | | |
| --- | --- | --- | --- | --- |
| | | DE | EW | WDE |
| Nested | 0 | $0.2311 \pm 0.0207$ | $0.2118 \pm 0.0118$ | $0.2008 \pm 0.0131$ |
| | 1 | $0.2332 \pm 0.0184$ | $0.2220 \pm 0.0138$ | $0.2079 \pm 0.0134$ |
| | 3 | $0.2474 \pm 0.0117$ | $0.2398 \pm 0.0076$ | $0.2368 \pm 0.0126$ |
| | 5 | $0.2616 \pm 0.0102$ | $0.2608 \pm 0.0113$ | $0.2559 \pm 0.0137$ |
| | 7 | $0.2781 \pm 0.0100$ | $0.2749 \pm 0.0113$ | $0.2742 \pm 0.0104$ |
| Interaction | 0 | $0.0359 \pm 0.0082$ | $0.0325 \pm 0.0061$ | $0.0315 \pm 0.0053$ |
| | 1 | $0.1012 \pm 0.0088$ | $0.0988 \pm 0.0089$ | $0.0979 \pm 0.0083$ |
| | 3 | $0.1844 \pm 0.0119$ | $0.1834 \pm 0.0131$ | $0.1802 \pm 0.0094$ |
| | 5 | $0.2157 \pm 0.0084$ | $0.2167 \pm 0.0112$ | $0.2160 \pm 0.0135$ |
| | 7 | $0.2287 \pm 0.0084$ | $0.2294 \pm 0.0090$ | $0.2292 \pm 0.0074$ |
| Nonlinear | 0 | $0.0377 \pm 0.0050$ | $0.0384 \pm 0.0035$ | $0.0357 \pm 0.0044$ |
| | 1 | $0.0636 \pm 0.0289$ | $0.0657 \pm 0.0286$ | $0.0633 \pm 0.0281$ |
| | 3 | $0.1148 \pm 0.0238$ | $0.1150 \pm 0.0242$ | $0.1138 \pm 0.0257$ |
| | 5 | $0.1735 \pm 0.0353$ | $0.1730 \pm 0.0356$ | $0.1718 \pm 0.0352$ |
| | 7 | $0.2183 \pm 0.0305$ | $0.2169 \pm 0.0279$ | $0.2164 \pm 0.0293$ |

Table 13: Comparison of TER under variable misspecification when $N = 5000$.

| Task | $N$ | mCoErr ↓ | | EnCoErr ↓ | | |
|------|-----|---------|------|----|----|-----|
| | | Diverse | Same | DE | EW | WDE |
| Infinite | 5000 | $0.3420 \pm 0.0267$ | $0.3775 \pm 0.0131$ | $0.6000 \pm 0.0516$ | $0.5804 \pm 0.0501$ | $0.3126 \pm 0.0277$ |
| | 10000 | $0.2649 \pm 0.0292$ | $0.3257 \pm 0.0282$ | $0.4880 \pm 0.0515$ | $0.4457 \pm 0.0477$ | $0.1954 \pm 0.0293$ |
| | 15000 | $0.1869 \pm 0.0147$ | $0.2900 \pm 0.0134$ | $0.4458 \pm 0.0492$ | $0.4393 \pm 0.0486$ | $0.1679 \pm 0.0117$ |
| Square | 5000 | $0.6582 \pm 0.1324$ | $0.7105 \pm 0.1117$ | $0.9472 \pm 0.0430$ | $0.9313 \pm 0.0668$ | $0.6359 \pm 0.1310$ |
| | 10000 | $0.7795 \pm 0.2307$ | $0.8026 \pm 0.2062$ | $0.9543 \pm 0.0669$ | $0.9597 \pm 0.0520$ | $0.6942 \pm 0.1659$ |
| | 15000 | $0.8015 \pm 0.2164$ | $0.8210 \pm 0.2017$ | $0.9840 \pm 0.0210$ | $0.9676 \pm 0.0455$ | $0.6263 \pm 0.0934$ |

Table 14: Comparison of mCoErr and EnCoErr under inherent misspecification.

| Task | $N$ | TER ↓ | | |
|------|-----|-------|------|-----|
| | | DE | EW | WDE |
| Infinite | 5000 | $0.1304 \pm 0.0110$ | $0.1104 \pm 0.0037$ | $0.0037 \pm 0.0001$ |
| | 10000 | $0.0892 \pm 0.0072$ | $0.0826 \pm 0.0023$ | $0.0024 \pm 0.0005$ |
| | 15000 | $0.0769 \pm 0.0070$ | $0.0632 \pm 0.0006$ | $0.0014 \pm 0.0002$ |
| Square | 5000 | $0.3034 \pm 0.0517$ | $0.2945 \pm 0.0403$ | $0.2466 \pm 0.0576$ |
| | 10000 | $0.2906 \pm 0.0307$ | $0.2792 \pm 0.0346$ | $0.2313 \pm 0.0453$ |
| | 15000 | $0.2971 \pm 0.0552$ | $0.2854 \pm 0.0277$ | $0.2222 \pm 0.0271$ |

Table 15: Comparison of TER under inherent misspecification for varying sample sizes $N$.

| $\Delta$ | $1-\alpha$ | mCoErr ↓ | | EnCoErr ↓ | | |
|----------|-----------|---------|------|----|----|-----|
| | | Diverse | Same | DE | EW | WDE |
| 30000 | 0.1 | $0.7060 \pm 0.0547$ | $0.7413 \pm 0.0939$ | $0.8054 \pm 0.1079$ | $0.7808 \pm 0.0543$ | $0.4887 \pm 0.0839$ |
| | 0.2 | $0.7124 \pm 0.0677$ | $0.7648 \pm 0.1372$ | $0.7883 \pm 0.0903$ | $0.8091 \pm 0.0833$ | $0.5250 \pm 0.0875$ |
| | 0.3 | $0.7012 \pm 0.0927$ | $0.7624 \pm 0.1667$ | $0.7968 \pm 0.1450$ | $0.7152 \pm 0.1279$ | $0.4903 \pm 0.0708$ |
| | 0.4 | $0.6753 \pm 0.0287$ | $0.6914 \pm 0.0353$ | $0.7552 \pm 0.0500$ | $0.7374 \pm 0.0307$ | $0.4697 \pm 0.0772$ |
| | 0.5 | $0.6855 \pm 0.0798$ | $0.7262 \pm 0.1486$ | $0.7733 \pm 0.1060$ | $0.7708 \pm 0.0918$ | $0.5121 \pm 0.1009$ |
| | 0.6 | $0.7473 \pm 0.1006$ | $0.7841 \pm 0.1530$ | $0.8098 \pm 0.1163$ | $0.7734 \pm 0.1133$ | $0.5152 \pm 0.1103$ |
| | 0.7 | $0.7161 \pm 0.1135$ | $0.7465 \pm 0.1424$ | $0.8243 \pm 0.1241$ | $0.7639 \pm 0.0996$ | $0.4345 \pm 0.0399$ |
| | 0.8 | $0.7240 \pm 0.1123$ | $0.7700 \pm 0.1610$ | $0.8369 \pm 0.1331$ | $0.8326 \pm 0.1305$ | $0.5365 \pm 0.0829$ |
| | 0.9 | $0.7191 \pm 0.1260$ | $0.7267 \pm 0.1506$ | $0.8246 \pm 0.1291$ | $0.8980 \pm 0.1400$ | $0.5612 \pm 0.1236$ |
| | 1.0 | $0.7032 \pm 0.0575$ | $0.7119 \pm 0.1166$ | $0.7635 \pm 0.0954$ | $0.7809 \pm 0.1000$ | $0.5012 \pm 0.1087$ |
| 50000 | 0.1 | $0.7612 \pm 0.0834$ | $0.8114 \pm 0.1346$ | $0.8323 \pm 0.1084$ | $0.9278 \pm 0.0777$ | $0.5572 \pm 0.1032$ |
| | 0.2 | $0.7273 \pm 0.0802$ | $0.8123 \pm 0.1617$ | $0.8272 \pm 0.1297$ | $0.8264 \pm 0.0742$ | $0.5081 \pm 0.0566$ |
| | 0.3 | $0.7474 \pm 0.1498$ | $0.8093 \pm 0.1650$ | $0.8514 \pm 0.1299$ | $0.9076 \pm 0.1326$ | $0.5570 \pm 0.1704$ |
| | 0.4 | $0.7313 \pm 0.1133$ | $0.7505 \pm 0.1365$ | $0.8420 \pm 0.1185$ | $0.7808 \pm 0.1083$ | $0.5016 \pm 0.0989$ |
| | 0.5 | $0.7533 \pm 0.1086$ | $0.7914 \pm 0.1482$ | $0.8281 \pm 0.1172$ | $0.8556 \pm 0.1043$ | $0.5234 \pm 0.1279$ |
| | 0.6 | $0.7196 \pm 0.1370$ | $0.7357 \pm 0.1439$ | $0.7859 \pm 0.1278$ | $0.7754 \pm 0.1375$ | $0.4910 \pm 0.1022$ |
| | 0.7 | $0.7331 \pm 0.1097$ | $0.7871 \pm 0.1531$ | $0.8479 \pm 0.1241$ | $0.7444 \pm 0.1161$ | $0.4888 \pm 0.0877$ |
| | 0.8 | $0.7503 \pm 0.1289$ | $0.7773 \pm 0.1593$ | $0.8249 \pm 0.1233$ | $0.7848 \pm 0.1387$ | $0.4843 \pm 0.0969$ |
| | 0.9 | $0.6789 \pm 0.0921$ | $0.6962 \pm 0.1296$ | $0.8073 \pm 0.1111$ | $0.7546 \pm 0.0664$ | $0.4915 \pm 0.1199$ |
| | 1.0 | $0.7032 \pm 0.0575$ | $0.7119 \pm 0.1166$ | $0.7635 \pm 0.0954$ | $0.7809 \pm 0.1000$ | $0.5012 \pm 0.1087$ |
| 100000 | 0.1 | $0.6972 \pm 0.0614$ | $0.7365 \pm 0.1425$ | $0.7828 \pm 0.1284$ | $0.7319 \pm 0.0341$ | $0.4785 \pm 0.1068$ |
| | 0.2 | $0.6764 \pm 0.0604$ | $0.6667 \pm 0.0677$ | $0.7569 \pm 0.1221$ | $0.6963 \pm 0.0618$ | $0.4494 \pm 0.0851$ |
| | 0.3 | $0.6821 \pm 0.0627$ | $0.7055 \pm 0.1094$ | $0.7603 \pm 0.1318$ | $0.7845 \pm 0.0961$ | $0.4989 \pm 0.1032$ |
| | 0.4 | $0.7349 \pm 0.0615$ | $0.7698 \pm 0.1324$ | $0.8319 \pm 0.1166$ | $0.7595 \pm 0.1053$ | $0.4746 \pm 0.0602$ |
| | 0.5 | $0.6648 \pm 0.0564$ | $0.7063 \pm 0.1106$ | $0.7871 \pm 0.1185$ | $0.7393 \pm 0.0723$ | $0.4433 \pm 0.0573$ |
| | 0.6 | $0.7252 \pm 0.0907$ | $0.7847 \pm 0.1508$ | $0.8142 \pm 0.1339$ | $0.7955 \pm 0.0997$ | $0.4958 \pm 0.0939$ |
| | 0.7 | $0.6417 \pm 0.0349$ | $0.6840 \pm 0.1144$ | $0.7664 \pm 0.1049$ | $0.7311 \pm 0.0324$ | $0.4406 \pm 0.0701$ |
| | 0.8 | $0.6768 \pm 0.0569$ | $0.7048 \pm 0.1127$ | $0.8013 \pm 0.1238$ | $0.7503 \pm 0.0461$ | $0.4779 \pm 0.1099$ |
| | 0.9 | $0.6833 \pm 0.0752$ | $0.7117 \pm 0.1135$ | $0.7576 \pm 0.1141$ | $0.7071 \pm 0.0697$ | $0.4727 \pm 0.1140$ |
| | 1.0 | $0.7032 \pm 0.0575$ | $0.7119 \pm 0.1166$ | $0.7635 \pm 0.0954$ | $0.7809 \pm 0.1000$ | $0.5012 \pm 0.1087$ |

Table 16: Comparison of mCoErr and EnCoErr under structural misspecification when $N = 5000$.

| $\Delta$ | $1-\alpha$ | TER $\downarrow$ | | |
| --- | --- | --- | --- | --- |
| | | DE | EW | WDE |
| | 0.1 | $0.1504 \pm 0.0443$ | $0.1191 \pm 0.0363$ | $0.1159 \pm 0.0352$ |
| | 0.2 | $0.1405 \pm 0.0505$ | $0.1160 \pm 0.0337$ | $0.1101 \pm 0.0325$ |
| | 0.3 | $0.1502 \pm 0.0458$ | $0.1061 \pm 0.0178$ | $0.1021 \pm 0.0178$ |
| | 0.4 | $0.1334 \pm 0.0263$ | $0.1028 \pm 0.0141$ | $0.0985 \pm 0.0146$ |
| | 0.5 | $0.1356 \pm 0.0482$ | $0.0984 \pm 0.0141$ | $0.0947 \pm 0.0152$ |
| 30000 | 0.6 | $0.1417 \pm 0.0511$ | $0.0958 \pm 0.0157$ | $0.0923 \pm 0.0150$ |
| | 0.7 | $0.1367 \pm 0.0386$ | $0.0923 \pm 0.0152$ | $0.0874 \pm 0.0145$ |
| | 0.8 | $0.1325 \pm 0.0595$ | $0.0943 \pm 0.0186$ | $0.0910 \pm 0.0197$ |
| | 0.9 | $0.1374 \pm 0.0701$ | $0.1033 \pm 0.0377$ | $0.0999 \pm 0.0382$ |
| | 1.0 | $0.1146 \pm 0.0350$ | $0.0927 \pm 0.0144$ | $0.0882 \pm 0.0165$ |
| | 0.1 | $0.1743 \pm 0.0556$ | $0.1441 \pm 0.0416$ | $0.1392 \pm 0.0421$ |
| | 0.2 | $0.1629 \pm 0.0421$ | $0.1261 \pm 0.0192$ | $0.1212 \pm 0.0214$ |
| | 0.3 | $0.1704 \pm 0.0614$ | $0.1382 \pm 0.0574$ | $0.1338 \pm 0.0568$ |
| | 0.4 | $0.1636 \pm 0.0491$ | $0.1175 \pm 0.0254$ | $0.1136 \pm 0.0265$ |
| | 0.5 | $0.1583 \pm 0.0606$ | $0.1202 \pm 0.0470$ | $0.1165 \pm 0.0471$ |
| 50000 | 0.6 | $0.1436 \pm 0.0513$ | $0.1044 \pm 0.0279$ | $0.1018 \pm 0.0267$ |
| | 0.7 | $0.1508 \pm 0.0535$ | $0.0972 \pm 0.0160$ | $0.0945 \pm 0.0177$ |
| | 0.8 | $0.1498 \pm 0.0544$ | $0.0917 \pm 0.0117$ | $0.0878 \pm 0.0137$ |
| | 0.9 | $0.1220 \pm 0.0397$ | $0.0874 \pm 0.0155$ | $0.0832 \pm 0.0158$ |
| | 1.0 | $0.1146 \pm 0.0350$ | $0.0927 \pm 0.0144$ | $0.0882 \pm 0.0165$ |
| | 0.1 | $0.1193 \pm 0.0348$ | $0.0873 \pm 0.0210$ | $0.0825 \pm 0.0175$ |
| | 0.2 | $0.1233 \pm 0.0366$ | $0.0900 \pm 0.0183$ | $0.0870 \pm 0.0186$ |
| | 0.3 | $0.1279 \pm 0.0471$ | $0.0909 \pm 0.0207$ | $0.0863 \pm 0.0217$ |
| | 0.4 | $0.1317 \pm 0.0546$ | $0.0937 \pm 0.0211$ | $0.0912 \pm 0.0208$ |
| | 0.5 | $0.1227 \pm 0.0363$ | $0.0883 \pm 0.0183$ | $0.0848 \pm 0.0170$ |
| 100000 | 0.6 | $0.1376 \pm 0.0495$ | $0.0946 \pm 0.0211$ | $0.0902 \pm 0.0204$ |
| | 0.7 | $0.1071 \pm 0.0206$ | $0.0867 \pm 0.0134$ | $0.0822 \pm 0.0149$ |
| | 0.8 | $0.1118 \pm 0.0311$ | $0.0864 \pm 0.0119$ | $0.0809 \pm 0.0124$ |
| | 0.9 | $0.1150 \pm 0.0388$ | $0.0891 \pm 0.0150$ | $0.0836 \pm 0.0136$ |
| | 1.0 | $0.1146 \pm 0.0350$ | $0.0927 \pm 0.0144$ | $0.0882 \pm 0.0165$ |

Table 17: Comparison of TER under structural misspecification when $N = 5000$.

