# OpenReview forum: "Weighted Deep Ensemble Under Misspecification"
_ICLR.cc/2026/Conference — Submitted to ICLR 2026_

### Official Review · Reviewer_Afi7 · 2025-10-25

**Soundness:** 2
**Presentation:** 1
**Contribution:** 1
**Rating:** 2
**Confidence:** 4

**Summary:**

The submission is concerned with weighted ensembles of neural networks.

**Strengths:**

Weighted ensembles of neural networks are a relevant topic.

**Weaknesses:**

## A review and comparison with the state of the art is missing.

A review and comparison with the state of the art is missing.

First, weighed ensembles are in no way new. The basic idea goes back to stacking

David H. Wolpert. Stacked generalization. Neural Networks, 5(2):241–259, 1992

For a more neural network focussed paper see for example:

Anders Krogh and Peter Sollich. Statistical mechanics of ensemble learning. Physical Review E, 55(1) 1997.

Second, there are a lot of papers dealing with — theoretically well motivated — weighting of deep neural networks.
For example:

Andrés R. Masegosa. Learning under model misspecification: Applications to variational and ensemble methods.
In Advances in Neural Processing Systems (NeurIPS), volume 33, 2020

Luis A. Ortega, Rafael Cabañas, and Andres Masegosa. Diversity and generalization in neural network ensembles.
In International Conference on Artificial Intelligence and Statistics (AISTATS),  2022

Third, there are also generalisation bounds for weighted ensembles, which are applicable to neural network ensembles:

Andrés R. Masegosa, Stephan S. Lorenzen, Christian Igel, and Yevgeny Seldin. Second order PAC-Bayesian
bounds for the weighted majority vote. In Advances in Neural Processing Systems (NeurIPS), 2020

Yi-Shan Wu, Andrés R. Masegosa, Stephan S. Lorenzen, Christian Igel, and Yevgeny Seldin. Chebyshev-Cantelli
PAC-Bayes-Bennett inequality for the weighted majority vote. In Advances in Neural Processing Systems529
(NeurIPS), 2021

Hauptvogel, Igel. On Uniform, Bayesian, and PAC-Bayesian Deep Ensembles. arXiv:2406.05469 [cs.LG], 2024

In addition, I was missing a reference to

Lars Kai Hansen and Peter Salamon. Neural network ensembles. IEEE Transactions on Pattern Analysis and
Machine Intelligence, 12(10):993–1001, 1990.


## The paper lack mathematical rigour.

The paper lack mathematical rigour. Examples:

Line 165: incomplete, meaningless statement. Seems like a part of the equation is missing.

Lines 180-190: Statement lack rigour: Example makes no sense without stating something about h.
In expectation over all hypothesises?

Line 229: Should this be $\hat{f}$ on the RHS?

Condition 1: $\epsilon$ is not defined

## There are several misleading statements.

Theorem 1: The theorem only talks about n. Should there not be specific assumptions about n_train and n_val.
E.g., does this hold for small constant n_val?

Just consider the first three sentences:

„Model misspecification in statistics arises  […]  inclusion of
irrelevant variables […]. In such cases, the best possible approximation […] still maintains a significant approximation error from the true
function“:  Could you please cite a rigorous  theoretical result that states that adding irrelevant variables must cause a significant approximation error

„In deep learning, the neural networks are always assumed to be well-specified.“: In  general not true. I do not assume that - and I do not know anybody who does.

„To the best of our knowledge, this is the first study to offer a theoretical guarantee for weighted deep ensemble.“: Clearly wrong, see the many reference given above and references therein as starting points.

## General comments

Inherent misspecification: How much does this matter on digital computers (aka  „in practice“)?
This should be discussed.

While I think it is interesting to study weighted neural network ensembles, I have to say that I could not identify exciting novel insights in the manuscript.
Theorem 1 does not come as a surprise and is not put into relation to other (e.g., PAC-Bayesian) bounds.


## Minor comments

* „Ensemble methods is“ -> „Ensemble methods are“

* l 79: Strange references for deep ensembles. Why not not

Lars Kai Hansen and Peter Salamon. Neural network ensembles. IEEE Transactions on Pattern Analysis and
Machine Intelligence, 12(10):993–1001, 1990.

and the later cited

Balaji Lakshminarayanan, Alexander Pritzel, and Charles Blundell. Simple and scalable predictive uncertainty
estimation using deep ensembles. In Advances in Neural Processing Systems (NeurIPS), volume 30, 2017.

* I think the discussion of mode misspecification should go along with a brief discussion of parametric vs non-parametric models.
* Line 282: Why „without loss of generality“?

**Questions:**

see "Weaknesses" above

---

> ### Author Response · Authors · 2025-11-23
>
> Dear Reviewer Afi7,
>
> We would like to express our sincere gratitude to the reviewers for their time and effort in reviewing our manuscript. Their insightful feedback has led to significant improvements in the presentation and depth of our work. We provide our responses to the comments in the following.
>
> **(W1-1, W1-3).  Previous Work.**
>
> |Method|Misspecification|GeneralizationBound|DeterministicOptimalWeights|AsymptoticOracleOptimality|
> |-|-|-|-|-|
> |Stacking[1][2]|✗|✓|✓|✗|
> |PAC-Bayesian[3][4]|✗|✓|✗|✗|
> |**Ours**|✓|✓|✓|✓|
>
> 1. Traditional stacking methods combine multiple predictors through linear aggregation, but their theoretical guarantees have not extended to modern deep neural networks. Moreover, they do not analyze ensembles under misspecification, nor do they provide asymptotic optimality guarantees for the weighted estimator (See in Theorem 3).
> 2. Empirical heuristics (e.g., greedy, Fisher) exist, but no existing method provides theoretical guarantees for any data-dependent weight vector.
> 3. PAC-Bayesian analysis studies voting-based randomized ensembles, where a distribution $\rho$ over base classifiers defines either a Gibbs classifier or a $\rho$-weighted majority vote [3, 4].  PAC-Bayesian ensembles fundamentally rely on probability distributions rather than deterministic weights, and therefore they cannot choose optimal weights, nor can they establish asymptotic optimality.
>
> [1] Van der Laan M J, Polley E C, Hubbard A E. Super Learner[J]. *Statistical Applications in Genetics and Molecular Biology*, 2007, 6(1).
>
> [2] Chen X, Klusowski J M, Tan Y S. Error reduction from stacked regressions[EB/OL]. arXiv:2309.09880, 2023.
>
> [3] Ortega L A, Cabañas R, Masegosa A. Diversity and generalization in neural network ensembles[C]//International Conference on Artificial Intelligence and Statistics (AISTATS). PMLR, 2022.
>
> [4] Masegosa A R, Lorenzen S S, Igel C, Seldin Y. Second order PAC-Bayesian bounds for the weighted majority vote[C]//Advances in Neural Information Processing Systems (NeurIPS). 2020.
>
> **(W1-2)** We thank the reviewer for the suggestion to compare our work with Masegosa (NeurIPS 2020) [1].
> After carefully examining that paper, we would like to clarify that although both works use the term *misspecification*, they address fundamentally different problems, operate in entirely different theoretical frameworks, and produce outputs that are not comparable. Therefore, a direct comparison is neither meaningful nor technically appropriate.
>
> 1. Different problem settings and goals
>
> - Masegosa (2020) studies Bayesian model averaging under distributional misspecification, and develops second-order PAC-Bayes bounds to construct alternative Bayesian-like posteriors.
> - Our work studies deep ensemble learning, a non-Bayesian paradigm, and derive the optimal weight for heterogeneous neural networks under misspecification (missing variables, limited capacity, non-smooth true functions).
>
> Thus, the objectives and learning paradigms of the two papers do not overlap.
>
> 2. Different definitions of misspecification
>
> - Masegosa focuses on misspecification within a probabilistic family $p(x|\theta)$, where the true data-generating distribution lies outside the assumed parametric class.
> - Our paper first defines misspecification in deep learning architectures, including
>   (i) variable misspecification,  (ii) structural misspecification, and  (iii) inherent misspecification.
>
> These are functional/representational limitations of neural networks, not likelihood misspecification.
>
> 3. Incompatible methodologies and outputs
>
> - Masegosa’s method outputs a **posterior distribution** over models by minimizing a PAC-Bayes bound.
> - Our method outputs a **deterministic weight vector** for combining multiple neural networks via validation-risk minimization.
> - PAC-Bayes requires an explicit likelihood and prior, whereas deep ensemble do not rely on probabilistic modeling.
>
> Therefore, the algorithms operate on incompatible assumptions and cannot be applied to each other’s settings.
>
> [1] Masegosa A R, Lorenzen S S, Igel C, Seldin Y. Second order PAC-Bayesian bounds for the weighted majority vote[C]//Advances in Neural Information Processing Systems (NeurIPS). 2020.

---

> > ### Author Response · Authors · 2025-11-23
> >
> > **(W2)**  Thank you for the reviewer’s helpful comments. We address each point below.
> >
> > (i, iii) The expression in line 171 (formerly line 165) and line 238 (formerly line 229) has been revised.
> >
> > (ii) We may be misunderstanding the reviewer’s concern, but our understanding is as follows. In Definition 2, we explicitly define $\mathcal{H}$ as the function class corresponding to a given neural network architecture, and $h$ is already rigorously defined as an element of the function class  $\mathcal{H}$ . The example does not require an additional definition of $h$; it merely applies the previously defined concept of risk minimization over a given architecture class. The dependence on $n$ comes from the known approximation error rates of these architectures, not from any expectation over hypotheses.
> >
> > (iv) We would like to clarify that $\varepsilon$ has already been explicitly defined earlier in line 151 (formerly line 144) in Section 3.1 in the manuscript.
> >
> > **(W3)**
> >
> > (i) In our setting, when splitting the data, we assume that the validation sample size $n\_{\mathrm{val}}$ is of the same order as the total sample size $n$. Under this assumption, expressing the rate in terms of $n$ or $n\_{\mathrm{val}}$ leads to the same $O\_p(\cdot)$ behavior. This point is stated and used in the proof in the Appendix (line 738, formerly line 710), where we clarify that the two quantities are not treated as identical but are assumed to be of the same order, so the asymptotic rate remains unchanged.
> >
> > (ii) I am sorry if our wording was confusing. Our intention is to illustrate that one classical source of model misspecification is the inclusion of irrelevant variables, as documented in [1] and [2].
> >
> > (iii) As pointed out by [3] and [4], the neural networks are always assumed to be well-specified.  Sufficiently large neural networks have the ability to approximate any continuous function, which in principle allows the approximation error $\|f\_0-f^*\|$ to approach zero.
> >
> > (iv) As stated in our earlier response, the references mentioned by the reviewer do **not** provide a constructive set of weights that is guaranteed to achieve oracle prediction. PAC-bayesian studies do not specify determined optimal weights, nor do they establish an oracle property for weighted deep ensemble in the sense considered in our paper.
> >
> > **(W4)** About Inherent misspecification. A simple clarification is that universal approximation theorems apply only to continuous functions on compact domains. In practice, many commonly encountered target functions do not satisfy these assumptions and therefore cannot be approximated arbitrarily well by any fixed neural network architecture. Examples include:
> > (i) discrete or combinatorial functions, such as the parity function or other combinatorial decision rules; and
> > (ii) discontinuous functions, such as the Bayes optimal classifier in classification problems.  These cases represent genuine forms of inherent misspecification.
> >
> > **(W5)**
> >
> > (i) Thanks for pointing out, and we have changed this typo.
> >
> > (ii) Thank you for pointing this out.
> > The references we originally included also relate to ensemble methods and were intended to provide context for the discussion. However, the two works suggested by the reviewer are indeed closely related and highly relevant to deep ensembles. We have therefore added both references to the revised manuscript, and we appreciate the reviewer for bringing them to our attention.
> >
> > (iii) Thank you for the helpful suggestion.
> > In our work, all theoretical results are established in a non-parametric framework, where neural networks are treated as flexible function classes whose complexity grows with the sample size. Under this perspective, model misspecification refers to the discrepancy between the true regression/classification function and the best approximation within this non-parametric class, rather than differences across fixed-dimensional parametric models.
> >
> > (iv) Thank you for the question.
> > We used “without loss of generality” to indicate that the notation $\tilde f$ simply denotes the candidate model with the smallest misclassification rate among the $M$ candidates. Since the ordering of models is arbitrary, we can relabel the best-performing model as $\tilde f$ without affecting any generality of the argument or the statement of the theorem. The phrase was intended only to clarify this notational choice.
> >
> >
> > [1] Maasoumi E. How to live with misspecification if you must[J]. *Journal of Econometrics*, 1990, 44(1–2): 67–86.
> >
> > [2] Blackwell M, Olson M P. Reducing model misspecification and bias in the estimation of interactions[J]. *Political Analysis*, 2022, 30(4): 495–514.
> >
> > [3] Barron A R. Approximation and estimation bounds for artificial neural networks[J]. *Machine Learning*, 1994, 14(1): 115–133.
> >
> > [4] Elbrächter D, Perekrestenko D, Grohs P, et al. Deep neural network approximation theory[J]. *IEEE Transactions on Information Theory*, 2021, 67(5): 2581–2623.

---

### Official Review · Reviewer_uRvF · 2025-10-29

**Soundness:** 3
**Presentation:** 3
**Contribution:** 2
**Rating:** 4
**Confidence:** 3

**Summary:**

The paper addresses a challenge in deep ensemble learning with model misspecification, where the universal approximation does not hold, and proposes an optimal weighted ensemble approach. Typical deep ensemble suffers from collective blindness as they reinforce shared biases, while the proposed weighted deep ensemble strategy can achieve oracle-level optimality. Comprehensive theoretical analysis shows asymptotic bounds for the estimator for regression and classification compared to the best candidate model and convergence under misspecifications. Experiments on synthetic tasks show improvement under various misspecification scenarios.

**Strengths:**

1. Interesting problem formulation: Systematic categorization of misspecification in deep learning with rigorous definitions.
2. Rigorous theoretical guarantees: Provides a formal analysis of the weighted deep ensembles with an asymptotic error bound matching the best candidate, and oracle optimality $R(\hat{w})/\inf_w R(w) \rightarrow 1$. Proofs are technically sound and leverage modern empirical process theory.
3. Comprehensive numerical validation: Experiments span all three misspecification types with nicely designed ablations.

**Weaknesses:**

1. The proposed algorithm is not new. The idea of of weighted ensemble with learned simplex weights by validation risk minimization has been explored with similar theoretical guarantees, just not on neural networks.
2. The theory only proves a "no-regret" sense of guarantee--the weighted ensemble performs at least as good as the best expert asymptotically. But the author did not investigate the ensemble gain under the weighted ensemble. This is especially important in the misspecification scenarios defined in the paper, where all models suffer from one or more sources of misspecification and are imperfect. The paper did not discuss how the weighted scheme affects the diversity or variance reduction that brings the ensemble gain under equal weight averaging. Even under misspecifications, candidate models may still have uncorrelated errors, which also explains the observed improvement in the numerical experiments.
3. Experiments are only done on synthetic datasets with shallow networks, which is good for demonstrating how different ensemble strategies perform under various misspecifications. It would be great to see how the algorithm works on standard small-scale vision benchmarks like CIFAR-10 or 100, or Tiny-ImageNet.

**Questions:**

1. This paper is primarily motivated by this notion of collective blindness, but it was not discussed later in the analysis. Can we somehow formalize the collective blindness of equal-weight ensembles through some sort of error decomposition and get a quantitative improvement bound for WDE compared with equal-weight ensembles?
2. In the proof of Theorem 1, the author leverages a Rademacher complexity term for the simplex with respect to both $M$ and $n$, $r\sqrt{\frac{2\log M}{n}}$. $M$ is omitted in the big-O as the number of ensemble members is finite, and the resulting bound becomes $\frac{r}{\sqrt{n}}$. But in practice, the number of ensemble members should somehow scale as a function of $n$, and you can only safely omit it if $M$ grows sub-polynomially with $n$. The author should clarify this somewhere in the paper, as you are deriving asymptotic bounds with $n$ while treating $M$ fixed.
3. The paper asserted the convexity of the VRM objective in classification because the ensemble is an affine mapping of logs. But this is only true for post-softmax probabilities. If the ensemble averages the logits instead (common practice in ensemble learning), would this break the convexity?

---

> ### Author Response · Authors · 2025-11-23
>
> Dear Reviewer uRvF,
>
> We are grateful to the reviewers for their thorough evaluation and insightful remarks. Their comments have helped us clarify important aspects of the manuscript and guided us toward several useful revisions. We address their concerns in detail below.
>
> **(W1)** We appreciate the reviewer’s observation, and we fully agree that the idea of learning simplex-constrained ensemble weights by minimizing validation risk is not new. Classical theoretical works on stacking (e.g., [1, 2]) have established lower bounds for such procedures. Our contribution does not lie in proposing a new algorithmic template, but rather in developing the first theoretical guarantees for weighted deep ensemble in settings that are substantially more challenging than those considered in prior work. Specifically:
>
> 1. **Extending validation-based weighting to deep neural networks is technically nontrivial.**
>
>  For neural networks, the approximation error and the stochastic error interact in more intricate ways due to their highly expressive function class, and the resulting convergence rates are generally slower than $n^{-1/2}$. As a consequence, the conditions required to obtain theoretical guarantees differ substantially from those in linear settings. Nevertheless, our analysis demonstrates that the classical validation–minimization principle continues to yield meaningful guarantees even in this more complex regime.
>
> 2. **We study weighted ensembles under model misspecification, which is essential in practice.**
>  Real-world data rarely aligns with the inductive biases of any fixed architecture.  Our results explicitly characterize how weighted deep ensemble behave when all base models are misspecified.
>
> 3. **We establish asymptotic optimality (Theorem 3).**
>  We prove that the weighted deep ensemble can achieve oracle-level predictive accuracy using only observable data. This result has not been shown before in weighted deep ensemble. We further validate the theoretical claims empirically in Figure 2.
>
> [1] van der Laan M J. Super Learner[J]. *Statistical Applications in Genetics and Molecular Biology*, 2007, 6(1).
>
> [2] Chen X, Klusowski J M, Tan Y S. Error reduction from stacked regressions[C]//2023 IMS International Conference on Statistics and Data Science (ICSDS). 2023: 141.
>
> **(W2)** Thank you for raising this important point. Classical ensemble analysis decomposes the risk into a bias term, a variance term, and a diversity term. Equal-weight deep ensemble implicitly adopt a fixed weighting strategy that attempts to reduce variance by exploiting diversity through uncorrelated errors.  Under model misspecification, however, some networks may exhibit substantially larger bias or variance than others. Because equal averaging assigns a fixed positive weight to every model, the influence of a poor candidate never vanishes and can significantly deteriorate the overall ensemble performance. In other words, equal weighting ignores the heterogeneity in bias and variance across models, which becomes particularly harmful in misspecified settings.
>
> In contrast, our estimator chooses the optimal weight vector within the admissible simplex by directly minimizing the overall predictive risk. The risk objective automatically balances bias, variance, and diversity, so the method does not ignore diversity; rather, it performs a data-driven allocation that retains models whose bias and variance are well controlled and whose predictions contribute useful diversity. Theorem 3 establishes that our estimator is asymptotically optimal: its risk converges to the minimal achievable risk. A direct implication is that the weight assigned to a poor model will converge to zero, providing a level of variance and error control that equal averaging cannot guarantee.
>
> It is also worth emphasizing that the equal-weight vector is included in our weight space. If equal weighting happens to be optimal in a particular scenario, our method will converge to the same risk. However, in the far more common case where equal weighting is suboptimal, our estimator converges to the truly optimal weight vector, while equal averaging offers no such theoretical guarantee.

---

> > ### Author Response · Authors · 2025-11-23
> >
> > **(W3)**  We thank the reviewer for the helpful comment. To complement the controlled settings in the main text, we include experiments on the real-world CIFAR-10  and CIFAR-10-C dataset. We report the overall performance, while the detailed results are presented in **Appendix F**. Our methods attain the strongest overall performance in this benchmark.
> >
> > 1. Accuracy
> >
> > |Dataset|DE(preactresnet18)|DE(resnet18)|DE(stochasticdepth18)|DE(googlenet)|DE(mobilenetv2)|EW|WDE|
> > |-|-|-|-|-|-|-|-|
> > |CIFAR10|91.18%|91.10%|91.20%|90.99%|87.81%|92.48%|**92.76%**|
> > |CIFAR10-C|22.60%|21.46%|19.88%|21.97%|18.73%|23.88%|**25.59%**|
> >
> >
> > 2. Expected Calibration Error (ECE), which measures how well model confidence aligns with its actual accuracy, with lower values indicating better calibration.
> >
> > |Dataset|DE(preactresnet18)|DE(resnet18)|DE(stochasticdepth18)|DE(googlenet)|DE(mobilenetv2)|EW|WDE|
> > |-|-|-|-|-|-|-|-|
> > |CIFAR10|0.0599|0.0564|0.0599|0.0583|0.0768|0.0507|**0.0438**|
> > |CIFAR10-C|0.5889|0.6468|0.5889|0.5614|0.5735|0.6128|**0.4884**|
> >
> >
> > **(Q1)** The root cause of collective blindness lies in the fact that traditional deep ensemble often rely on models that share the same architecture and differ only in random initialization. So, the diversity among ensemble members is much lower than expected.
> >
> > To visualize this phenomenon, we plot a t-SNE embedding of CIFAR-10 predictions generated by  initializing the same architecture 10 times, see **Figure 3 in Appendix G**. The points corresponding to the same architecture cluster tightly together, revealing that different initializations lead to highly similar predictions. In contrast, incorporating different architectures brings  greater diversity, producing predictions that are far less correlated.
> >
> > To further quantify the effect of collective blindness, we introduce three metrics:
> >
> > For each model $m$, We define the error set on slightly hard samples to ensure that correlations are measured only on non-trivial mistakes: $E\_m = \lbrace n : |\hat{f}\_m(\mathbf{X}\_n) - Y\_n| > \tau \rbrace$, and for the ensemble, $E\_{\mathrm{ens}} = \lbrace n : |\hat{f}(\mathbf{X}\_n; \mathbf{w}) - Y\_n| > \tau \rbrace.$ To measure how two models make errors on the same samples, we define their error correlation $\mathrm{CoErr}(i,j)  = \frac{|E\_i \cap E\_j|}{|E\_i \cup E\_j|}, $ and for each model $m$, the correction with the ensemble is $\mathrm{CoErr}(\mathrm{ens}, m) =
> > \frac{|E\_{\mathrm{ens}} \cap E\_m|}{|E\_{\mathrm{ens}} \cup E\_m|}.$
> >
> > (i) Model–Model Error Correlation (mCo-Err).
> > The average error correction across all model pairs is
> > $$
> > \mathrm{mCoErr} = \frac{1}{M(M-1)} \sum\_{i \neq j} \mathrm{CoErr}(i,j),
> > $$
> >
> > which measures the typical correction of errors across individual models.
> >
> > (ii) Ensemble–Model Error Correlation (EnCo-Err).
> > The weighted ensemble–model correction is
> > $$
> > \mathrm{EnCoErr} = \sum\_{m=1}^{M} w\_m \, \mathrm{CoErr}(\mathrm{ens}, m),
> > $$
> >
> > which measures the correction between ensemble errors and those of its base models.
> >
> > (iii) Tail Error Rate (TER).
> > $$
> > \mathrm{TER}
> > = \frac{1}{N} \sum\_{n=1}^{N}
> > \mathbf{1}\!\left( |\hat{f}(\mathbf{X}\_n; \mathbf{w}) - Y\_n| > \tau \right),
> > $$
> > which measures the proportion of predictions that incur non-trivial errors.
> >
> > Here, we report the overall results: for variable misspecification, we present the average performance over misspecification levels {0,1,3,5,7} at a sample size of 5000; for structural misspecification, we report the mean over misspecification strengths from 0.1 to 1 in increments of 0.1. The complete set of detailed results is available in **Appendix G**.
> >
> > |Misspecification|Task|mCoErr(diverse)|mCoErr(same)|EnCoErr(DE)|EnCoErr(EW)|EnCoErr(WDE)|TER(DE)|TER(EW)|TER(WDE)|
> > |-|-|-|-|-|-|-|-|-|-|
> > |Variable|nested|0.7819|0.8268|0.8815|0.8352|0.6799|0.2503|0.2419|0.2351|
> > ||interaction|0.7485|0.8065|0.8352|0.8146|0.6452|0.1532|0.1522|0.1510|
> > ||nonlinear|0.7817|0.8261|0.8545|0.8277|0.6826|0.1216|0.1218|0.1202|
> > |Inherent|Infinite|0.3420|0.3775|0.6000|0.5804|0.3126|0.1304|0.1104|0.0037|
> > ||Square|0.6582|0.7105|0.9472|0.9313|0.6359|0.3034|0.2945|0.2466|
> > |structural|$\Delta=30000$|0.7090|0.7425|0.7978|0.7862|0.5034|0.1373|0.1021|0.0980|
> > ||$\Delta=50000$|0.7305|0.7683|0.8211|0.8138|0.5104|0.1510|0.1120|0.1080|
> > ||$\Delta=100000$|0.6886|0.7182|0.7822|0.7476|0.4733|0.1211|0.0900|0.0857|
> >
> > These metrics jointly indicate that (a) collective blindness is weaker in diverse architectures than in single-architecture settings. (b) weighted ensembles reduce error correlations with individual models, and (c) extreme prediction errors become less frequent. These results offer a quantitative assessment of collective blindness and demonstrate that WDE effectively mitigates it.

---

> > > ### Author Response · Authors · 2025-11-23
> > >
> > > **(Q2)** Thank you for pointing this out. Our main theoretical results treat the number of ensemble members $M$ as finite when $n$ grows. In fact, the theory can be extended to the regime where $M$ diverges with $n$. In that case, the Rademacher
> > > complexity term involving $M$ must be retained, and the corresponding conditions and conclusions require appropriate modifications. We have revised these formal statements in Theorem 1, where we provide the version of our results that allows $M$ to grow with $n$. Condition 2 has also been adjusted so that Theorem 3 remains valid under a diverging dimension of the candidate models.
> > >
> > > **(Q3)** Thank you for raising this point. Our theoretical guarantees for the oracle weight do not rely on the convexity of the VRM objective. Although convexity holds naturally when averaging post-softmax probabilities, our analysis does not require this property. In the proofs, we work with a continuous approximation of the risk function.  Therefore, the asymptotic optimality guarantees in Theorem 3  continue to hold.

---

> > > > ### Comment · Reviewer_uRvF · 2025-11-24
> > > >
> > > > The rebuttal is quite comprehensive and addresses most of my concerns. Therefore, I am raising my score to 6. Great work!
> > > > Just one minor issue, although the theory does not assume convexity, obtaining the optimal weight is then not so trivial. But consider that in practice $M$ is small, i.e., 10 in most experiments, the optimization problem should not pose too big a challenge.

---

> > > > > ### Author Response · Authors · 2025-11-24
> > > > >
> > > > > Thank you very much for your thoughtful comments and for raising the score. We truly appreciate it!
> > > > >
> > > > > Regarding the minor point you mentioned, we agree with your observation: averaging the logits leads to a non-convex problem (averaging the post-softmax probabilities, as we do in the paper, yields a convex formulation). In practice, however, M is small, as you said, so the optimization is not particularly challenging, and some methods such as projected gradient descent may handle it.
> > > > >
> > > > > Thank you again for your careful reading and constructive feedback!

---

### Official Review · Reviewer_iDai · 2025-11-01

**Soundness:** 2
**Presentation:** 2
**Contribution:** 3
**Rating:** 2
**Confidence:** 3

**Summary:**

This work conducts a theoretical study on Weighted Deep Ensembles, which assume unequal weighting coefficients across ensemble members. In this framework, the ensemble weights are learned through empirical risk minimization on a pre-held validation dataset.

**Strengths:**

- It appears to be well grounded in existing theoretical results for deep neural networks. In particular, Corollaries 1–3, which provide asymptotic error bounds for practical architectures such as MLPs, CNNs, and RNNs, are quite compelling. Of course, the practical usefulness of such theoretical results remains somewhat unclear, but that’s often the nature of theoretical work.

- From a quick look, the derivations seem sound, and the experimental design appears reasonably solid. I particularly like that Table 4 highlights an important comparison with In-sample and Greedy Ensembles, and Figure 2 nicely shows convergence toward the oracle weights.

**Weaknesses:**

- The method used in this work to determine the weighting coefficients for combining ensemble members’ predictions is a form of stacking (also known as stacked generalization, functional aggregation, and perhaps other related terms, as it has been referred to under various names in the literature). This approach has been extensively studied since the seminal works of Wolpert (1992) and Breiman (1996), with further theoretical developments by Van der Laan et al. (2007), Arsov et al. (2019), Chen et al. (2024), and others. However, this line of research is not discussed at all in the paper. The proposed weighted deep ensemble should explicitly position itself within the stacked generalization framework and clarify both the established findings in this area and its specific contributions in the context of deep neural networks.

- Corollaries 1–3 are presented in a somewhat simplified form in the main text, and although Appendix B.4 offers a slightly more detailed version, it still appears insufficient. It would be beneficial to include a fully formalized version in the appendix that explicitly incorporates the necessary conditions outlined in works such as Jiao et al. (2023). While those formulations, as far as I know, involve a number of intricate and cumbersome assumptions, this work, as a theoretical contribution building upon them, should nonetheless aim to achieve a comparable level of rigor and completeness.

- At present, there is neither empirical nor theoretical validation of the claimed “collective blindness.” The only supporting evidence is the conceptual illustration in Figure 1, which does not pertain to “deep” ensembles. While the authors claim that traditional deep ensembles may suffer from “collective blindness,” this assertion seems questionable given the experimental scale considered here, which can hardly be described as involving truly “deep” ensembles. In my experience, in synthetic setups with small MLPs, ensembles trained from different random initializations through stochastic optimization (i.e., standard recipe for constructing deep ensembles) often exhibit limited diversity, which aligns with the notion of “collective blindness.” However, as the network depth and complexity increase, the highly non-convex nature of the loss landscape tends to induce substantial diversity among ensemble members, and even simple deep ensembles can perform remarkably well, as demonstrated by Fort et al. (2019). Hence, it becomes difficult to argue that “collective blindness” remains a meaningful concern in deep ensemble settings.

- The experimental results seem rather limited to be considered a proper evaluation of a weighted “deep” ensemble. Given the computational constraints, it might be a good idea to extend the results of Wortsman et al. (2022) as a way to demonstrate larger-scale experiments. Their official codebase already provides checkpoints that can be directly used as ensemble components, so there is no need for additional training. Since they have already considered Uniform and Greedy Ensembles, it would suffice to simply add the Weighted Ensemble for comparison.

---
- Wolpert (1992), Stacked generalization.
- Breiman (1996), Stacked regressions.
- Van der Laan et al. (2007), Super learner.
- Arsov et al. (2019), Stacking and stability.
- Chen et al. (2024), Error reduction from stacked regressions.
- Fort et al. (2019), Deep ensembles: a loss landscape perspective.
- Wortsman et al. (2022), Model soups: averaging weights of multiple fine-tuned models improves accuracy without increasing inference time.

**Questions:**

- Incomplete statement on line 164?
- How were the oracle weights in Figure 2 obtained?
- If the validation split (2 out of the 6:2:2 split) is used to “train” the weighting coefficients in WDE, then it is effectively being used as part of the ensemble “training” process. What if a standard deep ensemble is trained using the combined training and validation splits, since that data could alternatively be used to train the ensemble members rather than the ensemble weights?

---

> ### Author Response · Authors · 2025-11-23
>
> Dear Reviewer iDai,
>
> We appreciate the reviewers’ careful reading of our manuscript and their constructive criticisms. Their detailed feedback has been instrumental in improving both the technical precision and the exposition of our work. Our responses to each comment are presented below.
>
> **(W1) Stacking vs Ours**
>
> 1. **Extending validation-based weighting to deep neural networks is technically nontrivial.**
>  Existing stacking theoretical results primarily address linear models, where the parametric structure is transparent and parameter estimates typically achieve the root-$n$ convergence rate. In contrast, the main challenge for deep neural networks arises from the way their errors decompose. For neural networks, the approximation error and the stochastic error interact in more intricate ways due to their highly expressive function class, and the resulting convergence rates are generally slower than $n^{-1/2}$. As a consequence, the conditions required to obtain theoretical guarantees differ substantially from those in linear settings. Nevertheless, our analysis demonstrates that the classical validation–minimization principle continues to yield meaningful guarantees even in this more complex regime.
> 2. **We study weighted ensembles under model misspecification, which is essential in practice.**
>  Real-world data rarely aligns with the inductive biases of any fixed architecture.  Our results explicitly characterize how weighted deep ensemble behave when all base models are misspecified.
> 3. **We establish asymptotic optimality (Theorem 3).**
>  We prove that the weighted deep ensemble can achieve oracle-level predictive accuracy using only observable data. This result has not been shown before in weighted deep ensemble. We further validate the theoretical claims empirically in Figure 2.
>
> **(W2)** Thank you for this helpful suggestion. In response, we have added fully formalized statements of Corollaries 1, 2, and 3 in **Appendix B.4**. The main text presents these results in a simplified form because Theorem 1 is stated in a general framework, and listing the specific assumptions for particular network architectures would considerably reduce readability. Following your advice, the appendix now explicitly records the conditions required for the network settings considered in our experiments, in line with those in [1] and related work. We believe this revision makes the theoretical contribution more rigorous and complete while keeping the main exposition accessible.
>
> [1] Jiao Y, Shen G, Lin Y, et al. Deep nonparametric regression on approximate manifolds: Nonasymptotic error bounds with polynomial prefactors[J]. *The Annals of Statistics*, 2023, 51(2): 691–716.
>
> (W3)  Thank you for raising this important point. Our empirical findings show that collective blindness always persists.
>
> To illustrate this, we provide an empirical visualization using four widely used deep architectures:
> (i) ResNet152, (ii) PreActResNet152, (iii) ResNext152, (iv) Seresnet152,
>
> Specifically, we plot a t-SNE embedding of CIFAR-10 predictions generated by initializing the same architecture 10 times, see **Figure 3 in Appendix G**. The points corresponding to the same architecture cluster tightly together, revealing that different initializations lead to highly similar predictions. In contrast, incorporating different architectures brings  greater diversity, producing predictions that are far less correlated.
>
> **(W4).** Thank you for the suggestion. We fully agree that Model Soup (Wortsman et al., 2022) provides a convenient way to perform large-scale ensemble experiments without additional training. However, we would like to clarify that the setting considered in Model Soup is fundamentally different from ours. Model Soup performs ensembling in weight space by averaging model parameters, which in practice can only be done across models that share the same architecture. In contrast, our method performs ensembling in output space by combining predictions with learned weights. As discussed in our paper, ensembles formed from models with the same architecture tend to exhibit limited diversity, so the gains of their greedy soup method over uniform averaging are relatively modest.  Importantly, despite these limitations, our weighted ensemble method still achieves the best performance under the same experimental conditions.
>
> |Datasets|UniformSoup|GreedySoup|WDE|
> |-|-|-|-|
> |ImageNet|0.78638|0.78654|**0.79794**|
> |ImageNetV2|0.66860|0.66860|**0.68740**|
> |ImageSketch|0.43903|0.43901|**0.45096**|
> |ImageNetR|0.62403|0.62423|**0.63513**|
> |ImageNetA|0.23880|0.23853|**0.24653**|

---

> > ### Comment · Reviewer_iDai · 2025-11-24
> >
> > > (W3)  Thank you for raising this important point. Our empirical findings show that collective blindness always persists.
> > >
> > > To illustrate this, we provide an empirical visualization using four widely used deep architectures: (i) ResNet152, (ii) PreActResNet152, (iii) ResNext152, (iv) Seresnet152,
> > >
> > > Specifically, we plot a t-SNE embedding of CIFAR-10 predictions generated by initializing the same architecture 10 times, see Figure 3 in Appendix G. The points corresponding to the same architecture cluster tightly together, revealing that different initializations lead to highly similar predictions. In contrast, incorporating different architectures brings greater diversity, producing predictions that are far less correlated.
> >
> > The t-SNE visualization plots 10 points (for each model architecture), where each point represents one model initialization. Is the high-dimensional input for each model point the 100,000-dimensional vector formed by flattening the 10000*10 categorical predictions (softmax probabilities) made by that specific model on the entire test set?

---

> ### Author Response · Authors · 2025-11-23
>
> **(Q1)** Thank you for your helpful comment.  The expression has been revised.
>
> **(Q2)** We thank the reviewer for raising this point. The oracle weights in Figure 2 are constructed by minimizing the empirical loss on the test dataset. Specifically, we apply the same ensemble-weight optimization but evaluate the objective on the test samples. Since this uses information unavailable in practice, the resulting weights represent an ideal reference level of performance.
>
> **(Q3)** We thank the reviewer for the thoughtful question. In our implementation, the validation split in the 6:2:2 partition is already used for early stopping and for selecting hyperparameters of the individual models.  Moreover, using an additional validation set to learn the ensemble weights follows standard practice in the ensemble-learning literature. Methods such as stacking, super learner, and many meta-learning ensemble approaches explicitly train the meta-learner (or the weighting coefficients) on a separate validation set distinct from the data used to train the base models.

---

> ### Comment · Reviewer_iDai · 2025-11-24
>
> > (W4). Thank you for the suggestion. We fully agree that Model Soup (Wortsman et al., 2022) provides a convenient way to perform large-scale ensemble experiments without additional training. However, we would like to clarify that the setting considered in Model Soup is fundamentally different from ours. Model Soup performs ensembling in weight space by averaging model parameters, which in practice can only be done across models that share the same architecture. In contrast, our method performs ensembling in output space by combining predictions with learned weights. As discussed in our paper, ensembles formed from models with the same architecture tend to exhibit limited diversity, so the gains of their greedy soup method over uniform averaging are relatively modest.  Importantly, despite these limitations, our weighted ensemble method still achieves the best performance under the same experimental conditions.
>
> It appears the authors may have misunderstood my original comment. I was not requesting a comparison against the "Soup" approaches itself. Instead, my intention was to utilize the pre-trained component models that formed their UniformSoup, GreedySoup, and GreedyEnsemble results, and compare those components in an ensemble format, as previously suggested: "Since they have already considered Uniform and Greedy Ensembles, it would suffice to simply add the Weighted Ensemble for comparison."
>
> Please note that "Greedy ensemble" and "Greedy soup" are distinct concepts in Wortsman et al. (2022). My request specifically relates to the ensembling of component predictions (ensemble), not the evaluation of the weight-averaged result (soup). I mention their codebase only because it appears the authors can leverage the models they have already trained and provided, avoiding any additional training cost, to fulfill this comparison.

---

> > ### Author Response · Authors · 2025-11-24
> >
> > **(Q1).** Yes, your understanding is correct. In Fort et al. (2019) [1], the authors’ released code performs the t-SNE visualization in exactly this way. Specifically, as described in Section 4.1 [1]:
> >
> > > “More precisely, for each checkpoint we take the softmax output for a set of examples, flatten the vector, and use it to represent the model’s predictions.”
> >
> > Following this, we adopt the same procedure in our implementation.
> >
> > **(Q2).** Thank you very much for the clarification. We now better understand the source of the confusion.
> >
> > To be explicit: the original Model Soup methods operate in the weight space, where the model parameters are averaged. However, because weight-space averaging is only applicable to models that share identical architectures, this setting does not align with the main focus of our paper.
> >
> > To conduct the exact experiment you requested, we therefore converted the original setting in Model Soup, into an output-space ensemble setting. Specifically, we took the pre-trained component models released by Wortsman et al. (2022), kept their weights fixed, and applied our weighted output-space ensembling directly on their predictions, without any additional training. The table in our rebuttal corresponds precisely to these output-space ensembles.
> >
> > In other words, we were not comparing against the original Model Soup algorithm, but instead ensembling the same component models that were used to form UniformSoup and GreedySoup, exactly in the way you described in your initial comment. This matches your intended comparison.
> >
> > We also note that all provided models are different random initializations of the same architecture (CLIP ViT-B/32), which corresponds to the standard deep-ensemble scenario. Under this homogeneous-architecture setting, our weighted ensembling achieves clear improvements.
> >
> > We hope this revision clearly aligns our experimental setting with your requested comparison. Please let us know if any additional clarification would be helpful.
> >
> > [1] Fort S, Hu H, Lakshminarayanan B. Deep ensembles: a loss landscape perspective. arXiv:1912.02757, 2019.
> >
> > [2] Wortsman M, Ilharco G, Gadre S Y, et al. Model soups: averaging weights of multiple fine-tuned models improves accuracy without increasing inference time[C]//International Conference on Machine Learning (ICML). PMLR, 2022.

---

> ### Comment · Reviewer_iDai · 2025-11-24
>
> > To be explicit: the original Model Soup methods operate in the weight space, where the model parameters are averaged. However, because weight-space averaging is only applicable to models that share identical architectures, this setting does not align with the main focus of our paper.
>
> I already know that the model soup algorithm is only applicable in a homogeneous setup.
>
> > To conduct the exact experiment you requested, we therefore converted the original setting in Model Soup, into an output-space ensemble setting. Specifically, we took the pre-trained component models released by Wortsman et al. (2022), kept their weights fixed, and applied our weighted output-space ensembling directly on their predictions, without any additional training. The table in our rebuttal corresponds precisely to these output-space ensembles.
>
> I already know the reported WDE results construct an unequally weighted ensemble based on the method proposed by the authors for the ensemble members. However, the necessary comparison should be against the “Uniform  and Greedy Ensembles,” not the “Uniform and Greedy Soups.” (I also note that those results seem anomalous; the negligible performance difference between Uniform Soup and Greedy Soup is highly suspicious and suggests an underlying experimental error)
>
> ---
>
> Again, I specifically stated that the comparison must be made against the “Uniform and Greedy Ensembles,” not “Soups,” cf. _‘Since they have already considered Uniform and Greedy Ensembles, it would suffice to simply add the Weighted Ensemble for comparison.’_ The term “Greedy Ensemble” refers to the identical concept in both this work (cf. line 475) and Wortsmanet al. (2022).
>
> As a result, all we need is to compare the proposed WDE to “Ensemble (=Uniform ensemble)” and “Greedy ensemble (= greedy ensemble)” results provided in Table 3 of Wortsman et al. (2022). In other words, the comparison I intended extends Table 3 of Wortsman et al. (2022) by focusing exclusively on output-space ensemble techniques. This results in a structure nearly identical to the authors' comparative tables: 'DE=Ensemble' versus 'WDE'
>
> | | ImageNet | Dist. shifts |
> | :- | :-: | :-: |
> | Best individual model | 80.38 | 47.83 |
> | Second best model | 79.89 | 43.87 |
> ||
> | excluding soup results | - | - |
> ||
> | Ensemble | 81.19 | 50.77 |
> | Greedy ensemble | 81.90 | 49.44 |
> | WDE | ? | ? |
>
> ---
>
> ++ In summary, this means the experiment effectively becomes a large-scale comparison of WDE versus Greedy Ensemble (as seen in Table 4), utilizing the checkpoints provided by Wortsman et al. (2022).

---

> > ### Author Response · Authors · 2025-11-28
> >
> > Thank you very much for your clarification and suggestions. We sincerely apologize for previously misunderstanding your intent.
> >
> > Initially, we did not notice the ensemble branch in the original Model Soup codebase. As a result, we directly modified the main function and evaluated the results using a subset of test images. This led to slight deviations from the results reported in Wortman et al. (2022), and consequently the baseline ensemble numbers we originally reported appeared lower than those presented in the original paper. We apologize for the confusion this caused.
> >
> > We have now rerun the experiments strictly following the ensemble branch in the official codebase and evaluated using the full test set. One note is that, due to current time constraints, we do not yet have access to ObjectNet. If necessary, we will download it and report the full averaged shift accuracy in the revised version. For now, we report the performance on  the averaged shift accuracy without ObjectNet.
> >
> > To ensure correctness and facilitate reproducibility, we have made the updated implementation and outputs available at:
> > https://anonymous.4open.science/r/soup_WDE-8518/
> >
> > Reviewers may either run our modified main function within the Model Soup framework or directly inspect our generated results in the repository.
> >
> > Below are the updated evaluation results obtained under the correct experimental configuration:
> >
> >
> > |Method|ImageNet|Dist.shifts$^{++}$|
> > |-|-|-|
> > |Best individual model|80.39|49.31|
> > |Second best model|79.89|45.50|
> > |excluding soup results|-|-|
> > |Ensemble|81.19|52.14|
> > |Greedy ensemble|81.90|50.19|
> > |WDE|**81.96**|**52.39**|
> > |++ Without ObjectNet|||
> >
> >
> > As shown in Table, under the fully aligned experimental setting, our weighted ensemble remains superior to both the uniform ensemble and the greedy ensemble, which supports the necessity of learning optimal weights rather than relying on equal averaging. We also clarify that although using the same network architecture across ensemble members yields only limited gains, our weighted approach still achieves the best performance.  Moreover, WDE benefits further from incorporating diverse network architectures.
> >
> > Once again, thank you for bringing this issue to our attention. Your comments significantly helped us strengthen the evaluation and ensure consistency with prior work.

---

> > > ### Comment · Reviewer_iDai · 2025-11-28
> > >
> > > The timing was perfect, allowing me to review your response immediately. It is excellent to see these results! You are absolutely right that ObjectNet is quite a cumbersome benchmark to evaluate, which often leads to its exclusion in many studies (and frankly, I don't think you need to prioritize it right away; you could handle it later, or even choose to omit it entirely). Thanks again to the authors for their hard work.
> > >
> > > That said, given the current scope, this verification using the setup of Wortsman et al. (2022) can certainly be considered sufficient for a practical and large-scale "deep" ensemble scenario, as was initially intended. Furthermore, using the terminology of Wortsman et al. (2022), this would be considered a "learned ensemble" (they termed the case where weighting coefficients are learned within the soup as a "learned soup," cf. Appendix I). It is quite encouraging that the WDE in this setting manages to retain the superior in-distribution performance of the greedy ensemble while simultaneously securing robust performance of the (uniform) ensemble against distribution shift.

---

> > > > ### Author Response · Authors · 2025-11-28
> > > >
> > > > Thank you very much for your timely and encouraging feedback! We truly appreciate your thoughtful comments and the careful consideration you have given to our work.
> > > >
> > > > We have updated the related work section accordingly in the  manuscript. In light of these responses, we hope we have addressed your concerns, and we hope you will consider raising your score.
> > > >
> > > > Thank you again for your valuable time and constructive insights.

---

> ### Comment · Reviewer_iDai · 2025-11-28
>
> We are currently in the final week of the rebuttal period, and as such, I am submitting my response at this time. I have changed my score to 4 (++the editing function seems to be broken at the moment; I am waiting for it to be fixed).
>
> I first wish to thank the authors for their time and effort in addressing my concerns. Regarding the comparative experiment between their method and the Greedy ensemble (W4), specifically within the practical "deep" ensemble scenario utilizing the setup from Wortsman et al. (2022), there appears to have been some communication issue. Nevertheless, the authors' newly provided minimal experimental verification of collective blindness (W2) and the revisions to Appendix B.4 (W3) are valuable additions, and I acknowledge their merit.
>
> However, even setting aside W4 for the moment, W1 remains a persistent concern for me; upon reviewing the other reviewers' feedback, I note that Reviewer Afi7 expresses similar concerns. Currently, the main text limits its discussion of prior work on weighted ensembles to only the final sentence in the related work section. I believe the text requires a major revision to adopt a well-structured form that properly credits existing works while clearly articulating this work's contribution to the field. While reviewing the authors' response, it appears they possess a certain roadmap or blueprint for addressing these issues. However, since the information provided is currently fragmented, it is difficult to make a final judgment just now.

---

> > ### Author Response · Authors · 2025-11-28
> >
> > Thank you very much for your thoughtful follow-up and for updating your score. We truly appreciate the time you devoted to reviewing our manuscript and the clarity of your concerns.
> >
> > Regarding W4, sorry for the delay, but we have now fully supplemented the requested experiments following your guidance. We hope that these additional results help resolve the concerns you previously raised.
> >
> > For W1, we have already expanded our explanation in the response. To present the contribution and prior work more clearly, we are preparing a more polished and better-organized revision of this section. We will upload an updated version shortly to ensure that the narrative is coherent and that existing works are properly compared.
> >
> > Thank you again for your constructive feedback; it has been very helpful for strengthening both the technical content and the presentation of our work.

---

### Official Review · Reviewer_DdZD · 2025-11-01

**Soundness:** 3
**Presentation:** 3
**Contribution:** 3
**Rating:** 6
**Confidence:** 3

**Summary:**

This paper first defines three different kinds of model misspecifications, which if they occur in particular lead to traditional guarantees like universal approximation theorems for deep neural networks not holding. They furthermore argue that traditional ensembles, made up of models with identical architectures and each weighted equally, are also impacted by this issue since if every submodel is biased, this will usually lead to highly confident, systematically biased predictions of the ensemble. To address this issue, they propose and analyze the weighted deep ensemble method, which trains ensembles consisting of different models and optimizes the weights of the different ensemble members to optimize the error on the validation set. They prove that the ensemble achieves the convergence rate of the best model in the ensemble and empirically demonstrate the effectiveness of the method on synthetic datasets.

**Strengths:**

- The story of the paper was relatively clear and the paper was well-organized.
- By investigating the question when the assumptions of traditional machine learning approximation results fail to hold, the paper is making progress and bringing more attention to a very relevant question.
- Furthermore, by providing the weighted ensemble method, they also provide a new way of addressing the issues they point out. The analyses of the theoretical properties (i.e., showing both asymptotic error bounds and asymptotic optimality in certain cases) of the weighted ensemble method is original and insightful.

**Weaknesses:**

The paper claims that they are 'introduc[ing] [the] weighted deep ensemble method that learns the optimal weights'. At the same time, in the related work section, they state that 'recent studies have delved into weighted deep ensemble', but do not provide much more detail about these methods although this would be relevant to judge what exactly is novel in the paper.
Furthermore, it would be relevant and interesting to also see the performance of their method on non-synthetic datasets (and with models trained on these tasks) to investigate whether they can also provide significant advantages in real-world settings and potentially on more not misspecified settings.

**Questions:**

1. Could you clarify what kind of work has been done on weighted ensembles before? What are the key differences of your work from previous work on this topic?
2. In line 165, is $f_0(x)-g(\pi(\boldsymbol{X}))$ supposed to be $0$ almost surely?
3. Could you make the following statement more formal or illustrate it a bit more clearly:
> As stated before, traditional deep ensembles may suffer from "collective blindness" in the presence of variable, structural, or inherent misspecification.
4. We use our validation data to fit the weights of the ensemble, correct? Would this in the case of many different ensemble members lead to overfitting on the validation data? Could we then still use the same validation data for hyperparameter tuning, etc.?
5. Why do we need Conditions 1 and 2 for Theorem 3? What would lead to the Theorem not holding anymore if we relax these conditions?
6. Why did you decide not to additionally test your methods on real-world datasets or mode widely on non-misspecified settings?

---

> ### Author Response · Authors · 2025-11-23
>
> Dear Reviewer DdZD:
>
> We are grateful to the reviewers for their insightful comments and constructive feedback, which have significantly helped us improve the clarity and comprehensiveness of our work. We have addressed the raised points as follows:
>
> **(W1, Q1)  Previous Work.**
>
> |Method| Misspecification|Generalization Bound|Deterministic Optimal Weights|Asymptotic Oracle Optimality|
> |-|-|-|-|-|
> |Stacking [1][2]|✗|✓|✓|✗|
> |Diversity-based [3][4][5]|✗| $\circ $ (PAC-Bayesian)|✗(uniform)|✗|
> |**Ours**|✓|✓|✓|✓|
>
> 1. Traditional stacking methods combine multiple predictors through linear aggregation, but their theoretical guarantees have not extended to modern deep neural networks. Moreover, they do not analyze ensembles under misspecification, nor do they provide asymptotic optimality guarantees for the weighted estimator, as shown in **Theorem 3**.
>
> 2. The diversity literature establishes that, for uniformly averaged ensembles, showing that the ensemble risk is never worse than the uniform average of the individual risks, with improvement attributed to prediction diversity among models, but never compares to the best individual model [1] [2]. These results apply only to uniform averaging rather than weighted combinations.
>
> 3. Empirical heuristics (e.g., greedy, Fisher) exist, but no existing method provides theoretical guarantees for any data-dependent weight vector.
> 4. One of ensemble work on diversity, e.g., [5], derives the generalization bound through PAC-Bayesian, a voting-based randomized ensembles, where a distribution $\rho$ over base classifiers defines either a Gibbs classifier  or a $\rho$-weighted majority vote [6].  PAC-Bayesian ensembles fundamentally rely on probability distributions rather than deterministic weights, and therefore they cannot choose optimal weights, nor can they establish asymptotic optimality.
>
> Our contribution: (1) We are the first to consider weighted deep ensemble under misspecification. (2) We provide the theoretical guarantee that weighted deep ensemble can attain oracle-level accuracy using only observable data, and show that an asymptotic error bound for the weighted deep ensemble estimator converges at the same rate as the smallest error bound among all candidate networks.
>
> [1] Van der Laan M J, Polley E C, Hubbard A E. Super Learner[J]. *Statistical Applications in Genetics and Molecular Biology*, 2007, 6(1).
>
> [2] Chen X, Klusowski J M, Tan Y S. Error reduction from stacked regressions[EB/OL]. arXiv:2309.09880, 2023.
>
> [3] Wood D, Liu C, Zhuang S, et al. A unified theory of diversity in ensemble learning[J]. *Journal of Machine Learning Research*, 2023, 24(359): 1–49.
>
> [4] Abe T, Gong R, Sun Y, et al. Pathologies of predictive diversity in deep ensembles[C]//International Conference on Learning Representations (ICLR). 2024.
>
> [5] Ortega L A, Cabañas R, Masegosa A. Diversity and generalization in neural network ensembles[C]//International Conference on Artificial Intelligence and Statistics (AISTATS). PMLR, 2022.
>
> [6] Masegosa A R, Lorenzen S S, Igel C, Seldin Y. Second order PAC-Bayesian bounds for the weighted majority vote[C]//Advances in Neural Information Processing Systems (NeurIPS). 2020.
>
> **(Q2).** Thank you for your helpful comment.  The expression in line 171 (formerly line 165) has been revised.

---

> ### Author Response · Authors · 2025-11-23
>
> **(Q3) Collective Blindness.**
>
> The root cause of collective blindness lies in the fact that traditional deep ensemble often rely on models that share the same architecture and differ only in random initialization. So, the diversity among ensemble members is much lower than expected.
>
> To visualize this phenomenon, we plot a t-SNE embedding of CIFAR-10 predictions generated by  initializing the same architecture 10 times, see **Figure 3 in Appendix G**. The points corresponding to the same architecture cluster tightly together, revealing that different initializations lead to highly similar predictions. In contrast, incorporating different architectures brings  greater diversity, producing predictions that are far less correlated.
>
> To further quantify the effect of collective blindness, we introduce three metrics:
>
> For each model $m$, We define the error set on slightly hard samples to ensure that correlations are measured only on non-trivial mistakes: $E\_m = \lbrace n : |\hat{f}\_m(\mathbf{X}\_n) - Y\_n| > \tau \rbrace$, and for the ensemble, $E\_{\text{ens}} = \lbrace n : |\hat{f}(\mathbf{X}\_n; \mathbf{w}) - Y\_n| > \tau \rbrace.$ To measure how two models make errors on the same samples, we define their error correlation $\mathrm{CoErr}(i,j)  = \frac{|E\_i \cap E\_j|}{|E\_i \cup E\_j|}, $ and for each model $m$, the correction with the ensemble is $\mathrm{CoErr}(\mathrm{ens}, m) =
> \frac{|E\_{\mathrm{ens}} \cap E\_m|}{|E\_{\mathrm{ens}} \cup E\_m|}.$
>
> (i) Model–Model Error Correlation (mCo-Err).
> The average error correction across all model pairs is
> $$
> \mathrm{mCoErr} = \frac{1}{M(M-1)} \sum\_{i \neq j} \mathrm{CoErr}(i,j),
> $$
>
> which measures the typical correction of errors across individual models.
>
> (ii) Ensemble–Model Error Correlation (EnCo-Err).
> The weighted ensemble–model correction is
> $$
> \mathrm{EnCoErr} = \sum\_{m=1}^{M} w\_m \, \mathrm{CoErr}(\mathrm{ens}, m),
> $$
>
> which measures the correction between ensemble errors and those of its base models.
>
> (iii) Tail Error Rate (TER).
> $$
> \mathrm{TER}
> = \frac{1}{N} \sum\_{n=1}^{N}
> \mathbf{1}\!\left( |\hat{f}(\mathbf{X}\_n; \mathbf{w}) - Y\_n| > \tau \right),
> $$
> which measures the proportion of predictions that incur non-trivial errors.
>
> Here, we report the overall results: for variable misspecification, we present the average performance over misspecification levels {0,1,3,5,7}; for structural misspecification, we report the mean over misspecification strengths from 0.1 to 1 in increments of 0.1. The complete set of detailed results is available in **Appendix G**.
>
> |Misspecification|Task|mCoErr(diverse)|mCoErr(same)|EnCoErr(DE)|EnCoErr(EW)|EnCoErr(WDE)|TER(DE)|TER(EW)|TER(WDE)|
> |-|-|-|-|-|-|-|-|-|-|
> |Variable|nested|0.7819|0.8268|0.8815|0.8352|0.6799|0.2503|0.2419|0.2351|
> ||interaction|0.7485|0.8065|0.8352|0.8146|0.6452|0.1532|0.1522|0.1510|
> ||nonlinear|0.7817|0.8261|0.8545|0.8277|0.6826|0.1216|0.1218|0.1202|
> |Inherent|Infinite|0.3420|0.3775|0.6000|0.5804|0.3126|0.1304|0.1104|0.0037|
> ||Square|0.6582|0.7105|0.9472|0.9313|0.6359|0.3034|0.2945|0.2466|
> |structural|$\Delta=30000$|0.7090|0.7425|0.7978|0.7862|0.5034|0.1373|0.1021|0.0980|
> ||$\Delta=50000$|0.7305|0.7683|0.8211|0.8138|0.5104|0.1510|0.1120|0.1080|
> ||$\Delta=100000$|0.6886|0.7182|0.7822|0.7476|0.4733|0.1211|0.0900|0.0857|
>
>
> These metrics jointly indicate that (a) collective blindness is weaker in diverse architectures than in single-architecture settings. (b) weighted ensembles reduce error correlations with individual models, and (c) extreme prediction errors become less frequent. These results offer a quantitative assessment of collective blindness and demonstrate that WDE effectively mitigates it.
>
> **(Q4).** Thank you for the helpful suggestion.
> In the current setup, we use a 6/2/2 split of the data (train / validation / test), and the validation portion is used both for hyperparameter tuning and for estimating the ensemble weights. As the reviewer suggested, one could alternatively adopt a 6/1/1/2 split, where hyperparameter tuning and weight estimation are carried out on two separate validation subsets.
>
> We would like to note that our current approach is still statistically valid: although both steps use the validation data, this dataset remains entirely separate from the training set, so neither hyperparameter tuning nor weight estimation leads to overfitting with respect to the training data. The resulting ensemble weights are still selected based on out-of-sample performance.
>
> That said, the reviewer’s suggested 6/1/1/2 split is indeed a reasonable alternative. The main drawback is reduced data efficiency, as dividing the validation set further decreases the sample size available for each task. For this reason, we chose the more data-efficient 6/2/2 split, which is standard in practice.

---

> > ### Author Response · Authors · 2025-11-23
> >
> > **(Q5).** We thank the reviewer for this helpful question.
> >
> > Condition 1 is a mild moment assumption that ensures uniform concentration of the validation loss. Such boundedness and sub Gaussian moment conditions are standard in nonparametric regression theory and are used only to control the deviation between empirical and population risks, for example see [1] and [2]. If Condition 1 is relaxed so that the errors may have heavier tails or unbounded moments, then the analysis would require alternative assumptions to control the tail behavior, for example, explicit bounds on higher order moments or tail decay rates. Without such additional assumptions, the deviation between empirical and population risks may no longer vanish, and several key steps in the proof would break down. From this perspective, Condition 1 is a convenient sufficient condition that ensures all necessary tail properties hold automatically.
> >
> > Condition 2 is the essential misspecification condition. It requires that the oracle risk does not vanish faster than the sampling error and the estimation error, which ensures that the irreducible approximation error remains the dominant term and that the ratio $R(\widehat{w})/\inf\_{w} R(w)$ is stable. If the oracle risk decreases too fast, the denominator can become too small relative to stochastic fluctuations, and the conclusion of Theorem 3 may fail. Therefore, Condition 2 is fundamental for the asymptotic optimality.
> >
> > [1] Jiao Y, Shen G, Lin Y, et al. Deep nonparametric regression on approximate manifolds: Nonasymptotic error bounds with polynomial prefactors[J]. *The Annals of Statistics*, 2023, 51(2): 691–716.
> >
> > [2] Schmidt-Hieber J. Nonparametric regression using deep neural networks with ReLU activation function[J]. *The Annals of Statistics*, 2020, 48(4): 1875.
> >
> > **(Q6).** We thank the reviewer for the helpful comment. To complement the controlled settings in the main text, we include experiments on the real-world CIFAR-10  and CIFAR-10-C (OOD) dataset. We report the overall performance, while the detailed results are presented in **Appendix F**. Our methods attain the strongest overall performance in this benchmark. We believe these results provide further evidence that our conclusions extend beyond misspecified scenarios and hope they adequately address the reviewer’s question.
> >
> > 1. Accuracy
> > |Dataset|DE(preactresnet18)|DE(resnet18)|DE(stochasticdepth18)|DE(googlenet)|DE(mobilenetv2)|EW|WDE|
> > |-|-|-|-|-|-|-|-|
> > |CIFAR10|91.18%|91.10%|91.20%|90.99%|87.81%|92.48%|**92.76%**|
> > |CIFAR10-C|22.60%|21.46%|19.88%|21.97%|18.73%|23.88%|**25.59%**|
> >
> >
> > 2. Expected Calibration Error (ECE), which measures how well model confidence aligns with its actual accuracy, with lower values indicating better calibration.
> > |Dataset|DE(preactresnet18)|DE(resnet18)|DE(stochasticdepth18)|DE(googlenet)|DE(mobilenetv2)|EW|WDE|
> > |-|-|-|-|-|-|-|-|
> > |CIFAR10|0.0599|0.0564|0.0599|0.0583|0.0768|0.0507|**0.0438**|
> > |CIFAR10-C|0.5889|0.6468|0.5889|0.5614|0.5735|0.6128|**0.4884**|

---

> > ### Comment · Reviewer_DdZD · 2025-11-28
> >
> > I thank the authors for their responses to my questions and for addressing my concerns.
> >
> > My main remaining concern is that while some discussion on related work was provided in the reply from the authors, this related work is not discussed in detail either in the related work section or throughout the paper, which I believe would be very relevant.
> >
> > I will maintain my score.

---

> > > ### Author Response · Authors · 2025-11-28
> > >
> > > Thank you again for your valuable comments and for clarifying your remaining concerns.
> > >
> > > To address your feedback, we have made  updates to the related work section and have incorporated detailed discussions of the relevant prior work.
> > >
> > > We sincerely hope that the additional  experiments and  detailed rebuttal fully address your concerns and encourage you to consider a more positive score.

---

### Author Response · Authors · 2025-11-30

We sincerely appreciate the time and effort that you and the reviewers have devoted to our manuscript.  The main contributions of our paper are summarized below:

1. **We provide the first systematic study of deep ensembles under model misspecification.**
   We formally define misspecification in neural networks through three categories: variable misspecification, structural misspecification, and intrinsic misspecification. We further show that, even in the presence of such misspecification, our method still converges at the fastest rate achieved by the optimal candidate model.
2. **We establish the first  asymtotically optimal theoretical guarantees for the weighting estimator in deep ensembles.**
   Using only observable data, the weighted ensemble attains oracle-level accuracy, and its asymptotic error converges at the same rate as the smallest error among all candidate networks.
3. **We extend classical stacking theory to the setting of neural networks.**
   Traditional stacking results typically apply to linear models and often converge more slowly than $n^{1/2}$. We generalize these results to neural networks, where approximation and stochastic errors interact in more complex ways, and we prove an asymptotic error bound demonstrating that the validation–minimization principle continues to provide meaningful guarantees in this more challenging regime.

---

> ### Author Response · Authors · 2025-11-30
>
> We  further provide a summary below to facilitate your reading and to offer a clearer understanding of our paper:
>
> **[Review 1 Q1, Review 2 W1, Review 3 W1, Review 4 W1] Comparison with existing theoretical work.**
>
> **Comparison with diversity-based ensemble theory:**
> (i) Existing results only show that the ensemble risk is no worse than the uniform average of the individual risks, but they do not compare the ensemble to the best individual model.
> (ii) These results typically apply only to uniform averaging rather than weighted combinations.
>
> **Comparison with PAC-Bayesian theory:**
> (i) PAC-Bayesian guarantees require specifying an explicit likelihood and prior.
> (ii) PAC-Bayesian ensembles rely on probability distributions and therefore cannot choose optimal weights or establish asymptotic optimality as in our setting.
>
> **Comparison with traditional ensemble methods (e.g., stacking):**
> (i) Classical stacking theory usually applies to linear models and often converges at rates slower than $n^{1/2}$.
> (ii) These methods do not obtain the asymptotic optimality of weighted ensembles (as established in Theorem 3 of our paper) and do not consider model misspecification.
>
> **[Review 1 W6, Review 3 W3] Additional Results in CIFAR10:**
>
> To complement the controlled settings in the main text, we include experiments on the real-world CIFAR-10  and CIFAR-10-C dataset. We report the overall performance, while the detailed results are presented in **Appendix F**. Our methods attain the strongest overall performance in this benchmark.
>
> **[Review 1 W3, Review 2 W2,  Review 3 Q1] Illustracton of Collective Blindness:**
>
> The root cause of collective blindness lies in the fact that traditional deep ensemble often rely on models that share the same architecture and differ only in random initialization. So, the diversity among ensemble members is much lower than expected.
>
> To visualize this phenomenon, we plot a t-SNE embedding of CIFAR-10 predictions generated by initializing the same architecture 10 times, we includes four widely used achitecture, covering both shallow (depth-18) and deep (depth-152) networks, see **Figure 3 in Appendix G**. The points corresponding to the same architecture cluster tightly together, revealing that different initializations lead to highly similar predictions. In contrast, incorporating different architectures brings greater diversity, producing predictions that are far less correlated。
> To further quantify the effect of collective blindness, we introduce three metrics：Model–Model Error Correlation, Ensemble–Model Error Correlation and Tail Error Rate. The complete set of detailed results is available in **Appendix G**.
>
> These metrics jointly indicate that (a) collective blindness is weaker in diverse architectures than in single-architecture settings. (b) weighted ensembles reduce error correlations with individual models, and (c) extreme prediction errors become less frequent. These results offer a quantitative assessment of collective blindness and demonstrate that WDE effectively mitigates it.
>
> We further summarize the disussion with reviewers:
>
> **Reviewer 1.**
> We have addressed the majority of this reviewer's concerns. In response to the follow-up comment requesting a more detailed discussion of related work, we have added a dedicated paragraph to the Related Work section. We also provided a thorough comparison in the rebuttal, which is also summarized above.
>
> **Reviewer 2.**
> We believe that our responses have addressed this reviewer’s concerns. The reviewer increased the score to 4 even before we added the model-soup results and updated the related work. After we included the model-soup experiments, the reviewer provided very encouraging feedback. We have responded comprehensively to all points raised.
>
> **Reviewer 3.**
> This reviewer noted that “the rebuttal is quite comprehensive and addresses most of my concerns” and raised the score to 6. We appreciate the reviewer’s positive assessment of our work.
>
> **Reviewer 4.**
> This reviewer did not participate in the discussion phase. Nevertheless, we carefully addressed the concerns raised in the initial review, particularly those related to the positioning of our work within the existing literature. We clarified why the referenced papers do not reduce the novelty of our contributions and highlighted the distinctions of our approach. We hope that our explanations sufficiently address these concerns.
>
>
>
> We have added additional comprehensive experiments in **Appendices F and G** as suggested by the reviewers. We also provided the full details of the proofs in **Appendix B.4**, updated the related work section, and offered detailed and well-supported responses to all concerns raised. We hope these revisions adequately address the reviewers’ comments on our paper. We sincerely thank you once again for your time and for your dedicated efforts

---

### Meta-Review · Area_Chair_DnUg · 2026-01-07

**Summary:**

This paper studies weighted deep ensembles learned by validation-risk minimization in settings where individual networks may be misspecified. The authors formalize several sources of misspecification, argue that uniform averaging can suffer from “collective blindness,” and propose learning simplex weights on a held-out validation set. The main technical claims are asymptotic guarantees that the weighted ensemble matches the rate of the best candidate and can attain oracle-level predictive risk, supported by synthetic experiments and additional real-data results added during rebuttal (including e.g. CIFAR-10/CIFAR-10-C).

Reviewers broadly agree the topic is relevant and the direction is potentially useful. However, there is persistent concern that the paper’s positioning and novelty are not yet sufficiently convincing relative to established work on stacking / super learner / weighted ensembles and ensemble generalization theory, and that several broad claims (“first,” and the framing around “collective blindness” as a central motivating pathology) are not matched by a correspondingly careful and complete treatment in the main text. While the rebuttal improved experiments and clarified multiple points, it did not fully resolve these core concerns.

As an additional note, several bibliography entries contain incorrect or incomplete metadata. The paper would benefit from a reference double-check.

**Reviewer Concerns:**

Reviewer DdZD: Positive overall; main remaining issue is insufficient related-work discussion and positioning of weighted ensembles in the paper text.

Reviewer uRvF: Rebuttal addressed most concerns and added useful evidence; minor practical concern about optimization when convexity is not relied upon.

Reviewer iDai: Improvements acknowledged (formal statements, added experiments/verification), but positioning within stacking/weighted ensemble theory remains a major gap; requests a substantially clearer, properly credited narrative in the main paper.

Reviewer Afi7: Strongly negative on novelty framing and rigor/attribution; disputes “first” claims and highlights multiple issues in presentation and precise statements.

**Reviewer Scores:**

Reviewer DdZD: likely stays 6 (remaining concern is mostly editorial/positioning).

Reviewer uRvF: already moved to 6 and likely stays there.

Reviewer iDai: moved to 4; may remain 4 unless the main text is substantially reorganized to address positioning/attribution.

Reviewer Afi7: likely remains 2, as their critique focuses on core framing/rigor issues.

---

### Decision · Program_Chairs · 2026-01-26

Reject